# Wildfire smoke, Arctic haze, and aerosol effects on mixed-phase and cirrus clouds over the North Pole region during MOSAiC: An introduction

Ronny Engelmann[1], Albert Ansmann[1], Kevin Ohneiser[1], Hannes Griesche[1], Martin Radenz[1], Julian Hofer[1], Dietrich Althausen[1], Sandro Dahlke[2], Marion Maturilli[2], Igor Veselovskii[3], Cristofer Jimenez[1], Robert Wiesen[1], Holger Baars[1], Johannes Bühl[1], Henriette Gebauer[1], Moritz Haarig[1], Patric Seifert[1], Ulla Wandinger[1], and Andreas Macke[1]

[1]Leibniz Institute for Tropospheric Research, Leipzig, Germany
[2]Alfred Wegener Institute, Helmholtz Centre for Polar and Marine Research, Potsdam, Germany
[3]Prokhorov General Physics Institute of the Russian Academy of Sciences, Moscow, Russia

**Correspondence:** R. Engelmann
(ronny@tropos.de)

**Abstract.**

An advanced multiwavelength polarization Raman lidar was operated aboard the icebreaker Polarstern during the MOSAiC (Multidisciplinary drifting Observatory for the Study of Arctic Climate) expedition to continuously monitor aerosol and cloud layers in the Central Arctic up to 30 km height. The expedition lasted from September 2019 to October 2020 and measurements were mostly taken between 85°N and 88.5°N. The lidar was integrated in a complex remote sensing infrastructure aboard the Polarstern. In this article, novel lidar techniques, innovative concepts to study aerosol-cloud interaction in the Arctic, and unique MOSAiC findings will be presented. The highlight of the lidar measurements was the detection of a 10 km deep wildfire smoke layer over the North Pole region between 7–8 km and 17–18 km height with an aerosol optical thickness (AOT) at 532 nm around 0.1 (in October-November 2019) and 0.05 from December to March. The dual-wavelength Raman lidar technique allowed us to unambiguously identify smoke as the dominating aerosol type in the aerosol layer in the upper troposphere and lower stratosphere (UTLS). An additional contribution to the 532 nm AOT by volcanic sulfate aerosol (Raikoke eruption) was estimated to be always lower than 15%. The optical and microphysical properties of the UTLS smoke layer are presented in an accompanying paper (Ohneiser et al., 2021). This smoke event offered the unique opportunity to study the influence of organic aerosol particles (serving as ice-nucleating particles, INP) on cirrus formation in the upper troposphere. An example of a closure study is presented to explain our concept of investigating aerosol-cloud interaction in this field. The smoke particles were obviously able to control the evolution of the cirrus system and caused low ice crystal number concentration. After the discussion of two typical Arctic haze events, we present a case study of the evolution of a long-lasting mixed-phase cloud layer embedded in Arctic haze in the free troposphere. The recently introduced dual-field-of-view polarization lidar technique was applied, for the first time, to mixed-phased cloud observations in order to determine the microphysical properties of the water droplets. The mixed-phase cloud closure experiment (based on combined lidar and radar observations) indicated that the observed aerosol levels controlled the number concentrations of nucleated droplets and ice crystals.

# 1 Introduction

Rapid sea ice loss, unusual Arctic warming, and our incomplete knowledge about the complex processes controlling the Arctic climate motivated the MOSAiC (Multidisciplinary drifting Observatory for the Study of Arctic Climate) (MOSAiC, 2020) expedition, the largest Arctic research initiative in history. On 20 September 2019, the German research icebreaker Polarstern (Knust, 2017) left Tromsø in Northern Norway towards the central part of the Arctic and started drifting through the Arctic Ocean trapped in the ice in the beginning of October 2019. The goal of the MOSAiC expedition was to take the closest look ever at the Arctic as the epicenter of global warming and to gain fundamental insights that are key to better understand global climate change. Hundreds of researchers of more than 70 research institutions from 20 countries were involved in this exceptional expedition. The MOSAiC campaign brought a modern research icebreaker close to the North Pole for a full year especially, for the first time, in polar winter. The mission was spearheaded by the Alfred Wegener Institute, Helmholtz Center for Polar and Marine Research (AWI).

We continuously operated a state-of-the-art multiwavelength aerosol/cloud Raman lidar (Engelmann et al., 2016) aboard the research vessel (RV) Polarstern side by side with the ARM (Atmospheric Radiation Measurement) mobile facility 1 (AMF-1) (ARM, 2020) and collected tropospheric and stratospheric aerosol and cloud profile data throughout the expedition period from September 2019 to October 2020. Our role in the MOSAiC consortium was to provide a seasonally and height-resolved characterization of aerosols and clouds in the North Pole region from the surface up to 30 km height. Our specific focus was to explore the impact of aerosol particles on mixed-phase-cloud and cirrus evolution in the free troposphere up to tropopause level. Aerosol-cloud interaction, especially in the upper troposphere, is poorly understood. Advances in our understanding of the influence of local and long-transported aerosol pollution especially on ice formation processes is, however, of fundamental importance for an improved modelling of atmospheric and climate processes in the Arctic. Clouds in general sensitively influence the energy and water cycles, their accurate representation in models is thus critical for robust future climate projections.

It is noteworthy to mention that Willis et al. (2019) recently pointed out that the majority of our current knowledge about Arctic aerosol (including their impact on cloud evolution) comes from ground-based in situ monitoring stations. However, as outlined in the review of Abbatt et al. (2019) with the increasing number of aircraft observations and advanced satellite remote sensing, especially after the launch of the spaceborne CALIPSO (Cloud-Aerosol Lidar and Infrared Pathfinder Satellite Observation) lidar (Devasthale et al., 2011a, b; Di Pierro et al., 2013; Di Biagio et al., 2018; Yang et al., 2021) it became clear that this view on Artic aerosol conditions is only valid for the lowest several hundreds of meters of the Arctic atmosphere, and only holds for the summer season. Airborne in situ and CALIPSO lidar observations corroborate that the Arctic free troposphere is significantly polluted during the late winter and early spring months.

Our most impressive and outstanding observation during the entire MOSAiC expedition was the detection of a persistent, 10 km deep aerosol layer of aged wildfire smoke (Ohneiser et al., 2021). We monitored this smoke layer in the upper troposphere and lower stratosphere (UTLS) from about 7-8 km up to 17-18 km height for more than seven months from the beginning of the lidar observations in late September 2019 until May 2020. The wildfire smoke layer most probably originated from severe and long-lasting fires in Siberia occurring in July and August 2019. Because of the large number of cirrus clouds,

occurring in the persistent smoke layer, the favorable opportunity was given to investigate, for the first time, the role of aged smoke particles (mainly consisting of organic material) in heterogeneous ice formation processes and to contrast these findings with respective ones for the summer half year when anthropogenic haze and mineral dust dominate and influence cirrus formation in the Arctic (Grenier et al., 2009; Jouan et al., 2012, 2014). Furthermore, a record-breaking reduction of the ozone

concentration, mainly between 15 and 20 km height, was found over the Arctic in the spring of 2020 (DeLand et al., 2020; Wohltmann et al., 2020; Innes et al., 2020; Manney et al., 2020). The ozone-depleted layer partly overlapped with the smoke layer so that the question arose to what extent the wildfire smoke particles contributed to the strong ozone depletion. More discussion is given by Ohneiser et al. (2021).

Our specific research goal is the study of aerosol-cloud interaction with the focus on ice formation in the middle and upper

troposphere by means of active remote sensing (Bühl et al., 2016; Ansmann et al., 2019; Radenz et al., 2021). In the framework of the MOSAiC expeditions we tested several new, recently introduced lidar techniques and new data analysis concepts to investigate the role of aerosol particles in cloud evolution processes. Two examples, a case study on the evolution of mixed-phase cloud evolution in Artcic haze and a case study of the formation of ice clouds in wildfire smoke close to the tropopause, will be presented in this introductory paper. Compared to snapshot-like aircraft observations, active remote sensing allows us to

continuously monitor aerosol-cloud interaction (like a running camera) and thus cloud evolution processes and also to sample hundreds of cloud layers and systems within a short time period.

The article is organized as follows. After a brief description of the instrumentation and lidar products in Sect. 2, key observations are highlighted. An overview of the smoke situation during the MOSAiC winter half year is given in Sect. 3.1. An extended discussion can be found in Ohneiser et al. (2021). In Sect. 3.2, we present two cases of Arctic haze observations

performed in February and March 2020. Then we continue with two aerosol-cloud interaction studies. In Sect. 3.3, we start with a case of a shallow mixed-phase cloud system consisting of a liquid-water-dominated cloud top layer and an extended ice virga zone below the main cloud layer. We present a new lidar technique, the so-called dual field-of-view (FOV) polarization lidar technique (Jimenez et al., 2020a, b) that allows us to retrieve the microphysical properties of the liquid-water droplets in the cloud top layer (including the cloud droplet number concentration, CDNC) and to combine these observations with

lidar-derived estimates of cloud condensation nucleus concentrations (CCNC) around and below the liquid-water cloud layer. Furthermore, the lidar-based estimates of the ice-nucleating particle concentration (INPC) were compared with the ice crystal number concentration (ICNC) derived from combined lidar-radar observations (Bühl et al., 2019) in the framework of so-called cloud closure studies (Marinou et al., 2019; Ansmann et al., 2019).

In Sect. 3.4, the impact of the wildfire smoke on cirrus evolution is finally illuminated. This effort can be regarded as a pilot

study. For the first time, we explore to what extent wildfire smoke (organic aerosol particles) can influence or even control cirrus formation. Organic particles are ubiquitous in the atmosphere around the world (Schill et al., 2020b) and there is an urgent need to investigate their potential to serve as efficient ice-nucleating particles. MOSAiC provides unique observations to make substantial progress in this research field. The importance and relevance of such smoke-cirrus-interaction studies significantly increased during the last 3-4 years with the increasing number of major wildfire events in both, the northern and southern

hemisphere (Baars et al., 2019; Ohneiser et al., 2020). Sect. 4 finally provides some concluding remarks and an outlook.

## 2 Polarstern cruise, lidar setup, and observational products

Figure 1 shows the track of the drifting RV Polarstern from October 2019 to May 2020. Each of the eight red circles along the Polarstern track indicates the beginning of a new month. The highest northern latitude with 88.6°N was reached around 20 February 2020. The Polarstern was at latitudes ≥85°N from the beginning of October 2019 to the beginning of April 2020.

A photograph of the Polarstern in the ice and snow-covered Arctic Ocean is shown in the left panel of Fig. 2 together with a photograph of the main ship-based MOSAiC atmospheric measurement platforms aboard Polarstern (right panel in Fig. 2). Six containers for in situ aerosol monitoring and for active remote sensing of aerosols and clouds with lidars and radars were deployed on the front deck of the research vessel. The ARM mobile facility AMF-1 on the right side consists of three portable shelters containing a baseline suite of instruments, communication, and data systems.

The OCEANET container of the Leibniz Institute for Tropospheric Research (TROPOS), routinely operated aboard Polarstern since 2009 (Kanitz et al., 2011, 2013; Bohlmann et al., 2018; Yin et al., 2019; Baars et al., 2020) is the third one to the back on the left side of the container spots. Besides continuous observations with our multiwavelength Raman lidar (installed inside the air-conditioned container with specially designed roof window for optimum laser-beam transmittance and sampling of backscattered photons), two microwave radiometers for water vapor and cloud liquid-water measurements, two photometers

for aerosol optical thickness (AOT) observations and a two-dimensional disdrometer were operated for ice crystal morphological studies. The TROPOS equipment was already aboard Polarstern two years ago and involved in the Arctic field campaign PASCAL (*P*hysical feedbacks of *A*rctic boundary layer, *S*ea ice, *C*loud and *A*eroso*L*) (Wendisch et al., 2019; Griesche et al., 2020a, b).

### 2.1 Lidar instrument and operational details

Two advanced lidar instruments, the multiwavelength Raman lidar Polly (*PO*ortab*L*e *L*idar s*Y*stem) (Engelmann et al., 2016) and a High Spectral Resolution Lidar (HSRL) (Eloranta, 2005) were operated continuously aboard the drifting RV Polarstern during the MOSAiC expedition. These two lidars are complementary regarding their capabilities. While the multiwavelength Raman lidar delivered detailed spectrally resolved optical properties of the aerosol in the Arctic and retrieval products regarding microphysical and cloud-relevant aerosol properties from October to March, i.e., during nighttime conditions, the single-

wavelength HSRL is of advantage during the summer half year (when Raman lidar observations are of limited use) and allows us to measure profiles of the climate-relevant particle extinction coefficient at 532 nm wavelength even during sunlight conditions. Both lidar systems are polarization lidars and thus permit detailed monitoring of cloud phase and thus of liquid-water, mixed-phase and ice clouds and morphological features (sphericity) of aerosol particles.

The MOSAiC expedition provided for the first time the unique opportunity to perform lidar observations north of 85°N

over the entire winter half year 2019-2020. This part of the Central Arctic is not covered by any other lidar measurement, neither by observations with the spaceborne CALIPSO lidar (only up to 81.8°N) nor by measurements of the ground-based Arctic lidar network (Nott and Duck, 2011). Thus, we add a new data set to the global aerosol data base. Di Biagio et al. (2018) were the first to run lidars (mounted on an ensemble of autonomous drifting buoys) in the Central Arctic, even north

of 82°N, to perform year-around aerosol profiling, including the winter half year of 2015-2016. These measurements together with respective CALIPSO lidar observations are used in our contrasting analysis regarding the aerosol conditions during the MOSAiC year 2019-2020 and the year 2015-2016 characterized by unperturbed aerosol conditions.

The setup and basic technical details of the Polly instrument are given in Engelmann et al. (2016), Hofer et al. (2017), and Jimenez et al. (2020b). Polly belongs to the lidar network PollyNET (Baars et al., 2016) which is part of the European Aerosol Research Lidar Network (EARLINET) (Pappalardo et al., 2014) organized within the Aerosols, Clouds and Trace gases Research InfraStructure (ACTRIS) project (ACTRIS, 2020). The lidar transmits linearly polarized laser pulses at 355, 532, and 1064 nm with a pulse repetition rate of 20 Hz. All laser beams are pointing to an off-zenith angle of 5° to avoid a bias in the observations of the optical properties of mixed-phase and cirrus clouds caused by strong specular reflection by falling and then frequently horizontally aligned ice crystals (Noel and Sassen, 2005; Noel and Chepfer, 2010; Avery et al., 2020).

The receiver unit consists of a near-range receiver part, optimized to provide aerosol and cloud optical properties from 120 m above the surface to several kilometers height, and a far-range receiver part which permits accurate aerosol and cloud profiling from about 800 m to 30-40 km height. Thirteen receiver channels are available to collect the following lidar return signals: elastically backscattered photons at the three laser wavelengths, the cross-polarized signal components at 355 and 532 nm at two different field of views (FOV), the vibrational Raman signals of nitrogen at 387 and 607 nm, and of water vapor at 407 nm. All signals are measured with photon-counting photomultipliers and stored with 30 seconds of resolution. We introduce "cross" and "co" to indicate the plane of polarization orthogonal and parallel to the plane of linear polarization of the transmitted laser pulses.

The recently introduced and implemented dual-FOV polarization lidar option (Jimenez et al., 2020a, b) permits the measurement of the cross and total (cross-polarized + co-polarized) signal components at 532 nm at two different FOVs. The dual-FOV lidar technique enables us to determine multiple scattering by droplets in liquid-water-dominated cloud layers and to retrieve from this multiple scattering information cloud microphysical properties (e.g., effective droplet size and cloud extinction coefficient) (Jimenez et al., 2020a). The method is based on the measurement of two volume linear depolarization ratios at two different FOVs. The depolarization ratio is defined as the ratio of the cross-polarized to the co-polarized backscatter coefficient.

The Polly instrument is designed for automated continuous profiling of aerosols and clouds and thus was running around the clock. Well-defined breaks were necessary to exchange laser flash lamps, to run different calibration procedures, to check the full setup, and to perform an overall alignment of the Polly instrument. Five TROPOS lidar scientists took care of the OCEANET instrumentation over the one-year MOSAiC expedition period.

## 2.2 Lidar products

The measurement and retrieval products obtained from the multiwavelength dual-FOV polarization and Raman lidar Polly are listed in Table 1. In addition, typical relative errors caused by signal noise and needed input parameters in the data analysis are given in the table. The basic data analysis procedure to obtain aerosol and cloud optical properties are described by Baars et al. (2012), Baars et al. (2016), Hofer et al. (2017), Haarig et al. (2018), and Ohneiser et al. (2020, 2021). Height profiles of the particle backscatter coefficient at the laser wavelengths, of the particle extinction coefficient at 355 and 532 nm, respective

extinction-to-backscatter ratio (lidar ratio) at 355 nm and 532 nm, the particle linear depolarization ratio at 355 nm and 532 nm, and the mixing ratio of water vapor to dry air by using the Raman lidar return signals of water vapor and nitrogen (Dai et al., 2018) can be determined. The Raman lidar method was exclusively applied to determine the particle backscatter and extinction profiles (Ohneiser et al., 2021).

The particle linear depolarization ratio (PLDR) is obtained from measured volume linear depolarization ratio (VDR, depolarization ratio influenced by Rayleigh and particle depolarization of backscattered laser light). The PLDR can be used to discriminate spherical aerosol particles such as haze particles and liquid droplets, producing PLDR close to zero, from non-spherical particles such as dust particles or ice crystals, causing PLDR around 0.3 and typically >0.4 at 532 nm, respectively. In the case of Polly, the cross-polarized and total (cross-polarized + co-polarized) backscatter signals are measured. The specific

approach to obtain VDR from the Polly observations is described by Engelmann et al. (2016).

Although PollyNET delivers automatically calculated PLDR profiles, the lidar observations presented in this article and in the accompanying article of Ohneiser et al. (2021) were manually analyzed. To keep the uncertainties in the derived aerosol quantities at an acceptably low level of <10% for the backscatter coefficients and depolarization ratios and of the order of 20–25% for the particle extinction coefficients and lidar ratios, large vertical signal smoothing and regression window lengths

of 500 to 2500 m and signal averaging times from 30 minutes to more than 10 hours had to be applied. Fortunately during the winter half year, the nighttime conditions allowed us to apply the Raman lidar methods around the clock. More details are given in Ohneiser et al. (2021).

Auxiliary data are required in the lidar data analysis in form of temperature and pressure profiles in order to calculate and correct for Rayleigh backscatter, extinction, and light-depolarization contributions to the measured lidar return signals. In the

MOSAiC data analysis, we used the Polarstern radiosonde observations. As an important contribution to MOSAiC, radiosondes were routinely launched every 6 hours (at 5, 11, 17, and 23 UTC) throughout the entire MOSAiC period (Maturilli et al., 2021).

The retrieval of aerosol microphysical properties such as particle volume, mass, and surface area concentration and estimates of cloud-relevant properties such as CCNC and INPC is performed by means of the POLIPHON (Polarization Lidar Photometer Networking) approach (Mamouri and Ansmann, 2016, 2017; Marinou et al., 2019; Ansmann et al., 2019, 2020). Lidar input

data sets are the height profiles of the 532 nm backscatter coefficient and the PLDR. Hofer et al. (2020), for example, shows the full set of height-resolved POLIPHON aerosol products in the cases of an 18-month Polly campaign in Dushanbe, Tajikistan, for central Asian aerosol. Alternatively to the POLIPHON method, we applied the multiwavelength lidar inversion technique (Müller et al., 1999, 2014; Veselovskii et al., 2002, 2012) to derive the microphysical properties of aerosol particles including the particle size distribution. The method of Veselovskii et al. (2002) is applied to MOSAiC smoke and Arctic haze observations

to estimated layer mean microphysical properties in pronounced smoke and haze layers with sufficiently accurate backscatter coefficients at 355, 532, and 1064 nm and extinction coefficients at 355 and 532 nm.

Details of the analysis of the dual-FOV polarization lidar measurements can be found in Jimenez et al. (2020a, b). The method allows us to retrieve the cloud extinction coefficient and the effective radius of the droplets at a height of 50-100 m above cloud base and, in the next step, to calculate the liquid water content (LWC, function of cloud extinction coefficient and

effective radius) and cloud droplet number concentration from LWC by assuming a gamma size distribution.

Relative humidity fields are obtained from the mixing-ratio measurements (Dai et al., 2018) (Raman lidar method) and temperature profiles measured with the Polarstern radiosondes. Quality checks of the continuously obtained water vapor fields were based on comparisons with the relative-humidity profiles obtained with radiosondes launched at 5, 11, 17, and 23 UTC.

A good knowledge of the tropopause height is important when profiling aerosol and cloud layers in the troposphere and stratosphere. The tropopause was computed from the MOSAiC radiosonde temperature and pressure profiles (Maturilli et al., 2021) by using the approach of the Global Modeling and Assimilation Office (GMAO), Goddard Space Flight Center, Greenbelt, Maryland, USA (GMAO, 2021). In this approach, the tropopause height $z_{TP}$ is found from the height profile of the difference $\alpha T(z) - \log_{10} p(z)$ with $\alpha = 0.03$, temperature $T$ (in Kelvin), pressure $p$ (hPa), and height $z$ (m). The tropopause pressure $p(z_{TP})$ is defined as the pressure where the defined difference reaches its first minimum above the surface. If no clear minimum was found up to $z = 13000$ m over Polarstern, a tropopause height $z_{TP}$ was not assigned. The obtained tropopause heights agree well with the ones we obtained by applying the definition of the World Meteorological Organization (WMO, 1992) to the radiosonde temperature profiles and considering refinements in the determination described by Klehr (2012). In most cases, the GMAO approach delivered 20-80 m lower tropopause levels and produced less outliers.

The large spectrum of retrieved aerosol and cloud microphysical properties forced us to use any favorable opportunity for comparison with airborne in situ observations of these quantities in the framework of validation experiments in the past. These combined lidar and airborne in situ observations to characterize the quality of the retrieval products and to check the respective uncertainty ranges were usually available during large field campaigns. Regarding the aerosol microphysical properties, the comparisons showed that particle number concentrations, surface area, volume and mass concentrations can be obtained with an uncertainty of 25-50% (see Table 1) in cases with a clearly dominating aerosol type, e.g., in dense desert dust plumes or lofted wildfire smoke layers (Wandinger et al., 2002; Groß et al., 2016; Mamali et al., 2018; Haarig et al., 2019). The size distributions of aerosol particles can be precisely identified and estimated from multiwavelength lidar measurements in cases with a pronounced accumulation mode (Müller et al., 2004) as it is the case for aged wildfire smoke and aged Arctic haze.

Validation studies are especially valuable to characterize the potential of lidar to estimate cloud-relevant aerosol parameters such as CCNC and INPC. Comparisons with airborne in situ measurements showed that CCNC can be obtained with an uncertainty of about 30% (inversion of multiwavelength data) to 50% (conversion of the 532 nm extinction coefficient) when the aerosol type (and thus the typical aerosol size distribution) is known, and about a factor of 2 if the aerosol type is not well known or mixtures of different aerosol types prevail (Düsing et al., 2018; Haarig et al., 2019; Georgoulias et al., 2020).

In a few efforts, the potential of lidar to deliver trustworthy INPC profile information was investigated based on simultaneous airborne in situ and lidar observations (Schrod et al., 2017; Marinou et al., 2019). These few attempts indicated that an INPC estimation is possible with an uncertainty within an order of magnitude. The large uncertainty is caused by the used INP parameterization (taken from the literature) and not from the aerosol input parameters obtained from the lidar observations. In this field, much more coordinated field activities are needed. One alternative approach is to compare the derived INP concentrations with the estimated ice crystal number concentration from lidar-radar observations in so-called closure experiments. Good consistency in these closures indicate then a high reliability of the selected INP parameterization (Marinou et al., 2019; Ansmann et al., 2019).

All these uncertainties in the aerosol retrievals can be considerably larger if the aerosol fraction of interest contributes to a minor part to the observed aerosol amount, i.e., when, for example, the interesting mineral dust fraction in Arctic haze is of the order of 10%. For this reason, accompanying airborne in situ observations are always useful to better characterize the uncertainty range of the aerosol and cloud products gained from active remote sensing.

Concerning microphysical properties of liquid-water cloud layers (cloud droplet number concentration, CDNC) and cirrus clouds (ice crystal number concentration, ICNC), validation studies based on aircraft-lidar comparisons are difficult because of the usually strong vertical and horizontal inhomogeneities in the cloud properties. Here, we make use of different, independent methods and techniques to derive estimates of CDNC and ICNC to check the quality of the retrievals (regarding CDNC see, e.g., Jimenez et al., 2020b). Product evaluation efforts have been reported by Casenave et al. (2019) in the case of the retrieval of ice crystal microphyscial properties from combined lidar-radar observations and were based on comparison with respective satellite retrievals or airborne in situ observations.

## 3 Observations

We begin with a series (snapshots) of typical aerosol and cloud scenes observed with our lidar during the winter months. According to Fig. 1, the Polarstern moved very slowly with the pack ice in December, January, and February and was mostly located between 86°N and 88°N. The exceptionally strong polar vortex of 2019-2020 was well established during that time. Figure 3 shows a 10-day measurement sequence (2-11 Dec 2019). Complex features of aerosol layering, cirrus evolution (Fig. 3a-d), and mixed-phase cloud life cycles (Fig. 3e-f) were found. Clear, fog, and cloud free periods occurred frequently as well and provided excellent conditions for a detailed characterization of Arctic haze and wildfire smoke.

The measured volume linear depolarization ratio (VDR) in the right panels of Fig. 3 allows us to precisely distinguish cirrus from layered mixed-phase clouds as explained above. Ice crystals cause large depolarization ratios (green to red colors in Fig. 3b, d, f), and, in contrast, liquid-water layers produce rather low depolarization ratios around zero in Fig. 3f. The increase of the depolarization values with increasing penetration of the laser beam into the water cloud layer is caused by multiple scattering by the cloud droplets. This aspect is explained in more detail in Sect. 3.3. The depolarization ratio of aerosol particles was found to be generally small (Fig. 3h) in the free troposphere and stratosphere which indicates spherical haze and smoke particles.

### 3.1 Wildfire smoke layer in the UTLS regime

Extreme and long-lasting wildfires in central and eastern Siberia, in closest neighborhood to the Arctic region, were most probably responsible for the UTLS smoke layer over the High Arctic in the winter half year of 2019–2020 (Johnson et al., 2021; Ohneiser et al., 2021). The main burning phase lasted from 19 July to 14 August 2019. In the absence of any pyrocumulonimbus (pyroCb) convection (Fromm et al., 2010) and thus of a very efficient and fast process to transport smoke into the UTLS height range, we hypothesize that the smoke was lifted by the so-called self-lifting process (Boers et al., 2010). Details are given in Ohneiser et al. (2021). In self-lifting processes, strong absorption of solar radiation by the black-carbon-containing smoke

heats the air in lofted smoke plumes. The created buoyancy then forces the smoke plumes to ascent up to the tropopause (at around 10 km) and into the lower stratosphere within about 3-5 days according to our simulations studies (Ohneiser et al., 2021).

Once in the UTLS height range, smoke particles become quickly distributed over a large parts of the Northern Hemisphere within a few weeks, as observed and documented for the first time in 2001 (Fromm et al., 2008) and recently confirmed after the record-breaking Canadian fires in the summer of 2017 (Khaykin et al., 2018; Baars et al., 2019). In the case of the strong Siberian fires, the smoke particles became probably quickly distributed over the entire Arctic region (as suggested by the satellite observations presented by Johnson et al. (2021)) and then remained in the UTLS height range for months. The decay of smoke-related stratospheric perturbations takes usually more than a half year (Baars et al., 2019).

Besides the strong and long-lasting fires in Siberia, the Raikoke volcano erupted in June 2019 (Kloss et al., 2021; Vaughan et al., 2021). The amount of the emitted $SO_2$ suggested that a stratospheric aerosol layer with a maximum 532 nm AOT of around 0.025 at latitudes $>50°$ would form. It was further expected that the AOT decreased to values of 0.01 over the High Arctic in autumn 2019 and to 0.005 during the winter months, as discussed in Ohneiser et al. (2021).

From the first days of the MOSAiC observations on, in late September 2019, we observed a pronounced and well-aged stratospheric aerosol layer with a backscatter maximum around or just above the tropopause, smooth internal structures, and clear wildfire smoke signatures. The 532 nm AOT was around 0.1 in autumn 2019 over the Polarstern and thus much higher than the expected AOT of 0.01. Further lidar observations at Leipzig (51.3°N, 12.4°E), Germany, and Ny-Ålesund, Svalbard (78.9°N, 11.9°E), Norway, indicated a strong increase of the stratospheric AOT in August 2019, probably as a result of the record-breaking Siberian fires, and the spread of the smoke over large parts of the northern hemisphere.

Figure 4 presents the optical properties of the smoke layer as measured on 11 December 2019 (Fig. 3g and h). The Polarstern was at 86.6°N and 120.7°E at 12:00 UTC. The lidar observations were averaged over the entire day, thus 24-h mean height profiles are shown. The UTLS aerosol layer extended from 8 to more than 18 km height. The internal vertical structures were rather smooth. The layer base as indicated in Fig. 4 was determined by the altitude at which the 1064 nm backscatter coefficient started to increase with height. The smoke layer top was set to the height where the total-to-Rayleigh backscatter ratio at 1064 nm dropped below a value of 1.1.

A clear and unambiguous optical fingerprint for the presence and dominance of aged smoke particles (after long-range transport) is the unusual spectral dependence of the extinction-to-backscatter ratio $L$ (Fig. 4d) together with a high 532 nm particle lidar ratio $L_{532}$ typically exceeding 70 sr. $L_{532}$ is much larger than the 355 nm lidar ratio $L_{355}$, typically by more than 20 sr (Haarig et al., 2018; Ohneiser et al., 2021). Another smoke signature is the weak wavelength dependence of the particle extinction coefficient, expressed in terms of the extinction-related Ångström exponent $Å_{\sigma,355,532} = \ln(\sigma_{355}/\sigma_{532})/\ln(532/355)$. For the extinction profiles in Fig. 4c, $Å_{\sigma,355,532}$ is about 0.5. In the case of volcanic sulfat aerosol the Ångström exponent is typically clearly $>1.0$. Note, that such an unambiguous identification of the wildfire smoke type is only possible with multiwvalength Raman lidars and high-spectral-resolution lidars (Müller et al., 2014; Burton et al., 2015) which allow us to determined particle extinction coefficient at several wavelengths.

The particle depolarization ratios at 532 nm (around 1%) and at 355 nm (1-2%) in Fig. 4b were very low. Such low values are indicative for spherical particles. We hypothesize that the comparably slow ascent (over several days) of the smoke particles up to the UTLS regime over Siberia in the summer of 2019 allowed the smoke particles to complete the aging process in the humid troposphere (with high levels of condensable gases). At the end of the particle aging process, the majority of the smoke particles showed a spherical core-shell structure and thus caused the low particle depolarization ratios. In contrast, in the case of pyroCb-related ascents (fast lifting within 30-90 minutes up to the tropopause) there is no time for particle aging. Then, the particles widely keep their original, non-spherical shapes and thus produce large depolarization ratios of about 20% at 532 nm.

Figure 5 presents several volume size distributions of the smoke particles. The volume size distributions were obtained from the Polly observation by applying the lidar inversion method to the layer-mean three backscatter and two extinction coefficients (Veselovskii et al., 2002). As typical for aged wildfire smoke, a well-defined accumulation mode was found. A distinct coarse mode was absent. The findings agree well with in-situ observations of long-transported aged smoke (Fiebig et al., 2003; Petzold et al., 2007; Dahlkötter et al., 2014).

An overview of the smoke conditions during the MOSAiC winter half year (October to April) is presented in Fig. 6. Most of the time, the smoke layer was observed between 7 and 17 km height with the backscatter maximum just above the tropopause. A trend of downward movement of the layer is not visible. The maximum extinction coefficients (532 nm) decreased with time from values $>10\,\mathrm{Mm}^{-1}$ in October and November to $<5\,\mathrm{Mm}^{-1}$ in April 2020 (Fig. 6a).

The AOT in Fig. 6b was computed from the particle backscatter height profiles. The directly determined extinction profiles were too noisy, especially in the upper part of the smoke layer. The 532 nm backscatter coefficients were multiplied with the smoke mean lidar ratio of 85 sr (computed from all smoke observations measured during the winter half year) (Ohneiser et al., 2021). Subsequently, we integrated the extinction values between the smoke layer base and top heights as given in Fig. 4 to obtain the AOT. During the winter months, especially in January to February 2020, polar stratospheric clouds (PSC) were frequently observed at the top and above the smoke layer, from 17 to 25 km height. We removed these PSC-affected parts from the height profiles of backscatter and extinction coefficients before we calculated the vertical-column-integrated smoke optical properties. However, several weak PSCs developed within the smoke layer, and in this case, the backscatter and extinction contributions from these optically thin PSCs were not removed. The PSC-related uncertainty in the 532 nm AOT was estimated to be of the order of 5% (Ohneiser et al., 2021).

In terms of the 532 nm AOT in Fig. 6b, the perturbation decreased from 0.05–0.12 in October and November to values of 0.03-0.06 in December to mid of March, and dropped to 0.01-0.02 in April 2020. Almost constant AOT conditions were observed from 10 December to 10 March. During that time period the strong polar vortex established, dominated the airflow, even below the vortex, and controlled the horizontal and vertical exchange of gases and particles. As mentioned by Ohneiser et al. (2021), from mid December 2019 to mid April 2020, the weather pattern was rather stable and meridional air-mass exchange was widely suppressed. The vortex started to collapse around 20 April (Lawrence et al., 2020).

The 532 nm AOT of the volcanic sulfate aerosol was estimated to be about 0.01 in October 2019 and 0.005 during the winter months January and February 2020. According to the estimations in Ohneiser et al. (2021), the Raikoke aerosol fraction was always lower than 15%.

The layer-mean 532 nm smoke extinction coefficients in Fig. 6c (obtained from the ratio of AOT divided by the layer geometrical depth in Fig. 6a) were on the order of 10 Mm$^{-1}$ in October, around 4-5 Mm$^{-1}$ during the main winter months and mostly $\leq$3 Mm$^{-1}$ at the end of the life time of the smoke layer. From the measured layer mean extinction coefficients, mass concentrations of the smoke particles were derived (Ansmann et al., 2020) and ranged from 0.4-2 $\mu$g m$^{-3}$ during the autumn and winter months. Note that AOT values for a clean stratosphere are around 0.001-0.002 (Sakai et al., 2016; Baars et al., 2019), minimum extinction coefficients of the order of 0.1-0.2 Mm$^{-1}$. Minimum stratospheric mass concentrations (at mid latitudes) are close to 0.01-0.02 $\mu$g m$^{-3}$ (Baars et al., 2019).

A PSC observation is shown in Fig. 7 at the end of this section. According to the PSC classification scheme (Achtert and Tesche, 2014), we observed type Ib PSCs. This type is made up of supercooled liquid ternary solutions that likely consist of H$_2$SO$_4$, HNO$_3$, and H$_2$O. In contrast to type Ia and II PSC particles (crystals) the liquid droplets of PSC type Ib only slightly depolarize incoming laser light. The temperature at PSC base height showed values of $-78°$C and at the backscatter maximum the Polarstern radiosonde measured a temperature of $-86°$C. We observed a much lower number of PSCs over the North Pole region (86° to 88.6°N) than the CALIPSO lidar (60° to 81.8°N) during the main PSC period from January to March (CALIPSO, 2020a, b).

## 3.2 Arctic haze vertical structures

The original and primary goal of the shipborne MOSAiC lidar measurements was to provide, for the first time, a height-resolved characterization of tropospheric aerosols and clouds over the North Pole region during the winter half year. Because of its importance for the climate and environmental conditions, Arctic haze has been intensively studied since more than 50 years (Law et al., 2014; Willis et al., 2018; Abbatt et al., 2019). However, knowledge about the vertical layering structures of these aged haze aerosols and about the composition and microphysical properties is still limited and mostly based on sporadic aircraft observations performed during field campaigns, preferably in spring (de Villiers et al., 2010; Quennehen et al., 2012; Ancellet et al., 2014). Ritter et al. (2016) presented a ground-based multiwavelength polarization Raman lidar study on Arctic haze in terms of backscatter, depolarization, and lidar ratios. This situation improved since 2006 with the start of the CALIPSO mission Di Pierro et al. (2013); Di Biagio et al. (2018); Yang et al. (2021).

As summarized by Willis et al. (2018), long-range transport of cold, polluted air masses from northern Eurasian source regions (mainly north of the Arctic front) prevails in winter and leads to the formation of Arctic haze with highest mass concentrations in late winter and early spring. Such a low-level aerosol transport is missing in summer. Stohl (2006) pointed out that the winter transport of aerosols towards the high Arctic occurs at low heights. Long-transported aerosols may reach heights up to the middle troposphere (5-7 km height).

In Fig. 8, we present two MOSAiC cases of Arctic haze observed on 4 February and 4 March 2020. RV Polarstern was drifting with the ice at latitudes of 87.5°N (4 February) and 88.1°N (4 March). The most striking feature in both figures is that aerosol layers occurred almost everywhere within the troposphere and in the lower stratosphere. Remnants of PSCs were visible around 13.5–14 km height on 4 February 2020 (Fig. 8a). Temperatures were around $-76°$C at 13.5 km height and thus sufficiently low to allow formation of type Ia PSC (Achtert and Tesche, 2014; DeLand et al., 2020). Type Ia PSCs are thought

to consist of nitric acid trihydrate (NAT) crystals and produce significant depolarization of backscattered laser light. Note, that the Polarstern was fully below the strong polar vortex from the beginning of January to mid-April 2020 (Ohneiser et al., 2021).

According to the backward trajectory analysis shown in Fig. 9, the aerosol pollution in the pronounced aerosol layers around 5 km (4 February, 11 UTC, Fig. 8a) and 4 km height (4 March, 11 UTC, Fig. 8b) originated from central and western European regions (4 February) and from Russia and the Black Sea area (4 March). The height-resolved trajectory analysis indicated that most of the aeorosol circled around in the Arctic (at latitudes > 70°N) for several days before crossing the Polarstern.

In Fig. 10, the optical properties of Arctic haze for the two cases are illustrated. 12-hour (4 February) and 18-hour (4 March) mean height profiles of the basic lidar products (backscatter, extinction, extinction-to-backscatter ratio) are shown. The long signal averaging times were needed to keep the signal-to-noise related uncertainty below 10% (backscatter coefficient), 20% (extinction coefficient), and 25% (lidar ratio). In both cases, we found a near surface layer up to about 2.5 km height and a separated lofted aerosol layer up to 5 and 7 km height. A clear wavelength dependence of the backscatter and extinction coefficients was found on 4 March, as expected for fine-mode dominated particles in the Arctic (Quennehen et al., 2012) consisting of a mixture of anthropogenic haze and fire smoke (Wang et al., 2011). The Ångström exponent for the extinction coefficient was around 1.7 in the lofted layer above 3 km height and the lidar ratios were high with values up to 100±15 sr. This is indicative of the presence of small, strongly light-absorbing particles.

On 4 Februay 2020 (Fig. 10a), the aerosol optical properties were less well defined, the extinction coefficients were almost equal at both wavelengths (355 and 532 nm), and the noisy lidar ratios at 532 nm were larger than at 355 nm in the lofted layer above 3 km height which is typical for aged wildfire smoke as discussed in the foregoing section. The lidar ratios were lower than on 4 March, i.e., less absorbing particles were present. The volume depolarization ratios (not shown) were rather low in both cases and indicated the dominance of aerosol pollution. The 532 nm AOT was close to 0.025 (4 February, for the lowest 7 km height) and 0.02 (4 March, for the lowest 5 km), and thus considerably lower than the UTLS smoke AOT of about 0.04–0.05. The uncertainty in the tropospheric AOT values is about 20%.

The values for the extinction coefficients of 2-8 $Mm^{-1}$ in the lower layer (up to 2.5 km height) and 1-7 $Mm^{-1}$ in the lofted layer above 3 km height are in good agreement with CALIPSO lidar observations (Di Biagio et al., 2018; Yang et al., 2021). Di Biagio et al. (2018) showed 14-day mean and layer mean values of 2-8 $Mm^{-1}$ (0-2 km layer), 2-10 $Mm^{-1}$ (2-5 km), and 1-2 $Mm^{-1}$ (5-10 km layer) measured in the area from 5-25°E (north of Svalbard) and 80-82°N in February and March 2015. Yang et al. (2021) analyzed 13 years of CALIPSO observations (June 2006 to December 2019) for the Arctic (entire area from 65°N to 81.8°N) and found winter mean extinction values of 2-4 $Mm^{-1}$ (2-6 km height) and around 2 $Mm^{-1}$ (6-10 km height).

Our findings are also in good agreement with the results previously published and obtained during major field activities such as POLARCAT-IPY (Polar Study using Aircraft, Remote Sensing, Surface Measurements, and Models, of Climate, Chemistry, Aerosols, and Transport, International Polar Year) (Law et al., 2014) as well during ARCTAS (Arctic Research of the Composition of the Troposphere from Aircraft and Satellites) (Wang et al., 2011). Both campaigns took place in the spring of 2008. Arctic haze was mostly found up to 7 km height. The aerosol composition was analyzed in large detail based on aircraft observations.

Wang et al. (2011) summarized and reviewed earlier Arctic observations and compared observations with model calculations and found that the ratio of black carbon (BC) to organic aerosol (OA) is high with values of 0.1–0.15 for aerosols advected from Russia. This aerosol is a mixture of anthropogenic haze (sulfate aerosol) and domestic, forest, and agricultural fire smoke (organic aerosol). Sulfate aerosol prevails in the near-surface air, whereas OA becomes comparably large in the free tropo-sphere. This was observed during the POLARCAT-IPY and ACTRAS campaigns. Agricultural fires during spring (Europe, Asia) and flaring of natural gas (Russian oil industry) were found to be important sources for BC and OA particles. The fire smoke mix with anthropogenic haze mainly from Asia during the long-range transport towards the central Arctic, beginning in late winter with peak occurrence in the spring season. Agricultural and forest fires produce BC/OA ratios of typically <0.1 whereas anthropogenic haze may cause ratios >0.15.

## 3.3 Mixed-phase cloud evolution in Arctic haze

Two MOSAiC case studies of aerosol-cloud interaction are presented next. In this section, the evolution of a long-lasting mixed-phase altocumulus layer in Arctic haze within the lower free troposphere is discussed and, in the next section, we present a cirrus development in the lower part of the UTLS wildfire smoke layer.

The mixed-phase cloud system was shown in Fig. 3e and f. The cloud layer was observed for more than seven hours over the Polarstern on 10 December 2019. The dark band in the depolarization ratio panel between 2-3 km height in Fig. 3f indicates the liquid-water-dominated cloud top layer. The increase of the depolarization ratio above the dark zone at the liquid cloud base is caused by multiple scattering at cloud droplets. Favorable conditions with cloud top temperatures around $-28.5°C$ at 2.6 km height (at 03:00 UTC) were given for heterogeneous ice formation via immersion freezing, i.e., ice nucleation on INPs immersed in the water droplets (Kanji et al., 2017). After nucleation in the cloud top layer, ice crystals grow fast to sizes of 50–100 $\mu$m within minutes (Bailey and Hallett, 2012) and immediately start to fall out. As visible in Fig. 3e and f, the crystals formed long virga below the shallow altocumulus layer. The ice crystals partly evaporated on the way down, but partly reached the ground as precipitation. The liquid-water-dominated cloud top layer was not depleted at any time during the seven hour period.

About 40 long-lasting mixed-phase cloud events (ice-precipitating shallow altocumulus decks) with duration from 4-30 hours were observed from October 2019 to March 2020. Because of their sensitive influence on radiative transfer and the water cycle they have been in the focus of research since more than 15 years (Verlinde et al., 2007; Mauritsen et al., 2011; Morrison et al., 2012; Paukert and Hoose, 2014; Loewe et al., 2017; Andronache, 2018; Eirund et al., 2019). However, because of the complexity of influencing meteorological and aerosol aspects, there are still many open questions concerning their long lifetime, especially of the longevity of liquid-water layers and thus of water droplets in the presence of ice crystals. MOSAiC contributes to this research field by means of combined lidar and radar observations aboard RV Polarstern.

We applied our recently developed dual-FOV polarization lidar method (Jimenez et al., 2020a, b) to derive the microphysical properties of liquid-water droplets in the cloud top layer. The results are shown in Fig. 11. The dual-FOV lidar technique was originally designed for pure liquid-water cloud observations but can be applied to mixed-phase clouds as long as backscattering by ice crystals is negligible compared to droplet backscattering in the cloud top layer. This condition holds here with ice crystal

backscatter coefficients of 5-10 $Mm^{-1}$ $sr^{-1}$ in the virga (not shown) and thus probably also in the cloud top layer with droplet backscatter coefficients of the order of 700 $Mm^{-1}$ $sr^{-1}$ (not shown). In the case of crystal-to-droplet backscattering of 0.01, the contribution of ice crystals to the observed multiple scattering features, from which the microphysical properties of the droplets are retrieved, can be ignored. Note, that the method delivers the time series of droplet effective radius, CDNC (cloud droplet number concentration), and cloud extinction coefficients in Fig. 11 for the height of 75 m above cloud base. Thus, the properties of freshly formed droplets are mainly observed.

As can be seen, the retrieved CDNC values were around 20 $cm^{-3}$ in the beginning and around 100 $cm^{-3}$ in the cloud base region later on. With increasing CDNC the effective radius (a characteristic droplet size) decreased and vice versa as expected when assuming a constant water vapor reservoir for droplet nucleation. The cloud extinction coefficient showed typical values from 10-20 $km^{-1}$ in the first half of the cloud life time. Uncertainties in the lidar products are indicated by error bars (one standard deviation) and are of the order of 20-25% (cloud extinction coefficient, droplet effective radius) and 50% (CDNC).

In the next step, we wanted to know how many cloud condensation nuclei (CCN) were available. According to the HYSPLIT backward trajectories in Fig. 12, the altocumulus layer developed in aged Arctic haze, originating from northern parts of Europe. In order to estimate the CCN concentration ($n_{CCN}$ in Fig.13) just below the altocumulus layer, the lidar observations in clear skies (before 03:00 UTC) were analyzed. The result is shown in Fig.13. In this specific measurement, we used the multiwavelength Raman lidar technique and applied the lidar inversion method (see Sect. 2.2) (Veselovskii et al., 2002) to a set of three backscatter and two extinction coefficients. Here, the data analysis was based on the mean backscatter and extinction values (for the height range from 1500–2000 m height). The goal was to obtain (a) a value for the mean aerosol particle number concentration $n_{50}$ (particles with radius >50 nm.) and (b) a ratio of $n_{50}$ to the 532 nm backscatter coefficient. We assumed that the obtained $n_{50}$ value is a good estimate for $n_{CCN}$ (for a supersaturation level of 0.2%) (Mamouri and Ansmann, 2016). By using the obtained ratio of $n_{50}$ to the 532 nm backscatter coefficient, the entire profile of the 532 nm particle backscatter coefficient was converted into a $n_{CCN}$ profile as shown in Fig.13. The profile of the 532 nm particle extinction coefficient shown in Fig.13 was obtained by multiplying the backscatter coefficients with a lidar ratio of 30 sr as obtained from the backscatter and extinction Raman lidar observations. The dashed $n_{CCN}$ curves indicate an uncertainty of 30% in the $n_{50}$ and $n_{CCN}$ estimation. The uncertainty of 30% is much lower than indicated in Table 1. The uncertainty in the Table 1 holds in the case of the more simple conversion method (Mamouri and Ansmann, 2016).

Fig. 13 shows that the estimated $n_{CCN}$ values of 25-55 $cm^{-3}$ around 2500 m height were in the same range as the the numbers of the retrieved CDNC (10-40 $cm^{-3}$ in Fig. 11b) in the beginning of the cloud evolution. The actual updraft characteristics at cloud base determine the actual supersaturation levels. Strong updrafts may produce water supersaturation levels exceeding 0.5%. Then the CCNC values are 50-100% larger than our estimates in Fig.13 as discussed in Mamouri and Ansmann (2016). All in all, we can conclude that the lidar observations provide a clear link between low aerosol particle concentration and therefore low CDNC in the mixed-phase cloud top layer, The aerosol conditions controlled the microphysical properties in the liquid-water-dominated cloud top layer.

In order to better understand the entire role of aerosol particles in the evolution of mixed-phase clouds, we also estimated the ice crystal number concentration ICNC in the virga from combined lidar-radar observations (Bühl et al., 2019) and the

ice-nucleating particle concentration INPC for the cloud top temperature of $-28.5°C$ from the 532 nm extinction coefficient in Fig. 13 (Mamouri and Ansmann, 2016).

Without going into too much detail, the weakly enhanced PLDR values above 2 km height indicated a minor contribution of mineral dust particles (about 5%) to the overall Arctic haze backscatter coefficient. This is in good agreement with studies of Di Biagio et al. (2018) and Yang et al. (2021), who analyzed spaceborne and ground-based lidar observations in combination with backward trajectory analysis and concluded that mineral dust typically contributes to the long-transported continental aerosol mixtures in the lower and middle free troposphere in the Arctic. Dust is the most favorable, if not the only relevant INP type in the case of immersion freezing at temperatures $> -30°C$ during the winter half year when biogenic and biological components are absent (DeMott et al., 2010; DeMott et al., 2015; Kanji et al., 2017, 2020; Schill et al., 2020a). We used the INP parameterization scheme of DeMott et al. (2015) to estimate the dust INP concentration for immersion freezing. Here, the particle number concentration $n_{250}$ of dust particles with diameters $> 500$ nm is an input parameter and obtained from the respective lidar observation of the dust-related 532 nm backscatter coefficient (Mamouri and Ansmann, 2016). The retrieval finally yielded INPC estimates in the range from 0.1-0.5 $L^{-1}$ for the altocumulus top temperature of $-28.5°C$. Uncertainties in the INPC estimates are of the order of a factor of 3 (i.e., withing an order of magnitude). The main uncertainty source is the INP parameterization and not the retrieval of $n_{250}$ values.

To obtain also an estimate for ICNC, we used the lidar observations of the 532 nm extinction coefficient in the first strong ice virga zone (4:00-5:00 UTC) below the liquid-water cloud layer together with radar reflectivity (8 mm wavelength) and Doppler information on the ice crystal fall speed spectrum in the virga from the AMF-1 KAZR (Ka-band zenith-pointing radar) (ARM-MOSAiC, 2020). These observations were compared with comprehensive model simulations of the lidar and radar observations as a function of ice crystal number concentration, size distribution, and shape (Bühl et al., 2019). The match between simulations and observations provide the range of retrieved ICNC values. In the case of the mixed-phase cloud we obtain 0.1-1.0 ice crystals per liter. Again, the retrieval uncertainty allows us to provide the order of magnitude of ICNC only.

Fig. 14 summarizes our data analysis efforts and highlights the overall potential of our advanced aerosol/cloud lidar to contribute to cloud research. Since radiosondes were launched every six hours (Fig. 14a and c) over the entire MOSAiC year, excellent conditions for a detailed lidar-based aerosol and cloud monitoring (Fig. 14b), including a coherent monitoring of the relative humidity field (Fig. 14b), were given during the MOSAiC winter half year.

The temperature and relative humidity ($RH_w$) profiles measured with four radiosondes before, during, and after the cloud event indicate that the liquid-water-dominated cloud top layer was never thicker than a few 100 m. The cloud system developed in a warm and moist air mass and vanished when the moist air mass was replaced by very dry air (see 17 UTC radiosonde $RH_w$ profile in Fig. 14c). A strong drop in the relative humidity at all heights occurred around 15:00 UTC according to the lidar observations in Fig. 14d. The relative humidity (from lidar) is obtained from the Raman lidar observation of the specific humidity (water-vapor-to-dry-air mixing ratio) and radiosonde temperature profiles permitting the computation of the respective water-saturation-related specific humidity levels (Dai et al., 2018).

As mentioned above, favorable conditions for ice nucleation (via immersion freezing on dust particles) were given at cloud top temperatures of $-28.5°C$ in the beginning of the cloud life time and at around $-25°C$ later on. Strong ice crystal evap-

oration in the virga zone caused the strong moisturing of the air mass below the main altocumulus deck after 3:00 UTC and caused also reduced crystal evaporation later on so that ice crystals could partly reach the ground as precipitation.

All estimated values for CCNC, CDNC, INPC, and ICNC are included in the lidar panel in Fig. 14b. We can summarize our observations as follows: During the early altocumulus development, the estimated CCNC values outside of the cloud were in a similar range to the estimated CDNC levels within the cloud as it was also the case for the estimated INPC and ICNC levels. Thus, our data suggest that the estimated cloud active particle levels could be high enough to control the case-study cloud at given favorable moisture conditions and in the absence of other processes that might influence CDNC and ICNC levels (e.g., secondary ice formation, crystal-crystal collision and aggregation processes, or droplet collision and coagulation events). This hypothesis would be in line with numerous other Arctic studies that have previously observed this phenomenon (e.g. Mauritsen et al., 2011). However, higher resolution in-cloud microphysical data are required to verify this lidar-based hypothesis.

It is noteworthy to mention that the estimated INP concentrations of 0.1-0.5 $L^{-1}$ in Fig. 14b are also in line with previous Arctic in situ observations at similar temperatures. The measured INP values ranged from 0.001 to around 2.5 $L^{-1}$ for temperatures from $-25$ to about $-28°C$ (Mason et al., 2016; Creamean et al., 2019; Wex et al., 2019; Hartmann et al., 2020). As part of the MOSAiC data analysis, we plan to analyze many winter as well as summer altocumulus events and thus cloud formation under contrasting aerosol conditions to obtain an improved view on the role of long-transported aged aerosol pollution on the evolution of mixed-phase clouds in the high Arctic.

## 3.4 Cirrus evolution in wildfire smoke

According to a study of Barahona et al. (2017), Arctic ice clouds tend to form almost exclusively by heterogeneous ice nucleation with a contribution of only 10% by homogeneous freezing. The occurrence of a persistent smoke layer in the UTLS height range over the drifting Polarstern for more than seven months offers an excellent opportunity to investigate the potential impact of aged wildfire smoke particles on cirrus formation. More than 50 cirrus systems developed in the upper troposphere in a smoke-influenced environment during the winter half year. The importance of such a smoke impact study arises from the fact that organic aerosol (OA) is besides dust and marine particles ubiquitous in the atmosphere (Schill et al., 2020b). However, in contrast to dust and marine particles the INP potential of OA is not well understood. Scarce field data are the main reason for the lack of clarity and knowledge.

Because of the complex chemical, microphysical, and morphological properties of aged fire smoke particles, which can occur as glassy, semi-liquid, and liquid aerosol particles, the development of smoke INP parameterization schemes is a crucial task. Smoke particles from forest fires are largely composed of organic material (organic carbon, OC) and, to a minor part, of black carbon (BC). The BC mass fraction is typically <5% (Dahlkötter et al., 2014; Yu et al., 2019). Biomass burning aerosol also consists of humic like substances (HULIS) which represent large macromolecules. The particles and released vapors within biomass burning plumes undergo chemical and physical aging processes during long-range transport such as coagulation, condensation, and heterogeneous reactions, resulting in changes in its morphological characteristics (size, shape, and internal structure) and mixing state.

Our knowledge on the potential of smoke particles to serve as INP is mainly based on laboratory and field studies of the ice nucleation potential of fresh smoke (with focus on black carbon (BC) or soot particles) (Petters et al., 2009; Twohy et al., 2010; Prenni et al., 2012; McCluskey et al., 2014; Levin et al., 2016; Kanji et al., 2020; Schill et al., 2020a). However, aged smoke in the upper tropopshere (at cirrus level, several day, weeks, or months after emission) has fundamentally different properties to the ones of wildfire particles emitted just a few minutes to hours ago (Reid and Hobbs, 1998; Fiebig et al., 2003; Dahlkötter et al., 2014). As China et al. (2015) pointed out, freshly emitted soot or black carbon (BC) particles are typically hydrophobic, lacy fractal-like aggregates. During transport, lacy soot undergoes compaction upon humidification. Schill et al. (2020b) mention that, during the aging process, the particles can also accumulate secondary sulfate mass from condensation of gaseous sulfuric acid that is present in the background atmosphere.

In this section, we present a MOSAiC cirrus case study to illuminate, for the first time, the potential of aged wildfire smoke to influence ice formation in the upper troposphere. An overview of organic particles, their complex properties, and ice nucleating efficiacy is given in the review article of Knopf et al. (2018). It is assumed that aged smoke particles show an almost perfect spherical core-shell structure and that the ability of smoke particles to serve as INP mainly depends on the organic material (OM) in the shell of the coated smoke (or soot) particles. At low temperatures, e.g., in the UTLS region, where the atmospheric temperature can be as low as 180 K, it is conceivable to assume that the particles are in a glassy state. Aerosol particles serving as INPs usually provide an insoluble, solid surface that can facilitate the freezing of water (Knopf et al., 2018). Deposition ice nucleation is defined as ice formation occurring on the INP surface by water vapor deposition from the supersaturated gas phase. When the supercooled smoke particle takes up water or its shell deliquesces, immersion freezing can proceed, where the INP immersed in an aqueous solution can initiate freezing (Knopf et al., 2018; Berkemeier et al., 2014). If the smoke particle becomes completely liquid (and no insoluble part within the particle is left), homogeneous freezing will take place at temperatures below 235 K (Koop et al., 2000).

Our MOSAiC observation was performed on 6 December 2019. As shown in Fig. 3c and d, a long lasting cirrus event was monitored. The full cirrus lifetime was about 36 hours. Large virga of falling ice cyrstals permanently removed ice crystales from the cloud top region where they were nucleated. The cirrus system developed in the lower part of the UTLS smoke layer. The bow-like feature of the lower boundary of the extended virga zone was caused by a high pressure ridge (with dome-like temporal evolution of the influenced height range). The ridge crossed RV Polarstern during this day and was characterized by a warmer and very dry air mass in the lowest few kilometers of the troposphere. Ice crystals falling into this dry air mass immediately evaporated.

Figure 15 shows the mean cirrus layer structures measured from 6:00-12:00 UTC and in the early afternoon (12:30-13:30 UTC) together with the meteorological conditions in terms of air temperature, relative humidity, and ice supersaturation measured with three MOSAiC radiosondes at about 5:30, 12:30, and 17:30 UTC (30 minutes after launch). Very constant meteorological conditions were found from about 4 km up to cirrus top at 8-9.5 km height. Temperatures were below $-40°C$ at heights $>5.5$ km so that development of high altitude liquid-water layers was impossible. Clear ice supersaturation conditions ($S_i > 1.1$) were given in the upper part of the cirrus system.

It is noteworthy to mention that equilibrium at ice supersaturation conditions as observed in the extended virga zone over the whole day (Fig. 15c) is a sign for a low crystal number concentration ($<35$ L$^{-1}$) (Murray et al., 2010). Such a low amount of ice crystals is not able to quench the supersaturation which is in turn indicative for heterogeneous ice nucleation. Homogeneous freezing would produce crystal concentrations of $>500$ L$^{-1}$ so that equilibrium at ice saturation level would occur within a
short time period. Figure 15 is further discussed below.

The HYSPLIT backward trajectories in Fig. 16 indicate that the humid air mass, in which the cirrus started to form, originated from the remote northern Pacific. The air mass ascended to upper tropospheric heights 8-10 days before reaching the Polarstern at $86°$N and $122°$E. During the last five days of the travel the humid and probably unpolluted Pacific airmass had the chance to mix with wildfire smoke (entrained from above). We hypothesize that these smoke particles then served as ice nuclei in the
cirrus formation process. Remaining marine particles may have served as INPs as well, but are usually regarded as inefficient INPs (McCluskey et al., 2018).

The data analysis with respect to ice nucleation on smoke particles is explained in Fig. 17 and 18. We applied the water-activity-based immersion freezing model (ABIFM) that allows predicting of immersion freezing under cirrus conditions (Knopf and Alpert, 2013). The following equation is used to compute the number concentration of smoke INP for the immersion
freezing mode (Knopf and Alpert, 2013):

$$n_{\mathrm{INP,I}} = s J_{\mathrm{het,I}} \Delta t \tag{1}$$

with the surface area concentration $s$ of the smoke particles in cm$^2$ m$^{-3}$, the ice crystal nucleation rate coefficient $J_{\mathrm{het,I}}$ (in cm$^{-2}$ s$^{-1}$), and the time period $\Delta t$ (in seconds) for which constant or almost constant ice supersaturation conditions are assumed. This can be the time period of a short updraft event (of a few minutes, 120-300 s) or the duration of the lifting period
of a gravity wave of typically $600\pm200$ s (Kalesse and Kollias, 2013). The ice nucleation rate coefficient $J_{\mathrm{het,I}}$ is a function of the organic material in the (liquid) shell of the smoke particle and the water-activity criterion $\Delta a_{\mathrm{w}}$ (Koop et al., 2000). $\Delta a_{\mathrm{w}}$, shown in Fig. 15c, describes the difference between the ice melting conditions (melting temperature) and the observed freezing conditions in terms of temperature and humidity. In the computation, $\Delta a_{\mathrm{w}}$ is obtained from the difference between the actually occurring RH$_{\mathrm{w}}$ (over water) and the ice-saturation-related RH$_{\mathrm{w,sat}}$ value for the observed (actually occurring) temperature
(with RH$_{\mathrm{w}}$ and RH$_{\mathrm{w,sat}}$ in decimal numbers). $\Delta a_{\mathrm{w}}$ must reach values of 0.2-0.27 or even 0.29 to initiate significant ice nucleation (see vertical lines in Fig. 15c). Homogeneous ice nucleation is characterized by $\Delta a_{\mathrm{w}} \approx 0.313$ (Knopf and Rigg, 2011). As can be seen, $\Delta a_{\mathrm{w}}$ takes values of 0.1-0.2 according to the radiosonde observations. $\Delta a_{\mathrm{w}}$ values of 0.22-0.29 are only reached here during vertical motion of air parcels.

For demonstration of our method, we chose to apply the ABIFM for leonardite (a standard humic acid surrogate material)
to represent the amorphous organic coating of smoke particles. An alternative natural organic substance would be Pahokee peat (Knopf and Alpert, 2013). Leonardite, an oxidation product of lignite, is a humic-acid-containing soft waxy particle (mineraloid), black or brown in color, and soluble in alkaline solutions. Both substances (leonardite or Pahokee peat) served as surrogates for humic-like substances (HULIS) in extended immersion freezing laboratory studies

Similarily, we calculate the INP number concentration for the deposition ice nucleation (DIN) mode (Wang and Knopf, 2011):

$$n_{\mathrm{INP,D}} = s J_{\mathrm{het,D}} \Delta t. \tag{2}$$

The ice nucleation rate coefficient $J_{\mathrm{het,D}}$ requires information on temperature $T$ and ice supersaturation $S_{\mathrm{i}}$, also shown in Fig. 15c, and parameters describing the ice nucleation potential of the organic material. Also in the case of DIN, laboratory results for leonardite were used (Wang and Knopf, 2011). A mored detailed description of smoke INP computation may be found in Ansmann et al. (2020).

Figure 17 provides an overview of all retrieval products necessary to evaluate the potential of smoke particles to serve as INP. From the aerosol lidar observations (backscattering at cloud free condition) we obtain estimates for the particle surface area concentration $s$ (Ansmann et al., 2020). We show the aerosol conditions at different days in Figure 17b (before, during, and after the cirrus event) and used the respective lidar backscatter profiles to estimate the surface area concentration of the smoke particles. It can be concluded that the smoke surface area concentration was in the range from 0.05-0.15 cm$^2$ m$^{-3}$ during the formation of ice crystals at 7-7.5 km and at 9.0-9.5 km in the case of the cirrus layer in Figure 17a. In Fig. 17b, also the estimated number concentration of large smoke particles $n_{250}$ (with radius $\geq$250 nm, lower axis) is shown. $n_{250}$ may be interpreted as the overall reservoir of potential smoke INPs. It is assumed that larger particles provide better ice nucleation conditions than smaller particles. The uncertainty in the aerosol estimates is about 20-25% (Table 1).

Fig. 17c shows the results of the INPC calculations with Eqs. (1) and (2). The range of immersion-freezing INPC (i.e., the length of the respective two horizontal bars in Fig. 17c) is obtained by computing $n_{\mathrm{INP,I}}$ with Eq. (1) for $\Delta a_{\mathrm{w}}$ of 0.225 and 0.25. The particle surface area concentration of 0.05 cm$^2$ m$^{-3}$ is used in these computations and $\Delta t$ is set to 600 s. In the case of the deposition nucleation INPC, the shown INPC range (length of each bar) is obtained by computing $n_{\mathrm{INP,D}}$ with Eq. (2) for the corresponding ice supersaturation values $S_{\mathrm{i}}$ from 1.37–1.41 (7-7.5 km height, $-55.4°$C) and from 1.41–1.44 (9-9.5 km height, $-64°$C). As can be seen, $n_{\mathrm{INP,I}}$ ranged from 1–63 L$^{-1}$ at both heights (for the same $\Delta a_{\mathrm{w}}$ range), and for $n_{\mathrm{INP,D}}$, we obtained values from 0.4-3 L$^{-1}$ at 7-7.5 km height and $<$0.01 L$^{-1}$ at 9-9.5 km height. The uncertainty in these estimates is unspecified. This is the first attempt to estimate aged smoke INP under realistic atmospheric conditions with parameterizations developed under laboratory conditions.

In the final step, we compare the immersion-freezing INPC values in Fig. 17c, which can also be interpreted as predicted ice cyrstal numbers, with estimated ice crystal numbers derived from combined lidar-radar observations. The lidar-radar ICNC retrieval was already explained in Sect. 3.3. In this MOSAiC cirrus closure study, we used the strong and most accurate observations of radar reflectivity in the lower part of the virga zones from 4-7 km height. Since Eqs. (1) and (2) primarily deliver ice crystal nucleation rates (for $\Delta t = 1$ s), we compare nucleation rate values in Fig. 18b. As pointed out by Bühl et al. (2019), the ice crystal downward flux (in m$^{-2}$ s$^{-1}$) is the most direct result of lidar-radar retrievals because besides radar reflectivity also the falling velocity is measured with the Doppler radar. And this crystal flux rate can be interpreted as ice nucleation rate as long as crystal-crystal collision and aggregation processes do not change the ice crystal number concentration too much, on the way from the cloud top region (ice nucleation region) to the lower part of the virga zones. However, this assumption may be

strongly violated as Mitchell et al. (2018) concluded from CALIPSO lidar observations. They found that the ice crystal number concentration in the virga may be lower by a factor of 3-5 compared to the ICNC values at cloud top. On the other hand, the crystal nucleation rates we discuss below were quite low so that the aggregation effect may have been small (of the order of a factor of 2 or less for this Arctic ice cloud system). Other ICNC influencing effects, e.g., secondary ice formation can be

ignored in the case of cold cirrus clouds (Field et al., 2006; Korolev and Leisner, 2020). The low lidar-radar-estimated ICNC values of 40-70 $m^{-3}$ or 0.04–0.07 $L^{-1}$ (mean values for the virga between 4-6 km height for the period from 6-12 UTC, not shown here) and about 150 $m^{-3}$ or 0.15 $L^{-1}$ higher up (based on less accurate radar reflectivity measured at 6-7 km height) corroborate that the aggregation-related uncertainty in our lidar-radar estimate of the ice crystal nucleation rate at cloud top may have been low. For the time period from 12-14 UTC, the lidar-radar retrieval yielded ICNC values of 80-500 $m^{-3}$ or

0.08–0.5 $L^{-1}$ in the virga layer from 4-6 km height and about 800 $m^{-3}$ at 7 km height.

In our cirrus closure experiment, the goal is now to check how well the two numbers of the predicted and the estimated ice crystal nucleation rates are in agreement. The question behind is: Can the smoke INP parameterizations reproduce these values given by the lidar-radar observations so that we can conclude that smoke particles most likely controlled the evolution of the cirrus deck and tne virga and respective ice crystal microphysical properties. INP parameterization is appropriate even to be

used in models to desribe the smoke impact on cirrus formation.

As shown in Fig. 18b, the lidar-radar retrieval yielded ice crystal nucleation rates of 10-200 $m^{-3}\,s^{-1}$ and 50-1000 $m^{-3}\,s^{-1}$ (derived from the crystal flux observations between 4 and 6 km height) for the cirrus period from 6–12 UTC and 12–14 UTC, respectively. The prediction (based on INP retrieval with Eq. (1) for $\Delta t = 1$ s) reveals 1-100 $m^{-3}\,s^{-1}$ at cirrus top layer level. This can be regarded as a reasonable agreement keeping in mind that the particle surface area concentrations may have been

underestimated, strong updrafts may have occurred causing significantly higher $\Delta a_{\mathrm{w}}$ values than 0.25 (at a value of $\Delta a_{\mathrm{w}} = 0.27$ we obtain an order of magnitude higher nucleation rates), or leonardite is not representing well the organic material in the liquid shell of the smoke particles. With Pahokee peat parameters we obtain a factor of three higher ice crystal nucleation rates (or INPC values).

To sum up, heterogeneous ice nucleation is a complex process, especially in the case of organic aerosol particles. However,

the successful closure, indicated by a reasonable match between the predicted ice crystal nucleation rate (from the $n_{\mathrm{INP,I}}$ computation) and the estimated ice crystal nucleation rate (from lidar-radar observations) shows that the wildfire smoke was probably able to trigger cirrus formation (before the ice supersaturation onset for homogeneous freezing was reached) and to control the further evolution of the ice cloud system. The facts that (a) the radiosondes showed permanent ice supersaturation conditions in the virga over the day (a sign for low ICNC values in the virga) and (b) that the lidar-radar retrievals pointed to

rather low ice crystal nucleation rates are independent indications that the heterogeneous ice formation was most probably responsible for the observed cirrus evolution, at least controlled the microphyscial properties on 6 December 2019. It is clear that many more closure studies are needed to obtain a statistically trustworthy view on smoke and the role in cirrus ice nucleation. The respective extended data analysis will be part of our MOSAiC data analysis.

## 4 Conclusion and outlook

The goal of this introductory article was to present some highlights and key findings of the MOSAiC winter half year and also to provide an overview of the capabilities of modern lidar methods to contribute to Arctic research in the field of aerosols, clouds, and aerosol-cloud-interaction. Continuous observations were performed mostly at latitudes $> 85°$N during the MOSAiC year from September 2019 to May 2020.

The highlight of our observations was the detection of the long lasting UTLS wildfire smoke layer which was present over the North Pole region for more than seven months. A detailed analysis of the smoke properties and the potential impact on the record-breaking ozone reduction can be found in Ohneiser et al. (2021). Besides the smoke, we presented two days with typical Arctic haze layering features and properties. The results agree well with foregoing studies performed in the framework of POLARCAT-IPY and ARCTAS.

Our research focus is on aerosol-cloud interaction, especially on ice nucleation in mixed-phase clouds and cirrus in the middle and upper troposphere. We developed new techniques, data analysis concepts, and closure experiments and applied them for the first time to Arctic cloud studies. These successful closure experiments (demonstrated in two case studies) corroborate that aerosol particles are able to control cloud evolution and cloud microphysical properties.

As a future work, we will analyze the one-year data set to characterize the annual cycle of aerosol and cloud conditions and will contrast winter with summer conditions especially regarding aerosol-cloud interaction at cirrus level. Aerosol conditions during winter and early spring are very different from the ones during the summer and early autumn season and thus the impact of aerosol particles on mixed-phase cloud and cirrus evolution should be different as well.

Such large campaigns as the MOSAiC expedition offer the unique opportunity to introduce and apply new techniques and concepts of data analysis methods and integrate them into the existing instrumentation infrastructures and routine monitoring and observational data processing. However, to better characterize the uncertainties in our estimated and retrieved aerosol and cloud products and to narrow the uncertainty ranges, field campaigns with frequent overflights of remote sensing stations by aircraft in situ measuring cloud-relevant aerosol properties as well as cloud microphyscial properties are required in the framework of validation efforts. Such comprehensive aircraft-remote-sensing field campaigns are also needed to improve our knowledge on the basic features of the complex processes of aerosol-cloud interactions.

Because of the importance of the Arctic in the climate system it would be desirable to establish at least one supersite in the high Arctic for remote sensing and in situ observations (with balloon, unmanned aerial vehicles, aircraft) of aerosol and cloud properties in the lower, middle, and upper free troposphere, and lower stratosphere. This station should perform year-around measurements to document the fast changes of the atmospheric and environmental conditions in the Arctic, to better understand the role of aerosols in cloud formation and ozone depletion processes, and also to validate established and new remote-sensing-based retrieval methods.

## 5 Data availability

Polly lidar observations (level 0 data, measured signals) are in the PollyNET data base (PollyNET, 2020) with quicklooks at http://picasso.tropos.de. All the analysis products are available at TROPOS upon request (polly@tropos.de). The AMF-1 cloud radar data are provided by the ARM MOSAiC user facility (ARM-MOSAiC, 2020). In addition, all MOSAiC consortium members have early access to the data via the MOSAiC Central Storage (MCS) system before the data become publicly released on 1 January 2023. The radiosonde data are available at https://doi.org/10.1594/PANGAEA.928656 (Maturilli et al., 2021). PANGAEA is the primary long-term archive for the MOSAiC data set. KAZR data are available via the ARM data achieve (https://adc.arm.gov/discovery/).

## 6 Author contributions

The paper was written by RE and AA with contributions (data analysis) from KO, HB, CJ, JB, HGe, MH, SD, MM, IV, and PS. The co-authors HGr, MR, JH, and DA as well as RE took care of the lidar observations aboard Polarstern during the MOSAiC year. CJ and RW implemented and tested the new dual-FOV polarization lidar channels. UW and MA were involved in the data interpretation and discussions of the observations.

## 7 Competing interests

The authors declare that they have no conflict of interest.

## 8 Financial support

The data was produced as part of the international Multidisciplinary drifting Observatory for the Study of the Arctic Climate (MOSAiC) with the tag MOSAiC20192020 and Project ID AWI_PS122_00. This project has also received funding from the European Union's Horizon 2020 research and innovation program ACTRIS-2 Integrating Activities (H2020-INFRAIA-2014 - 2015, grant agreement no. 654109). We gratefully acknowledge the funding by the Deutsche Forschungsgemeinschaft (DFG,German Research Foundation) – project no. 268020496 - TRR 172, within the Transregional Collaborative Research Center "ArctiC Amplification: Climate Relevant Atmospheric andSurfaCe Processes, and Feedback Mechanisms (AC)3". The development of the lidar inversion algorithm used to analyze Polly data was supported by the Russian Science Foundation (project no. 16-17-10241).

The publication of this article was funded by the Open Access Fund of the Leibniz Association.

*Acknowledgements.* We are grateful to the MOSAiC teams and the RV Polarstern crew for their perfect logistical support. Cloud radar data was provided by the US Department of Energy's Atmospheric Radiation Measurement Program. We further thank the entire radiosonde

team, especially AWI, DWD, ARM, Jürgen (Egon) Graeser, and all volunteers for their enormous efforts of producing the exemplary and uninterrupted six-hourly dataset in the full MOSAiC year.

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

**Table 1.** Overview of Polly observational products and characteristic (or typical) relative uncertainties in the determined and retrieved properties. Particle backscatter coefficients are measured at 355, 532, and 1064 nm, the other aerosol optical properties at 355 and 532 nm. $r$ denotes aerosol particle radius.

| Aerosol optical properties | Uncertainty |
|---|---|
| Backscatter coefficient $[\text{Mm}^{-1}\ \text{sr}^{-1}]$ | $\leq 10\%$ |
| Extinction coefficient $[\text{Mm}^{-1}]$ | $20\%$ |
| Lidar ratio $[\text{sr}]$ | $25\%$ |
| Depolarization ratio | $\leq 10\%$ |

| Aerosol microphysical properties | |
|---|---|
| Volume size distribution $[\mu\text{m}^3\ \text{m}^{-3}\ \mu\text{m}^{-1}]$ | $\leq 30\%$ |
| Volume concentration $[\mu\text{m}^3\ \text{m}^{-3}]$ | $\leq 25\%$ |
| Mass concentration $[\mu\text{g}\ \text{m}^{-3}]$ | $\leq 30\%$ |
| Surface-area concentration $[\text{cm}^2\ \text{m}^{-3}]$ | $\leq 25\%$ |
| Number concentration ($r > 50$ nm) $[\text{cm}^{-3}]$ | $50\%$ |
| Number concentration ($r > 250$ nm) $[\text{cm}^{-3}]$ | $\leq 25\%$ |
| CCN concentration $[\text{cm}^{-3}]$ | Factor 1.5-2 |
| INP concentration $[\text{L}^{-1}]$ | Factor 3 |

| Droplet properties (liquid water clouds) | |
|---|---|
| Droplet number concentration $[\text{cm}^{-3}]$ | $50\%$ |
| Droplet effective radius $[\mu\text{m}]$ | $20\%$ |
| Liquid water content $[\text{g}\ \text{m}^{-3}]$ | $25\%$ |
| Cloud extinction coef. (532 nm) $[\text{Mm}^{-1}]$ | $15\%$ |

| Water vapor fields | |
|---|---|
| Water-vapor-to-dry-air mixing ratio $[\text{g}\ \text{kg}^{-3}]$ | $\leq 10\%$ |
| Relative humidity $[\%]$ | $\leq 10\%$ |

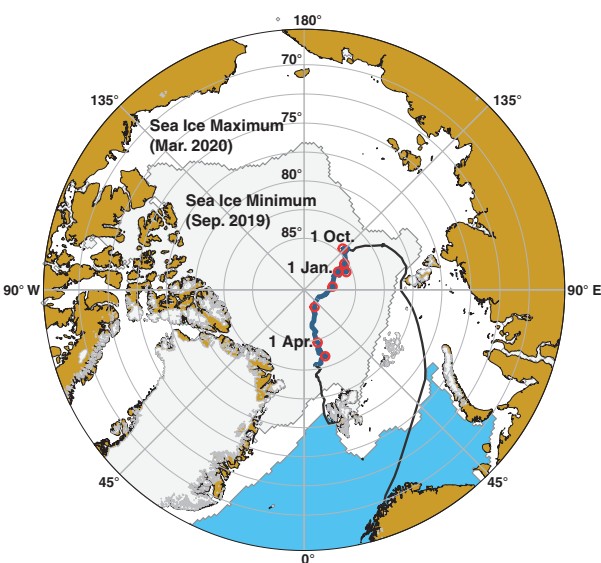

**Figure 1.** Travel (black) and drifting (blue) route of RV Polarstern from the beginning of October 2019 to mid of May 2020. Each of the eight red circles marks the beginning of the next month. The map was produced with 'ggOceanMaps' (Vihtakari, 2020) by using Sea Ice Index Version 3 data (Fetterer et al., 2020).

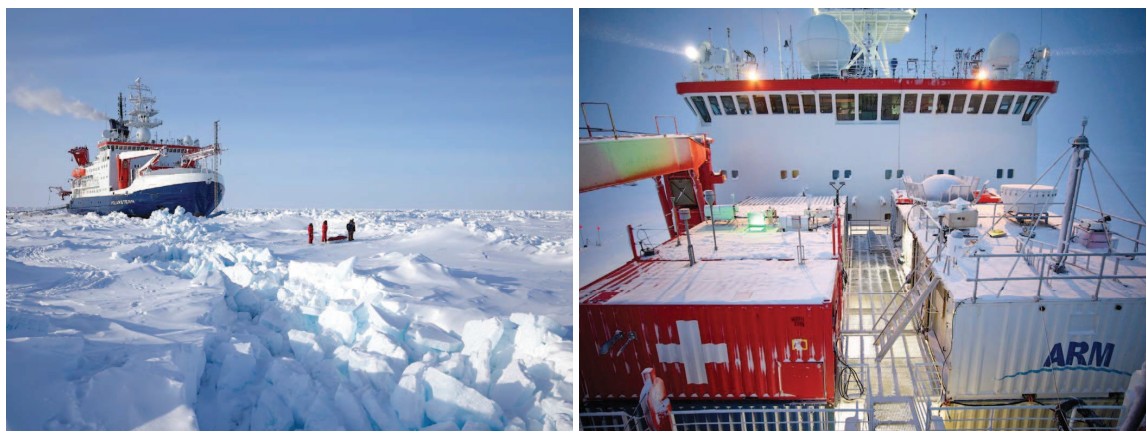

**Figure 2.** Polarstern drifting in the Arctic ice on 10 April 2020 (left panel) and measurement containers for in situ aerosol monitoring (the two first containers on the left side and the first container on the right side), and for remote sensing of aerosols and clouds (right panel). The OCEANET container of TROPOS is the third one to the back on the left side (with the bright spot on the roof caused by green laser light). The ARM mobile facility (AMT-1) is shown on the right side. The photographs are taken by Michael Gutsche (CC-BY 4.0), AWI.

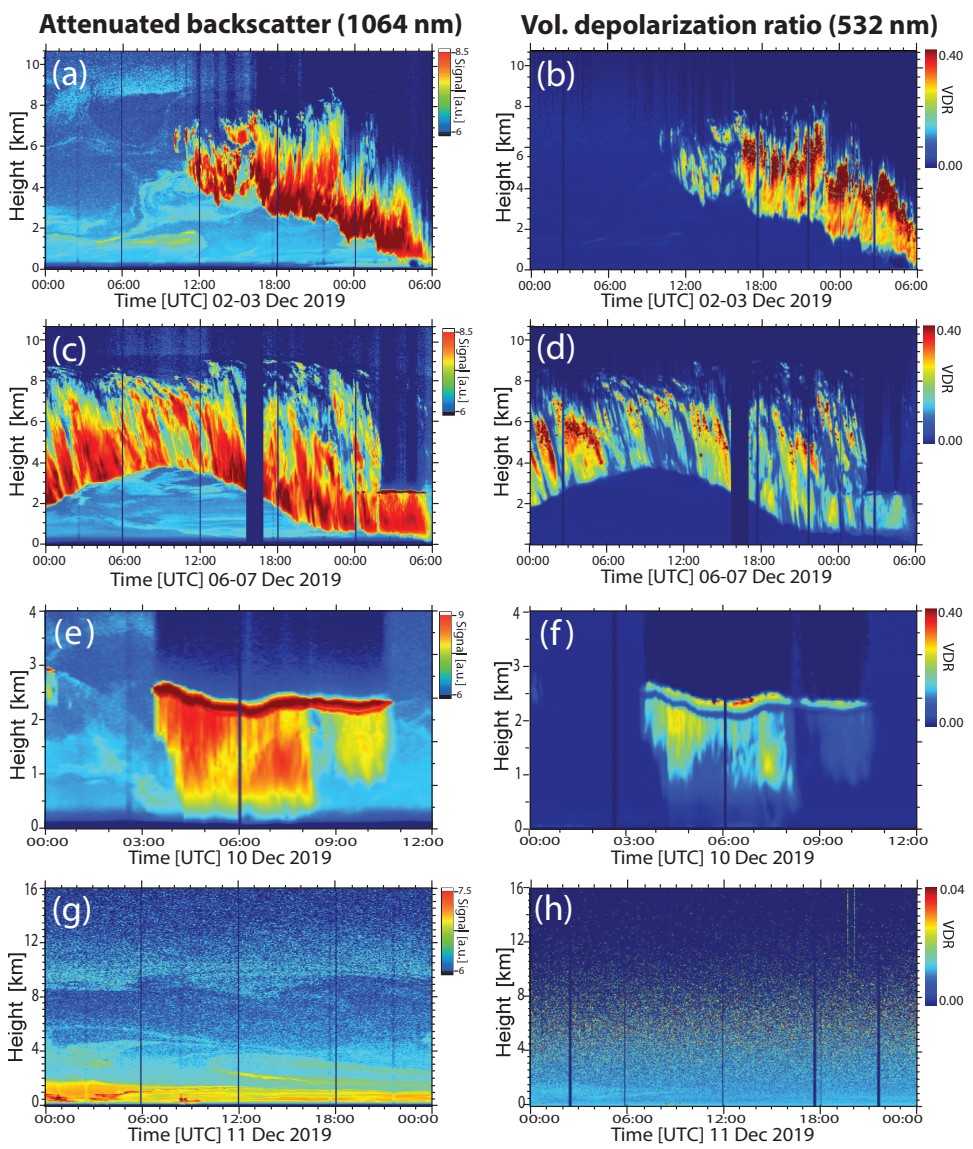

**Figure 3.** Ice clouds, mixed-phase clouds, and aerosols monitored with Polly aboard RV Polarstern from 2–11 December 2019. (a-d) Evolution of cirrus layers with strong virga embedded in wildfire smoke and Arctic haze, (e-f) development of a long-lasting mixed-phase altocumulus with shallow liquid-water layer at the top and ice virga below, and (g-h) Arctic haze (below 5 km height) and wildfire smoke (above 8 km) during clear sky conditions. The range-corrected 1064 nm signal (left panels, in arbitrary units, a. u., logarithmic scale, the given exponents indicate the signal range) and the 532 nm volume linear depolarization ratio (VDR, right panels) are shown. Note that the color scales vary from panel to panel.

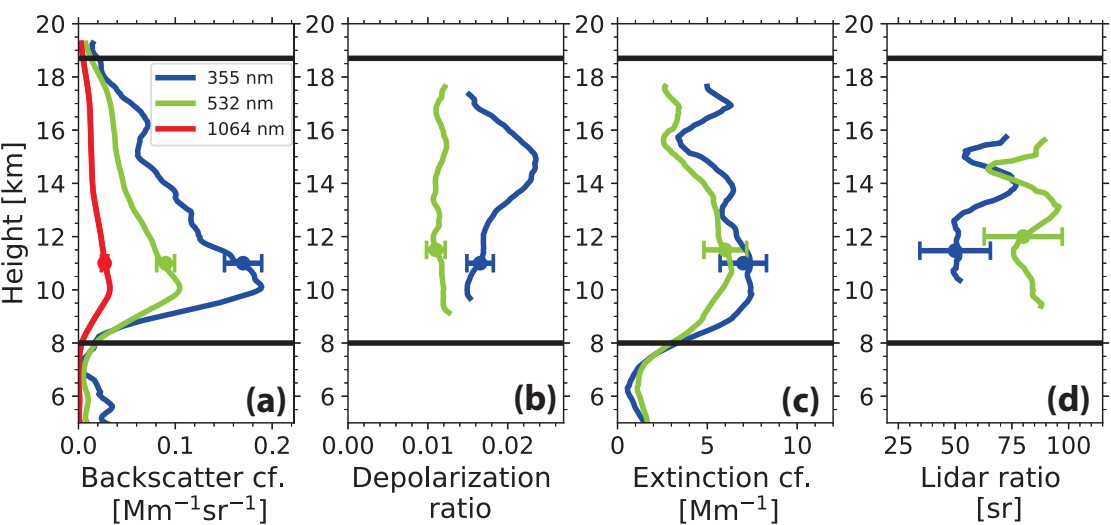

**Figure 4.** Profiles of optical properties (24 h mean values) of the wildfire smoke layer on 11 December 2019. Base and top heights of the smoke layer are indicated by black horizontal lines. (a) Particle backscatter coefficient at three wavelengths, (b) particle linear depolarization ratio at 355 and 532 nm, (c) smoke extinction coefficient at 355 and 532 nm, and d) respective smoke extinction-to-backscatter ratio (lidar ratio) are shown. All basic lidar signal profiles were strongly smoothed with vertical window lengths of 2000 m to strongly reduce the signal noise. Error bars indicate the estimated uncertainties (one standard deviation). More details are given in Ohneiser et al. (2021).

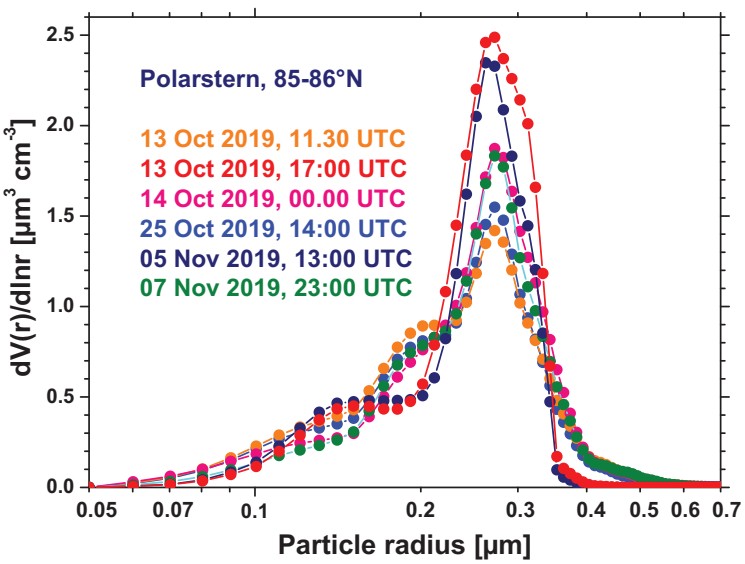

**Figure 5.** Size distributions of the stratospheric smoke particles retrieved from the multiwavelength lidar observations on five days in October and November 2019. A narrow accumulation mode with particle sizes (diameters) from 400 to 1000 nm and a weak Aitken mode to the left is typical for aged wildfire smoke particles.

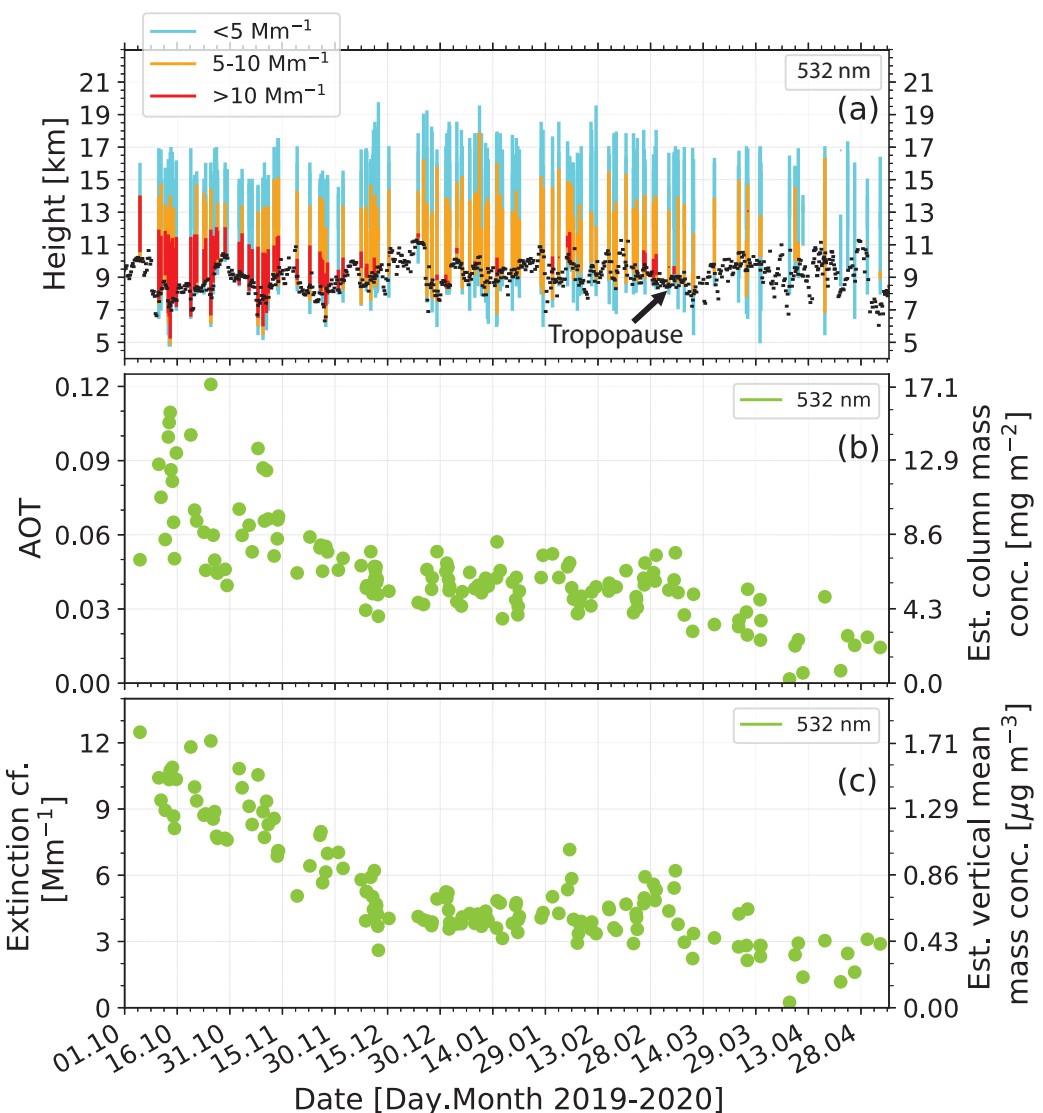

**Figure 6.** (a) Overview of Polly observations of the UTLS smoke layer (colored bars from base to top, one bar per day) for the winter half year (October 2019 to April 2020). Observational gaps between bars are caused by opaque low level clouds and fog. The colors in the bars provide information about the smoke particle extinction coefficient at 532 nm. The tropopause (black dots) separates the tropospheric from the stratospheric part of the smoke layer. (b) 532 nm AOT of the smoke layer and estimated column mass concentration. (c) Smoke layer mean 532 nm particle extinction coefficient and estimated vertical mean particle mass concentration. The AOT and layer mean extinction values are computed from the profile of the backscatter coefficient multiplied by a 532 nm lidar ratio of 85 sr. AOT uncertainty is about 10-20%. For comparison, background stratospheric extinction coefficients are expected to be of the order of 0.1–0.2 Mm$^{-1}$ as observed over northern midlatitudes (Baars et al., 2019). More details are given in Ohneiser et al. (2021).

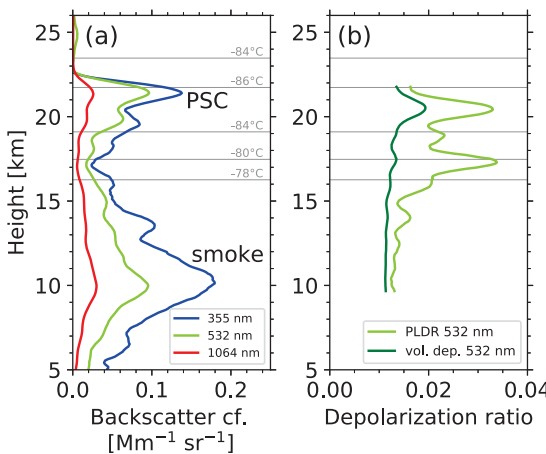

**Figure 7.** Polar stratospheric cloud (PSC) from 17.5–22.5 km height on top of the smoke layer on 14 January 2020, 21-24 UTC. Three-hour mean particle backscatter coefficients and volume and particle linear depolarization ratio (PLDR) are shown. The optical properties values are indicative for PSC Ib type. The 532 nm backscatter ratio (total-to-Rayleigh backscatter) peaks at 2.43 at 21.5 km height at about $-86°$C. PSC optical thickness was 0.0125 at 532 nm (computed from backscatter values multiplied by a lidar ratio of 50 sr) (Kim et al., 2018). Horizontal gray lines show different temperature levels.

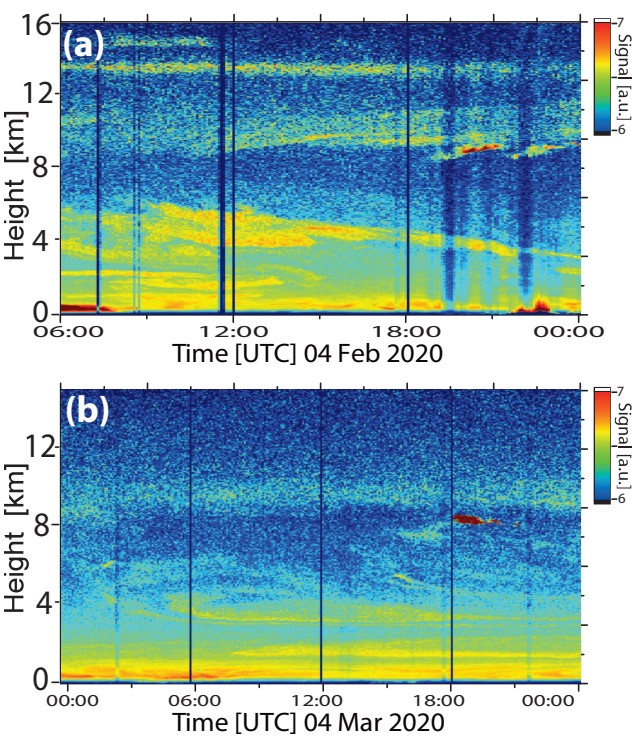

**Figure 8.** Arctic haze (below 8 km height) and wildfire smoke (above 8 km height) over the North Pole region in late winter on (a) 4 February 2020 and (b) 4 March 2020. PSC layers are present as well at 13.5 and 15 km height on 4 February (pronounced yellow layers). The range-corrected 1064 nm signal is shown in arbitray units (a.u., logarithmic scale).

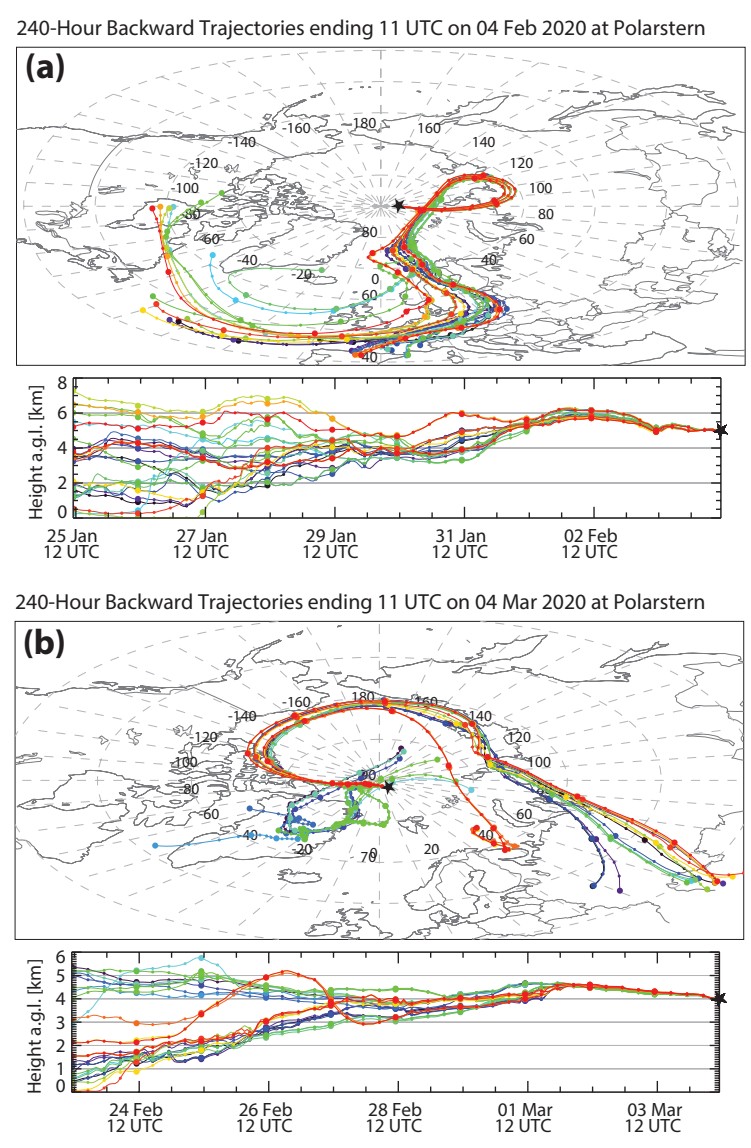

**Figure 9.** (a) HYSPLIT 10-day ensemble backward trajectories arriving at RV Polarstern on (a) 4 February 2020, 11 UTC arrival time, and on (b) 4 March 2020, 11 UTC arrival time (HYSPLIT, 2020; Stein et al., 2015; Rolph et al., 2017). According to Fig. 8, pronounced Arctic haze plumes were observed at the selected arrival heights of 5km (4 February) and 4 km (4 March).

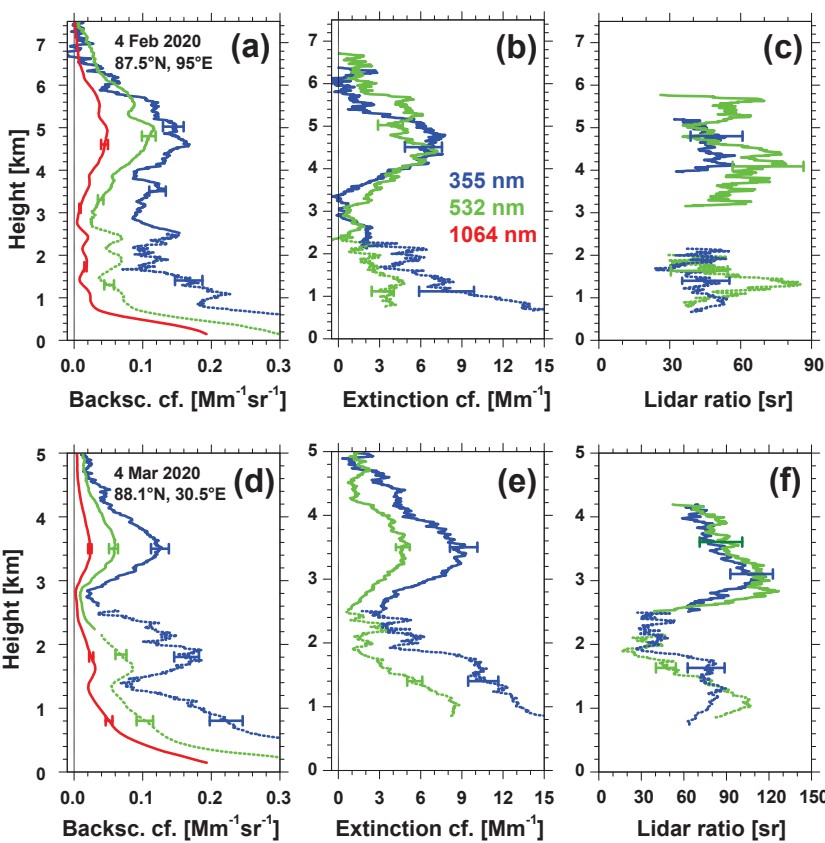

**Figure 10.** Arctic haze backscatter, extinction, and extinction-to-backscatter ratios (lidar ratios). Mean profiles for 4 February 2020, 6:00–17:37 UTC, and for 4 March 2020, 0:00–17:45 UTC, are shown. A composite of near-range (dotted lines up to 2-3 km height) and far-range lidar observations (solid lines) is presented. Lidar signals are smoothed with vertical window lengths of 300 m (backscatter) and 900 m (extinction, lidar ratio). Error bars indicate the uncertainty (one standard deviation) in the optical properties. 532 nm AOT was 0.024 on 4 February (for heights up to 7 km) and 0.022 on 4 March (for heights up to 5 km).

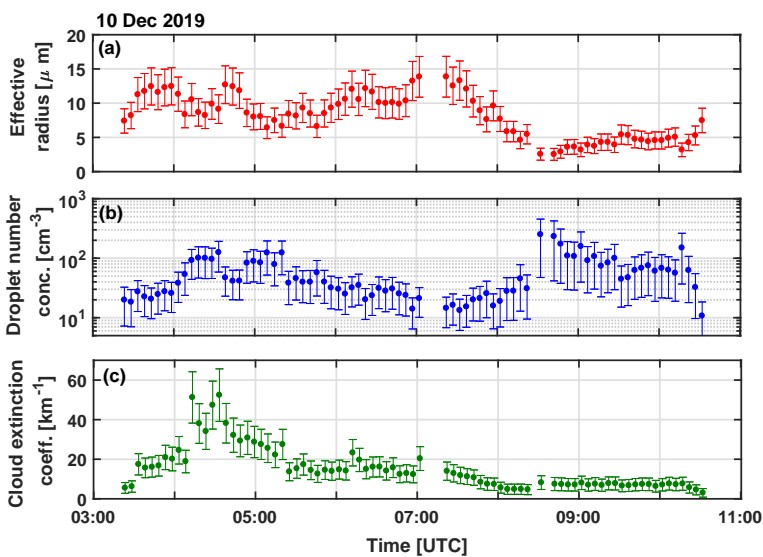

**Figure 11.** (a) Effective radius of cloud droplets, (b) cloud droplet number concentration, and (c) 532 nm cloud extinction coefficient (single-scattering) at 75 m above cloud base of the altocumulus layer in Fig. 3e and f. The effective radius can be interpreted as the characteristic droplet radius. Error bars indicate the uncertainty. The cloud properties were retrieved by means of the recently introduced dual-FOV polarization lidar technique.

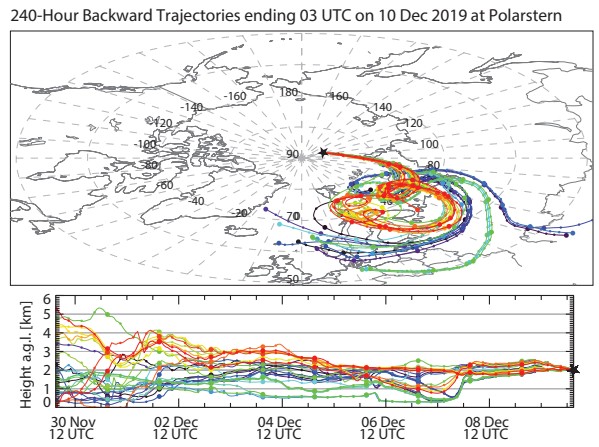

**Figure 12.** HYSPLIT 10-d ensemble backward trajectories arriving at 2 km height above RV Polarstern (black star) on 10 December 2019, 3:00 UTC. Thin and thick symbols indicate 6-hour and 24-hour time steps, respectively.

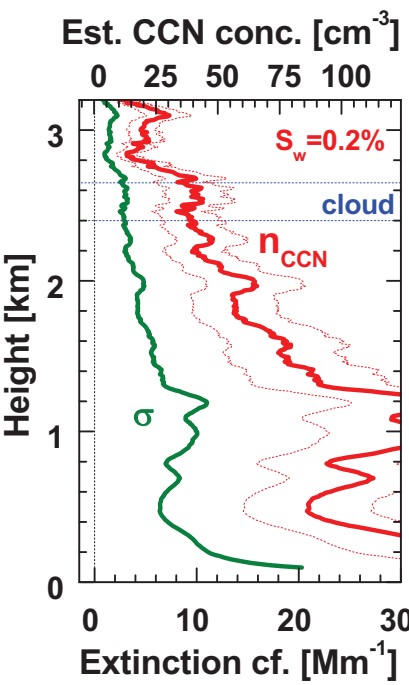

**Figure 13.** Aerosol observation on 10 Dec 2019, 2:15-2:45 UTC, just before the altocumulus layer was detected over the lidar station (see Fig. 3e). The profile of the 532 nm extinction coefficient $\sigma$ is estimated from the backscatter coefficient profile (assuming a lidar ratio of 30 sr obtained from the Raman lidar observations). The estimated CCN concentration ($n_{CCN}$ for a water supersaturation of 0.2%) is obtained from the multiwavelength lidar data analysis (inversion technique, see text for more details) The thin red dotted lines show the assumed uncertainty range of $\pm30\%$. Cloud base and top heights of the altocumulus layer at 3:30 UTC are indicated by horizontal lines.

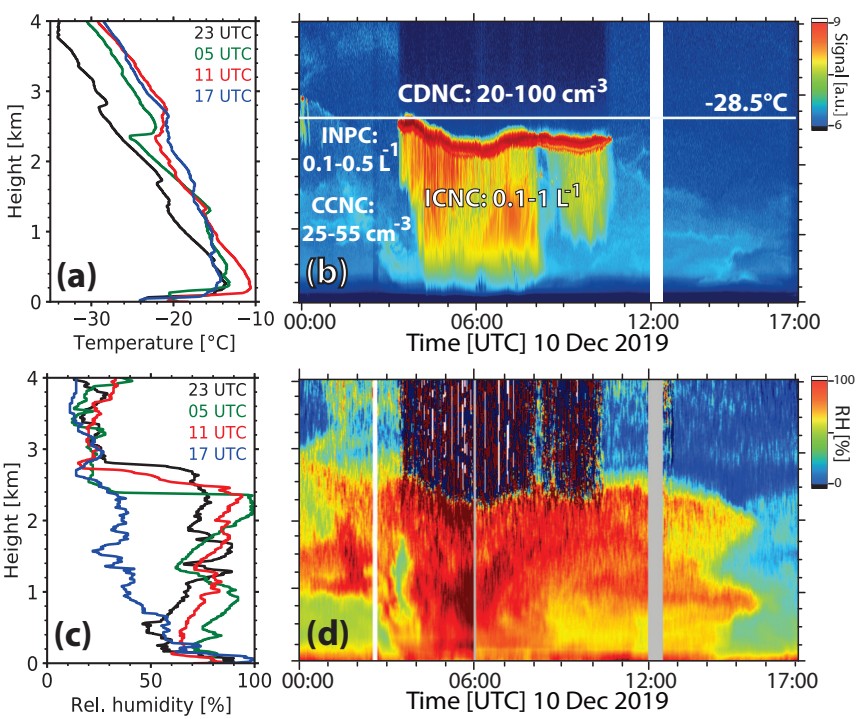

**Figure 14.** Mixed-phase-cloud closure study. (a, c) Profiles of temperature and relative humidity (over water) measured with four MOSAiC radiosondes launched at 23 UTC (9 December), 5, 11 and 17 UTC (10 December). (b) 1064 nm range-corrected signal showing the mixed-phase cloud layer between 2 and 2.6 km height with ice virga below the main cloud layer, and (d) height-time display of Raman lidar observations of relative humidity. In (c) cloud-droplet number concentration (CDNC) as obtained from the dual-FOV lidar observations (during the 3:15-5:00 UTC time period), CCN and INP concentrations (CCNC, INPC) estimated from the lidar observations at 2.5 km height (CCNC) and at 2.6 km height (ICNC, for the given cloud top temperature of $-28.5°C$) in clear sky (2:15-2:45 UTC, i.e., before the cloud layer appeared), and ice crystal number concentration (ICNC, 4:00-5:00 UTC mean value) as estimated from combined lidar-radar observations are given as numbers. In (c), the 1064 nm signal is biased in the near range ($<1000$ m), especially during the strong virga backscattering periods.

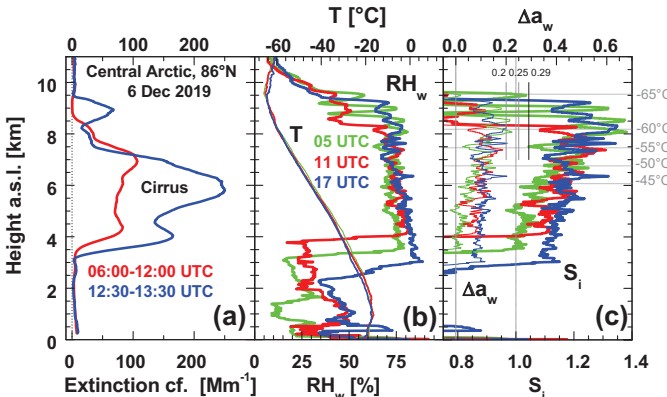

**Figure 15.** (a) Temporally averaged (mean) cirrus extinction coefficient $\sigma$ (532 nm) for two observational periods from 6-12 UTC (red) and from 12:30-13:30 UTC (blue, see Fig. 3c), (b) profiles of temperature T and relative humidity $RH_w$ (over water) observed with radiosondes launched at 5, 11, and 17 UTC on 6 December 2019, and (c) water activation criterium $\Delta a_w$ and ice supersaturation $S_i$ computed from the temperature and water vapour observations shown in (b). The water activation criterium $\Delta a_w$ is the difference between the observed $RH_w$ (in decimal numbers) and the ice-saturation-related $RH_w$ value (for the observed temperature). In (c), different temperature levels are indicated by thin horizontal lines and $\Delta a_w$ values of 0.2, 0.25, and 0.29, required to initiate ice nucleation, are shown as vertical line segments.

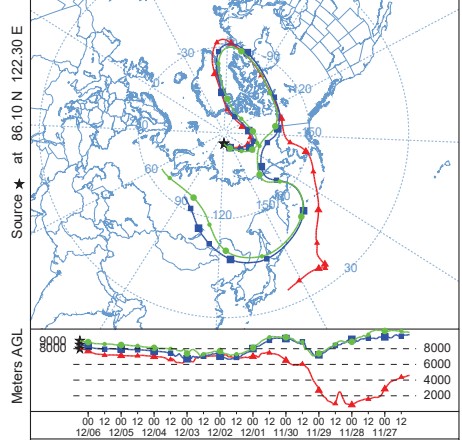

**Figure 16.** HYSPLIT 10-d backward trajectories arriving at at 8, 8.5, and 9 km height above RV Polarstern (black star) on 6 December 2019, 6:00 UTC. Thin and thick symbols indicate 6-hour and 24-hour time steps, respectively.

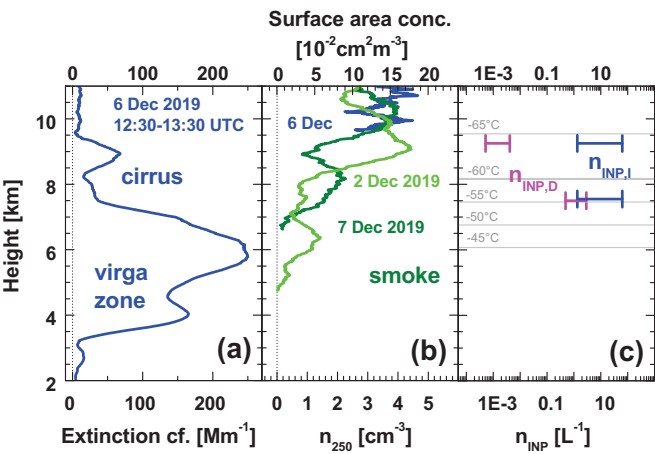

**Figure 17.** (a) Mean cirrus extinction coefficient $\sigma$ (532 nm) for the cirrus segment measured on 6 December 2019, 12:30-13:30 UTC. (b) Estimated smoke particle number concentration $n_{250}$ for large particles with diameters >500 nm and particle surface area concentration $s$ retrieved from the lidar observations for the cloud-free days of 2 and 7 December (green, olive) and for 6 December (in blue, above the cirrus), and (c) estimated ranges of INP number concentrations $n_{\mathrm{INP,I}}$ (for immersion freezing, blue) and $n_{\mathrm{INP,D}}$ (for deposition nucleation, in pink). $\Delta a_{\mathrm{w}}$ ranged from 0.225 to 0.25, correspondingly the ice supersaturation $S_{\mathrm{i}}$ from 1.37-1.41 (at 7.5 km height, $-55.4°$C) and from 1.4 to 1.44 (at 9.25 km height, $-64°$C), the particle surface area concentrations was set to $s = 0.05\ \mathrm{cm}^{-2}\ \mathrm{m}^{-3}$, and the updraft time period to $\Delta t$=600 s in the smoke INP computations with Eqs. (1) and (2).

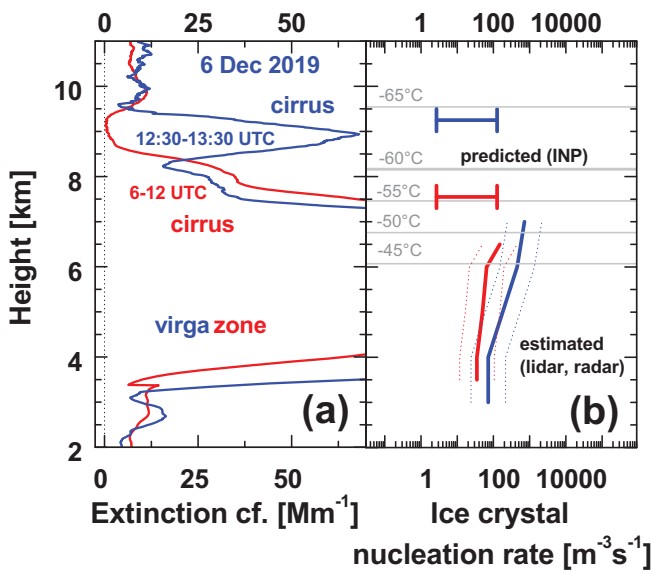

**Figure 18.** (a) Temporally averaged (mean) cirrus extinction coefficient $\sigma$ (532 nm) for two cirrus periods from 6-12 UTC (red) and from 12:30-13:30 UTC (blue) on 6 December 2019. (b) Predicted ice crystal nucleation rate, i.e., $n_{\text{INP,I}}$ for $\Delta t$=1 s in Eq. (1) (horizontal bars), and estimated ice crystal nucleation rates (vertical lines) obtained from the lidar-radar observations for the 6-12-UTC (red) and 12:30-13:30-UTC (blue) time periods. The thin dotted lines show the uncertainty (factor of 3) in the lidar-radar retrievals.