# Peer review of "Wildfire smoke, Arctic haze, and aerosol effects on mixed-phase and cirrus clouds over the North Pole region during MOSAiC: An introduction"

_Atmospheric Chemistry and Physics, 2020_

## Short Comment (SC1) · 24 Feb 2021

**Short comment on Engelmann et al. paper**

This is a very interesting paper with rare lidar ratio measurements in the high Arctic near the pole. The observation of a smoke layer getting trapped in the Arctic polar vortex is also very interesting. This short comment is primarily aimed at a statement made by the authors on the CALIPSO measurements over the high latitude regions during the period of their observation. In particular in page 9 they state that "It is noteworthy to mention that the CALIPSO data base (CALIPSO, 2020a, b) does not contain clear hints on the Arctic UTLS aerosol layer observed continuously over the RV Polarstern. According to HYSPLIT (Hybrid Single-Particle Lagrangian Integrated Trajectory-Model) backward trajectories (HYSPLIT, 2020), satellite remote sensing (Kloss et al., 2020), and ground-based Raman lidar observations of the Alfred Wegener Institute at Spitsbergen (Ohneiser et al., 2021), the aerosol layer covered large parts of the Arctic and thus should have been detectable along the CALIPSO flight track (south of 81.8°N)." Contrary to this assertion, we do find clear evidence of layers detected by CALIPSO over the high latitude regions. Figure 1 shows 2 CALIPSO transects through the Arctic region on December 11, 2019 around 3:00 UTC (top panels) and 8:00 UTC (bottom panels), the same day for which smoke plume observations were presented by the authors in their Figure 3g.

[Figure]

Figure 1. CALIPSO browse images of total attenuated backscatter and aerosol subtypes on December 11, 2019 near 3:00 UTC (top panels) and 8:00 UTC (bottom panels).

Note the clear detection of many layers around 10-12 km between 78ºN and 81ºN in both cases. The layers have been classified as sulfate/other by the CALIPSO stratospheric aerosol subtyping algorithm because of the low backscatter (Kim et al., 2018). Figure 2 shows another example of aerosol layers detected on November 14, 2019.

[Figure]

Figure 2. CALIPSO browse images of total attenuated backscatter and aerosol subtypes on November 14, 2019.

Once again a very extended and coherent plume can be observed between ~70ºN-80ºN as detected by the CALIPSO layer detection algorithm on November 14, 2019. As in Figure 1 the layers are mostly classified as sulfate/other, but note the 2 layers near 77ºN which are actually classified as smoke.

It is therefore rather surprising that the authors chose to make the statement about the lack of layer detection by CALIPSO at high latitude UTLS region during the period of their observation.

On another point, in page 8, lines 12-15, the authors try to explain the low depolarization ratio (~0.01) in the aged smoke in terms of the core-shell structure collapsing and the particles becoming more spherical. This seems to contradict the results of Ohneiser et al. (2020) who found high depolarization ratio (up to 0.2 at 532 nm) for aged pyroCb plumes transported from Australia over Argentina in January 2020. In fact, as shown by Christian et al. (2020), the depolarization ratio in the pyroCb plumes from the Canadian pyroCb in August 2017 continued to increase with time for several weeks.

Adding latitude/longitude information on Figures 3 and 4 may be useful.

**References**:

Kim, M. H., et al., CALIPSO version 4 automated aerosol classification and lidar ratio selection algorithm, Atmos. Meas. Tech., 11, 6107-6135, 2018, 2018.

Christian, K. et al., Differences in the evolution of pyrocumulonimbus and volcanic stratospheric plumes as observed by CATS and CALIOP space-based lidars, Atmosphere, 11, 1035, 2020.

---

## Referee Comment (RC1) · Anonymous Referee #1 · 11 Mar 2021

**Overview:**

This study has some really interesting data that are definitely worth sharing with the scientific community in some form. It is great to see these high latitude data from locations where there were hardly any data at all before. And as the authors pointed out, these data offer very valuable information to contrast and compare with CALIPSO, which up until now has been one of the only sources of lidar information at remote Arctic sea ice locations (but which misses the highest latitude areas and has problems with lidar ratios).

However, the methodologies and associated uncertainties in this paper were not well described, and many of the conclusions were not well supported by the presented data. In particular, a lot of new and interesting techniques are used that are not well validated, but the resulting data presented as if they are known to be accurate. For this reason, I honestly do not know whether I should recommend this paper to be published or not, and I would like to re-evaluate it after the authors have been given a chance to better characterize the uncertainties and reframe the discussion in context of these uncertainties. See more specific comments below.

**Specific comments:**

- The Punta Arenas and Cyprus data the authors cite for CCN and INP validation of this work (Jimenez et al., 2020a; Ansmann et al., 2020, 2019; Mamouri and Ansmann, 2016) were not actually validated with *in situ* data. I think it is really important to be upfront about this fact, which suggests an unknown degree of uncertainty in the CCN and especially INP estimates for these Arctic data. It should also be mentioned that the Arctic environment is colder and cleaner than these other locations, and has different types of aerosol particles, which might affect the estimates and render previous validations efforts less useful here.
- Many high altitude Arctic AOTs will be very small. How can we be sure that determinations about particle properties can be made at such small signals?
- Can the authors take advantage of the available complementary MOSAiC data (e.g., with INP and CCN near the surface or from tethered balloon) to somehow better validate the data?
- I have made various comments below asking the authors to better describe the uncertainties in various parameters. But I think it is very likely in many cases that uncertainties may not be easy to describe because cloud and aerosol parameters estimated from lidar depend not only on things like conversion coefficients and assumptions about mineral dust, but also on variables like optical thickness of the cloud, and the extent to which the lidar signal has been attenuated. For example, note how the signal has been attenuated beyond the top of the cloud in Fig. 3. Therefore, there is a fundamental challenge when trying to use the methods in this paper to compare quantities like estimated CDNC between clouds, or even between the base and top of clouds for a case study. I am not sure how the authors can address these issues. Possibly a sensitivity study might help.

- Places where uncertainties need to be better described:

  - P6L9: "*The retrieval of aerosol microphysical properties such as particle volume, mass, and surface area concentration and estimates of cloud-relevant properties (aerosol-type-dependent cloud condensation nuclei, CCN, and ice-nucleating particles, INPs) is performed by means of the POLIPHON (Polarization Lidar Photometer Networking) approach (Mamouri and Ansmann, 2016, 2017; Ansmann et al., 2019, 2020)*" Please describe the validation for and uncertainties in this measurement in greater detail. For example, in Jimenez et al. (2020b) it is mentioned that uncertainties in lidar-derived CCN are around 50%, but that is not mentioned here. Please quantify the uncertainties, discuss how were derived and where they cannot be quantified, and how this information affects the interpretation of these results.
  - P6L13: "*Alternatively to the POLIPHON method, we used the multiwavelength lidar inversion technique (Müller et al., 1999, 2014; Veselovskii et al., 2002, 2012) to derive microphysical properties of aerosols including the particle size distribution for detected pronounced aerosol layers.*" Please describe the validation for and uncertainties in this measurement in greater detail.
  - Fig. 5: How well validated are these data? Can error bars in these measurements be applied to this figure?
  - Fig. 6. These are extremely low AOTs. It would be helpful to indicate instrument detection limits on both figures, and to show uncertainty bars, as I would expect these to get increasingly large at low AOTs. The discussion of uncertainties in these data, and how they relate to the conclusions of the study should be further expanded upon in the text.
  - Fig 10: what are the detection limits?
  - Fig. 11: Please discuss whether these are averages over the cloud layer, and if so how that cloud layer was determined. Please change to "estimated effective radius" and "estimated droplet number" in the figure and caption. Please describe in the methods text how the uncertainty range was determined, and discuss the extent to which this uncertainty is meaningful.

  - P13L27: "In this way we estimated CCN concentrations of about 30-70 cm$^{-3}$ with an uncertainty of a factor of 2." Please describe how this uncertainty factor was estimated, and why this uncertainty estimate is meaningful.
  - P14L2: "*Here, the particle number concentration n250 of dust particles with diameters > 500 nm is an input parameter and obtained from the respective lidar observation of the extinction coefficient in Fig. 12 and by assuming a dust fraction of 3-10% in the conversion of the extinction profile into the n250 profil (Mamouri and Ansmann, 2016).*" Please provide the uncertainties in the input parameter of dust particle concentrations with diameters > 500 nm and discuss the impact of these uncertainties on the INP estimates? Why assume a dust fraction of 3-10%? As it reads now, there seems to be very large uncertainties,

with estimated INP levels based on unsubstantiated assumptions. Hopefully further discussion can clarify this.

- P16L30: "*In this figure, the number concentration of large smoke particles n250 (with radii > 250 nm, lower axis) is shown as well. This number indicates the overall reservoir of favorable INPs (Ansmann et al., 2020).*" Please change to "*estimated number concentration of large smoke particles n250.*" Please explain why this estimate should give us the overall reservoir of favorable INPs, and discuss associated uncertainties.
- Section 3.4: I find the last 2 paragraphs of section 3.4 to be not useful and mainly conjecture because there are so many assumptions. I suggest removing these paragraphs and Fig. 15 entirely.

- Other places further information is required:

  - For the smoke section, could the authors please clarify why it is definitely smoke, and not a mix of pollution and smoke?
  - P7L26: "*This fourth mechanism is responsible for the occurrence of ULTS wildfire smoke over the North Pole region in the MOSAiC winter half year.*" Please provide evidence or the reference for the deduction that this mechanism is the predominant responsible pathway for this transport during this entire time.
  - P7L30: "*Figure 4 presents the optical properties of the smoke layer as measured on 11 December 2019 (Fig. 3g and h). The smoke layer extended from 8 to more than 18 km height.*" I don't see evidence in Fig. 3 of the smoke layer going past ~13 km. In Fig. 4, the instrument detection limits have not been shown. It would be helpful to add those for the reader to better interpret these plots, and to see how high a detectable layer extended.
  - P8L1: "*No other aerosol type (or cloud type) produces an inverse spectral behavior in terms of the particle lidar ratio*" Please describe the lidar ratio of aged pollution plumes, and say why that could not be a main contributor to the haze event determined here to be mostly made up of smoke particles.
  - P9L16: "the smoke layer was continuously present and probably homogeneously distributed over large areas of the Arctic." If CALIPSO could not observe the layer, what reason is there to believe that the layer was homogeneously distributed over large areas of the Arctic?
  - P9L21: "*Note that we corrected our stratospheric smoke observations in Fig. 6 for PSC effects.*" The authors should describe how they corrected for these observations.
  - P13L10: "*The new method was originally designed for pure liquid-water cloud observations but can be applied to mixed-phase clouds as long as backscattering by ice crystals is negligible compared to droplet backscattering in the droplet-dominated cloud top layer. This condition holds here with ice crystal backscatter coefficients of 5-10 $Mm^{-1}\ sr^{-1}$ in the virga and thus also in the cloud top layer and droplet backscatter coefficients of the order of 700 $Mm^{-1}\ sr^{-1}$.*" Please

provide more information on why this backscatter coefficient can be considered negligible.

- P13L18: "*Later on, the updrafts became obviously stronger, and supersaturation levels exceeded 0.2%...*" Please discuss the evidence behind this statement.
- P13L19: "*With increasing CDNC the effective radius (characterizing the typical droplet size) decreased and vice versa for constant water vapor conditions*" Please discuss the evidence behind this statement.
- Fig. 13: This graph and the text describing it on P14 is a bit difficult to understand. The temperature in Fig. 13a is said to be derived from radiosonde, so why is there only one T value shown, and does it only correspond to a height of 2.5 km? I see RH in Fig. 13b, but no temperature at all? The estimated CCNC, CDNC, INPC, and CCNC values are provided in a range. Is this range meant for a single altitude? Or across the whole figure? Or is it a point measurement range? Please specify where the values are relevant for each parameter, and why the values are only shown for that location/set of locations.
- P15L24: "*During the 7-day travel in the Arctic the Pacific airmass mixed with the smoke above 7 km. These smoke particles then served as ice nuclei when cirrus formed after further lifting.*" Please discuss the evidence behind this statement.

- Other comments:

  - In the text, when discussing CDNC values, please change from "CDNC" to "estimated CDNC" to reflect the appropriate uncertainty and to avoid confusing readers.
  - P14L20: "*The good match between CCNC and CDNC (liquid-water cloud closure) and between INPC and ICNC (ice cloud closure, see numbers in Fig. 13a) during the early phase of the altocumulus development indicates that the aerosol particles controlled the cloud properties and thus had a strong influence of the evolution of the observed altocumulus cloud system as long as the humidity conditions were favorable. It should be emphasized that such a closure study with consistent findings is only possible if primary ice and droplet nucleation dominates and secondary ice formation, ice breakup processes, crystal-crystal collision and aggregation processes, as well as droplet collision and coagulation, and strong mixing and entrainment processes are absent.*"

    This statement seems overly confident and simplistic given the very high uncertainties involved (only some of which are discussed here). Please rephrase to reflect a more accurate level of uncertainty. As an example, if I were writing this study I might say,

    "*During early altocumulus development in the Figure 13 case study, the estimated CCNC values outside of the cloud are in a similar range to the estimated CDNC levels within the cloud, as are the estimated INPC and ICNC (Fig. 13a). Thus, our data suggest that the estimated cloud active particle levels could*

*be high enough to control the case study cloud given favorable moisture conditions and in the absence of other processes that might influence CDNC and ICNC levels (e.g., secondary ice formation). This hypothesis would be in line with numerous other Arctic studies that have previously observed this phenomenon (e.g., Mautritsen et al. (2011)). However, higher resolution in-cloud microphysical data are required to verify this lidar-based hypothesis."*

- P15L19: *"To demonstrate that the observed wildfire smoke particle were able to control cirrus evolution and life time we present the results of a first MOSAiC case study here. The observation is from 6 December 2019 (Fig. 3c and d)."* Is this cloud even a cirrus cloud? The lidar signal extends down to near the surface at times, and the top is below 8 km altitude. What has been done to ensure that this is not actually a mixed phase cloud? The temperatures near the base of the cloud appear to be as high as -30C or so, from Fig. 14b, and liquid water can be present at such temperatures in the Arctic.
- P15L29: *"This part of the smoke layer (above 9.3 km) can 30 be regarded as the main reservoir of INPs."* Why is it assumed that this aerosol layer is in contact with the ice cloud? To me it looks distinctly separate for most of the time.
- I like the introduction, it really gets the reader interested in the study.
- P.3, paragraph starting on L4: Here or elsewhere, you might also consider mentioning relevant Arctic high altitude smoke findings from Schill et al. (2020).
- P4.L22 *"… HSRL is of advantage during the summer half year (when Raman lidar observations are of limited use)"* Please specify why (and if relevant, which) Raman lidar observations are of limited use during the summer. Also, can't HSRL also be used during the winter? If so, for clarity please explain to the reader what additional capabilities the Raman lidar provided that the HSRL could not.
- P4L29: *"Di Biagio et al. (2018) were the first to run lidars (mounted on an ensemble of autonomous drifting buoys) in the Central Arctic, …"* The authors might consider mentioning that these data were collected on buoys.
- P6 *" 'Co' and 'cross' denote the planes of polarization parallel and orthogonal to the plane of linear polarization of the transmitted laser pulses, respectively."* This sentence should probably go in the previous section where the authors first mention the co- and cross terms.
- Fig. 3: It might be easier on the reader to just state: "range-corrected 1064 nm signal" and "linear depolarization ratio" above the columns in the figure instead of in the caption. To avoid confusion, the authors might also want to note in the caption and/or on P7L3 that that the y- and z-axis limits were varied between panels in order to highlight different features.
- P8L5: *"The size distributions of the smoke particles were obtained from the Polly observation by applying the lidar inversion method to the layer-mean backscatter and extinction information (Veselovskii et al., 2012)."* This information would be better placed in the methods section.
- Fig. 6a: This figure is not intuitive to me. Please tell readers what the height and base of the bars indicates (the top and bottom of a smoke layer?). Please tell

them whether the colors are the relative fraction, or the dominant feature at that altitude (or something else). I am confused about the colors also because in the caption it says "The color in the bars provides information about the smoke particle concentration in terms of particle extinction coefficient at 532 nm." Please explain exactly what the particle extinction coefficient tells us about estimated smoke particle concentration. Please state in the figure and not just the caption that colors relate to particle extinction coefficient at 532 nm. Again, how do detection limits play into these bars? Please state whether the bars are only the observations above detection limits of the lidar. If the observations are below the detection limits, please either get rid of them, or clearly state why the data are still useful (I would guess they would not be). To avoid confusion, perhaps get rid of the height levels redundantly shown on the right side of the figure. Are the black dots the tropopause on that day? If so, an arrow from the word "tropopause" pointing to the dots might help clarify things. I know it was mentioned in the text, but can the authors mention in the caption as well in just a few words how the tropopause was determined?

- Fig 6b and 6c captions: Please change "column mass concentration" and "vertical mean particle mass concentration" to "estimated column mass concentration" and "estimated vertical mean particle mass concentration" to indicate appropriate uncertainty
- P9L21: "*Note that we corrected our stratospheric smoke observations in Fig. 6 for PSC effects.*" This note should go in earlier with discussion of Fig. 6.
- P9L22: "*This type is made up of supercooled liquid ternary solutions that consist of H2SO4, HNO3, and H2O.*" Speculation on the chemistry may be beyond the scope of this paper. I suggest saying "likely consist" instead of "consist."
- P9: "*The temperature at PSC base height showed values of −78°C and at the backscatter maximum the Polarstern radiosonde measured a temperature of −86°C.*" Wow, that is cold!
- P10L12: "*Height-resolved lidar observations of Arctic haze, prevailing during the late winter and early spring months, are rare (Di Pierro et al., 2013; Di Biagio et al., 2018).*" I suggest rephrasing this. The CALIPSO observations have taken observations in clear conditions over the entire Arctic since 2006, taking observations of plenty Arctic haze events.
- P10L15: "*However, knowledge about the vertical layering structures of Arctic haze is still limited and mostly based on snapshshot-like aircraft observations performed during field campaigns, preferably in spring.*" Again, I am not sure that is entirely true, given the extensive CALIPSO observations.
- P10L27: "*Type-Ia PSCs consist of nitric acid trihydrate (NAT) crystals and produce significant depolarization of backscattered laser light.*" Suggest rephrase to "…are thought to consist of …"
- P12L30: "*After nucleation, the ice crystals grew fast to sizes of 50–100 μm within minutes (Bailey and Hallett, 2012) and immediately started falling out of the altocumulus layer. The ice crystals partly evaporated on the way down, but partly*

*reached the ground as precipitation*." Please clarify here whether you are talking about findings from the Bailey and Hallet, 2012 study, or whether you are talking about results observed during MOSAiC.

- Fig. 12: Please replace "CCN" with "Estimated CCN" in the figure and caption.
- P15L19: This paragraph would benefit from a Figure showing the trajectories being discussed.
- P16L34: "*As mentioned, ice nucleation occurs during updraft periods, more precisely when a certain (threshold) supersaturation level is exceeded*." The authors may want to mention that ice nucleation also requires cold enough temperatures.
- Fig. 15: Again, please put estimated ahead of any parameters that were not directly measured and that include substantial assumptions in the caption.

**Technical comments:**

- Title: "an introductory" should be changed to "an introduction." But maybe also consider making the title more succinct to make it more appealing to readers. Note: most readers will likely not know what UTLS is, suggest dropping it from title.
- L5: "… aboard **the** Polarstern."
- Caption, Fig. 2 (and also corresponding text p. 4, L.9): "Figure 2. Polarstern drifting in the Arctic ice on 10 April 2020 (left panel) and measurement containers for in situ aerosol monitoring (the two first containers on the left side and the first container on the right side), and for remote sensing of aerosols and clouds (right panel). The OCEANET container of TROPOS is the third one on the left side." Could the authors please clarify whether they meant third one to the back, or the one in the front?
- P6L12: "Hofer et al. (2020) exemplary shows…" Did the authors mean something like, "Hofer et al. (2020) is an example showing…"?
- P7L28 "poleward"
- P14L2: profile not profil.

**References**

Ansmann, A., Mamouri, R.-E., Bühl, J., Seifert, P., Engelmann, R., Hofer, J., Nisantzi, A., Atkinson, J. D., Kanji, Z. A., Sierau, B., Vrekoussis, M., and Sciare, J.: Ice-nucleating particle versus ice crystal number concentrationin altocumulus and cirrus layers embedded in Saharan dust:a closure study, 19, 15087–15115, https://doi.org/10.5194/acp-19-15087-2019, 2019.

Ansmann, A., Ohneiser, K., Mamouri, R.-E., Knopf, D. A., Veselovskii, I., Baars, H., Engelmann, R., Foth, A., Jimenez, C., Seifert, P., and Barja, B.: Tropospheric and stratospheric wildfire smoke profiling with lidar: Mass, surface area, CCN and INP retrieval, 1–45, https://doi.org/10.5194/acp-2020-1093, 2020.

Jimenez, C., Ansmann, A., Engelmann, R., Donovan, D., Malinka, A., Seifert, P., Wiesen, R., Radenz, M., Yin, Z., Bühl, J., Schmidt, J., Barja, B., and Wandinger, U.: The dual-field-of-view polarization lidar technique: a new concept in monitoring aerosol effects in liquid-water clouds – case studies, 20, 15265–15284, https://doi.org/10.5194/acp-20-15265-2020, 2020a.

Jimenez, C., Ansmann, A., Engelmann, R., Donovan, D., Malinka, A., Schmidt, J., Seifert, P., and Wandinger, U.: The dual-field-of-view polarization lidar technique: a new concept in monitoring aerosol effects in liquid-water clouds – theoretical framework, 20, 15247–15263, https://doi.org/10.5194/acp-20-15247-2020, 2020b.

Mamouri, R.-E. and Ansmann, A.: Potential of polarization lidar to provide profiles of CCN- and INP-relevant aerosol parameters, 16, 5905–5931, https://doi.org/10.5194/acp-16-5905-2016, 2016.

Mauritsen, T., Sedlar, J., Tjernström, M., Leck, C., Martin, M., Shupe, M., Sjogren, S., Sierau, B., Persson, P. O. G., Brooks, I. M., and Swietlicki, E.: An Arctic CCN-limited cloud-aerosol regime, Atmos. Chem. Phys., 11, 165–173, https://doi.org/10.5194/acp-11-165-2011, 2011.

Schill, G. P., Froyd, K. D., Bian, H., Kupc, A., Williamson, C., Brock, C. A., Ray, E., Hornbrook, R. S., Hills, A. J., Apel, E. C., Chin, M., Colarco, P. R., and Murphy, D. M.: Widespread biomass burning smoke throughout the remote troposphere, 13, 422–427, https://doi.org/10.1038/s41561-020-0586-1, 2020.

---

## Referee Comment (RC2) · Anonymous Referee #2 · 28 Mar 2021

The paper "UTLS wildfire smoke over the North Pole region, Arctic haze, and aerosol-cloud interaction during MOSAiC 2019/20: An introductory" presents and discusses remote sensing observations obtained during the MOSAiC campaign. The unique dataset provides new opportunities to explore aerosol-cloud interactions at the North Pole. Persistent smoke layers originating from large scale fires are discussed and analyzed in terms of their properties as well as the ability to act as Cloud Condensation Nuclei and Ice Nuclei (CCN and IN). Observations of the Arctic haze during the campaign are presented and discussed also, aiming to enrich the current observational dataset with winter time measurements.

The study falls within the scope of ACP. The manuscript is well-written and structured, the presentation is clear, the language is fluent and the quality of the figures are high. In order to help improve the manuscript, I would kindly suggest the authors to take into account the following comments.

**General comment**:
Please provide the typical uncertainties of the lidar-derived aerosol and cloud microphysical properties discussed, originating from the conversion assumptions and the uncertainties of the optical properties derived from the lidar measurements. Additionally, please discuss the effect of the low aerosol concentrations presented here, on these retrievals. Please provide information on independent validation studies of the lidar-derived microphysical products (e.g. using in-situ measurements).

**Specific comments:**

**Page 2, lines 29-32:** "The MOSAiC lidar observations (together with the radar observations of the ARM mobile facility) allow us, for the first time, to investigate the role of smoke in ice formation processes". The authors should specify the Arctic region in the sentence, since there have been several studies exploring the potential of smoke particles to act as IN (.i.e. Peters et al., 2009; Prenni et al., 2012; Levin et al., 2016; Schill et al., 2020). Or specify if the statement is meant for combined lidar and radar observations in the Arctic.

**Page 3, lines 1-3:** "A unique opportunity is thus given to explore to what extent the wildfire smoke particles, providing a significantly enhanced number of sites for heterogeneous chemical processes (chlorine and bromine activation), contributed to the strong ozone depletion": Since the article provides hints on the role of smoke particles on ozone depletion, rather than "findings", please rephrase accordingly.

**Page 3 lines 4-30:** "The article is organized as follows… Sect. 4 finally provides some concluding remarks": This section is hard to follow, since it mixes the scientific objectives of the study with the proposed methodology/techniques and the article layout. Please divide this part in three paragraphs, with the first containing the scientific objectives of the study (e.g. "Organic aerosol particles are ubiquitous in the atmosphere around the world and there is an urgent need to investigate..."), the second one containing the proposed methodology/techniques and the third containing the structure of the article with very brief descriptions for its section (e.g. "... In Sect. 3.2, we present two cases of Artic haze observations performed in February and March 2020").

**Page 3, lines 21-22:** "Recently introduced new remote sensing analysis concepts (closure studies) (Ansmann et al., 2019) are applied for the first time to Arctic clouds". Please also include the work of Marinou et al., (2019).

**Page 5, lines 9-10:** "which permits accurate aerosol and cloud profiling from about 800 m to 30-40 km height". Please specify separately the information on aerosol and cloud detection ranges and provide information on the accuracy of the aerosol and cloud retrievals for different altitude ranges. Which are the typical signal-to-noise (SNR) values of this lidar for aerosol layers at 10, 15 and 20km a.s.l? What are the uncertainties of the lidar-derived properties at these altitudes?

**Page 5, lines 16-19:** "This technique enables us to determine multiple scattering by droplets in liquid-water dominated cloud layers and to determine from this multiple scat-tering information cloud microphysical properties (e.g. effective droplet size and cloud extinction coefficient) (Jimenez et al.,2020a). The method is based on depolarization ratio (ratio of cross-to-co-polarized backscatter coefficient) observations at the two FOVs". This part is more appropriate for section 2.2 where the rest of the lidar products and retrievals are presented. I suggest to move it before **page 6, line 15-16**: "Details of the retrieval of microphysical properties of liquid-water cloud layers can be found in Jimenez et al. (2020a, b)".

**Page 5, lines 26-27:** "we use the preliminary radiosonde products that were directly available during the expedition". Explain the "preliminary" definition in the radiosondes used, and why you used these instead of the consolidated radiosonde products.

**Page 6, lines 9-16:** "The retrieval of aerosol microphysical properties such as particle volume, mass, and surface area concentration and estimates of cloud-relevant properties (aerosol-type-dependent cloud condensation nuclei, CCN, and ice-nucleating particles, INPs) is performed by means of the POLIPHON (Polarization Lidar Photometer Networking) approach (Mamouri and Ansmann, 2016,2017; Ansmann et al., 2019, 2020). Hofer et al. (2020) exemplary shows the full set of POLIPHON aerosol products in the cases of an 18-month Polly campaign in Dushanbe, Tajikistan, for central Asian aerosol. Alternatively to the POLIPHON method,we used the multiwavelength lidar inversion technique (Müller et al., 1999, 2014; Veselovskii et al., 2002, 2012) to derive microphysical properties of aerosols including the particle size distribution for detected pronounced aerosol layers. Details of the retrieval of microphysical properties of liquid-water cloud layers can be found in Jimenez et al. (2020a, b)". Please provide a short description of the assumptions used for the aerosol and cloud microphysical properties retrieved from the lidar products. Please comment on independent validation studies for these products (e.g. with in-situ measurements as in Marinou et al. (2019) study). Please quantify and discuss the uncertainties of the aerosol and clouds microphysical retrievals for the observations presented on this study.

**Page 7, line 2:** "The measured linear depolarization ratio in the right panels of Fig. 3 allows us to precisely distinguish cirrus from layered mixed-phase clouds as explained above". Above you mention in **page 3, line 15:** "We start with a case of a shallow mixed-phase cloud consisting of a liquid-water layer on top of the ice virga zone" and in **page 6, line 6:** "...in the case of clouds, liquid-droplet layers show PLDR≈0 at layer base where light depolarizing multiple scattering is low, and PLDR of 0.4-0.6 in,

e.g., cirrus layers", but you haven't explained how you distinguish cirrus from layered mixed-phase clouds in depolarization measurements above. So I suggest to skip "as explained above '' or explain it.

**Page 7, lines 22-25:** "The light-absorption-related lifting occurs during the spread of the smokeover the respective hemisphere and continues as long as the smoke layers are optically dense enough (aerosol optical thicknessAOT>1-2 at 500 nm) with the consequence that the smoke reaches, e.g., the Central Arctic at heights up to 5-10 km above the tropopause". Please provide a reference for these AOT required conditions.

**Page 7, line 33:** "The 532 nm lidar ratio is much larger than the 355 nm lidar ratio. ": please quantify how larger it was.

**Page 8, line 5:** "These specific optical properties are linked to the narrow size distribution of absorbing smoke particles which form a well-defined accumulation mode as shown in Fig. 5". Please provide references to support this claim. Moreover, discuss the role of the shape and refractive index of the smoke particles in defining the unique optical properties measured.

**Page 8, line 15:** "With increasing age the core structure obviously collapses, gets compact, and the particles become more and more spherical with time (Baars et al., 2019)". Please revise obviously as probably.

**Page 8, line 27:** "Downward mixing and transport into the lower troposphere had no impact on the UTLS AOT as well". Please explain this statement in more detail. How can this be supported, when the AODs observed are decreasing with time from 0.12 to <0.03 during the time period discussed?

**Page 9, line 7:** " But this smoke layer had no clear boundaries, at least no clear upper boundary (see Figure 4a)" and **page 8, line 30**: "The layer-mean 532 nm smoke extinction coefficients in Fig. 6c (obtained from the ratio of AOT divided by the layer geometrical depth in Fig. 6a)". Please include a comment on the effect of the unclear layer's boundary to the lidar retrievals presented in this work (e.g. the effect to the AOD and layer top heights).

**Page 9, lines 1-10:** "It is noteworthy to mention that the CALIPSO data base...aerosol observations and corroborate our hypothesis". Please revise taking into consideration the Interactive comment of Jayanta Kar on the detection of the smoke layers from CALIPSO.

**Page 9, line 21:** "Note that we corrected our stratospheric smoke observations in Fig. 6 for PSC effects". Please explain how.

**Page 9, lines 33-34:** "According to Vaughan et al. (2020), the volcanic aerosol layer consisting of sulfuric-acid-containing water droplets (75% H2SO4, 25% H2O) formed above the tropopause with maximum heights reaching 21 km". In Vaughan et al. 2020 paper they reported that the volcano "...send a plume of ash and sulphur dioxide into the stratosphere...", "the ash and sulphur dioxide plume initially moved westward before being entrained in a cyclonic circulation over the North Pacific" and "During the latter half of June and in early July, pyroconvection over Canada injected layers of smoke and ice clouds into the lower stratosphere (similar to the case described by Vaughan et al.

(2018)), making it difficult to distinguish the progression of volcanic ash remnants using the CALIOP profiles". Why do you not mention the ash contribution on these layers? As the smoke layers from these fires arrived in the MOSAiC altitudes and are discussed herein, why do you exclude possible mixture with the volcanic ash?

**Page 9, line 34 - page 10, line 2:** "From the 355 nm Raman lidar observations at Capel Dewi Atmospheric Observatory, United Kingdom (52.4°N, 4.1°W) it can be concluded that the 532 nm AOT was about 0.03 over UK in August 2019, of the order of 0.01 in December, and of around 0.005 during the first months of 2020, respectively". Please provide a reference for these AOTs.

**Page 10, lines 10-14:** "The original and primary goal of the shipborne MOSAiC lidar measurements was to provide, for the first time, a seasonally and height-resolved characterization of tropospheric aerosols and clouds for the North Pole region. Especially the coverage of the winter half year can be regarded as a valuable new contribution to Arctic aerosol research. Height-resolved lidar observations ofArctic haze, prevailing during the late winter and early spring months, are rare (Di Pierro et al., 2013; Di Biagio et al., 2018)". As there are plenty of CALIPSO overpasses in the Arctic region, please rephrase this part including for example the specific latitudes of the campaign.

**Page 10, line 22:** "The measurements are representative for many days during the winter months". Please quantify the days arctic haze was observed in the period of the experiment.

**Page 10, lines 23-24:** "The most striking feature in both figures is that aerosol layers were detected everywhere up to the tropopause, and because of the smoke layer even from 8 to almost 20 km height". This is not visible in these figures which are up to 15 and 16 km. Please state if you refer to fig. 6, which shows smoke up to 17km during March and April, and smoke up to 20km in February.

**Page 10, lines 27-28:** "Type-Ia PSCs consist of nitric acid trihydrate (NAT) crystals and produce significant depolarization of backscattered laser light". Please rephrase to "...are thought to consist of ...".

**Page 11, lines 6-7:** "The Ångström exponent for the extinction coefficient was around 1.7 in the lofted layer above 3 km height. The lidar ratios were high with values close to 100 sr in the lofted layer on 4 March". Please provide the uncertainties or standard deviation of these values.

**Page 11, lines 6-7:** "The 532 nm AOT was 0.024 (4 February, for the lowest307 km height) and 0.022 (4 March, for the lowest 5 km)...". Please provide the uncertainties/errors of these values.

**Page 12, lines 29-30:** "Favorable conditions with cloud top temperatures around −28.5C at 2.6 km height (at 03:00 UTC) were given for heterogeneous ice formation via immersion freezing, i.e., ice nucleation on INPs immersed in the water droplets". Please provide relevant reference for the "favorable conditions".

**Page 12, lines 30-32:** "After nucleation, the ice crystals grew fast to sizes of 50–100µm within minutes (Bailey and Hallett, 2012) and immediately started falling out of the altocumulus layer". Please clarify whether you refer to findings from Bailey and Hallet (2012) or to result from this specific case. If the

second appy, please provide a short description on the measurements/methods used for these findings.

**Page 13, lines 3-4:** "As discussed below in detail, there were always 20-200 droplets per cm3 in the altocumulus top layer, but only 0.1 to 1 ice crystals per liter". Please rephrase so as to be clear that these concentrations are estimated and not measured.

**Page 13, lines 12-14:** "This condition holds here with ice crystal backscatter coefficients of 5-10 Mm−1sr−1 in the virga and thus also in the cloud top layer and droplet backscatter coefficients of the order of 700 Mm−1sr−1". Please include the information that these values are not shown in the study.

**Page 13, line 15:** "As can be seen in seen Fig. 11, the CDNC values were about 20 cm−3 in the beginning and around 100 cm−3 later on". Please rephrase to "estimated values" or "retrieved values"

**Page 13, lines 16-18:** "Obviously, updraft velocity was weak and correspondingly water super saturation levels were below 0.2% so that fewer particles were activated to become cloud droplets as predicted (CCNC values in Fig. 12). Later on, the updrafts became obviously stronger, and supersaturation levels exceeded 0.2% so that more CCN nucleated cloud droplets as predicted by the retrievedCCNC values". Please rephrase or justify "obviously".

**Page 13, line 20:** "The cloud extinction coefficient showed typical values from 10-20 km−1most of the time in the droplet-dominated cloud layer". The plot shows cloud extinction coefficient values <10 km^-1 from 7:45 until the end of the cloud (~3hrs) and values between 10-20 km^-1 for ~2.5hrs. Please rephrase accordingly.

**Page 13, line 30:** "Soot and mineral dust particles are good candidates to serve as INPs". Please provide relevant references (i.e. Sassen and Khvorostyanov, 2008; Boose et al., 2016; 2019).

**Page 13, lines 31-33:** "Dust is left as potential INP. Our polarization lidar observations indicated the presence of a dust fraction of 3-10% according to the slightly enhanced particle depolarization ratio (not shown) above 2 km height..." Earlier, in page 9, you mentioned the influence of aerosol particles from "a strong eruption of the Raikoke volcano in the Kuril Islands…". Why do you exclude the possibility of the presence of ash non-spherical particles? Please discuss the effect of possible ash particle presence on the INP retrievals you provide.

**Page 14, lines 1-2:** "We used the INP parameterization scheme of DeMott et al. (2010) to estimate the dust INP concentration for immersion freezing. Here, the particle number concentration n250 of dust particles with diameters >500 nm is an input parameter and obtained from the respective lidar observation of the extinction coefficient in Fig. 12 and by assuming a dust fraction of 3-10% in the conversion of the extinction profile into the n250 profil". Please discuss why you use the INP parameterization scheme of DeMott et al. (2010), which was developed with minimum presence of dust particles in the analysed samples, to convert the lidar derived dust n250 concentrations to INP, instead of the dust-tailored INP parameterization scheme of DeMott et al. (2015). Can the authors comment on the effect of the dust-tailored parameterization to the lidar-derived INPC in the study?

**Page 14, line 17:** "until a dry air mass approached, leading to a strong decrease in relative humidity and dissolution of the stratiform cloud deck": Please support this discussion with updraft and wind data, if available in MOSAiC. Moreover, please provide a colorbar with more points so that it is easier for the reader to understand which values you are referring to.

**Page 14, line 19:** "ice crystals could partly reached the ground as precipitation". for convenience to the readers, you could refer to figure 3f, where this is evident.

**Page 14 line 20:** "The good match between CCNC and CDNC (liquid-water cloud closure) and between INPC and ICNC". Please rephrace as "the good match between the estimated CCNC.." or "the good match between the retrieved CCNC..".

**Page 14, lines 32-33:** "Organic aerosol (OA, the main aerosol component in wildfire aerosol) is besides dust and marine particles ubiquitous in the atmosphere". Please comment on the presence of ash particles, as mentioned in previous comments.

**Page 15 lines 20-25:** "HYSPLIT backward trajectory analysis ... During the7-day travel in the Arctic the Pacific airmass mixed with the smoke above 7 km". Consider including a figure showing the trajectories discussed, even in an appendix.

**Page 16, lines 13-15:** "Ice supersaturation conditions are usually given or produced during updrafts (e.g., during the ascending period of a gravity wave) that could, in principle, be detected and measured with the AMF-1 Doppler radar". Is this information available for the case discussed here?

**Page 17, line 2:** "600 s may represent here a typical time period of the lifting phase of a gravity wave". Please provide a reference on this, or discuss if these updrafts were actually observed during MOSAiC.

**Page 17, lines 2-5 and lines 13-16:** "As can be seen the nICE and nINP,I values (blue and red bars) are in the same range of values which suggests that organic particles may be able to control the evolution of the cirrus layer via the immersion freezing mode ... The impact of deposition INP nINP,D (cyan and orange bars) is comparably weak in this example… However, the successful closure, indicated by a reasonable match between nINP,I and nICE, indicates that the wildfire smoke was able to trigger cirrus formation (before homogeneous freezing can take place on stratospheric background or even liquid smoke particles) and control of the further evolution of the ice cloud system". The authors should consider the possibility that the discussed cloud was formed in an aerosol reacher environment. As they show in fig. 6, all the dust plumes observed in the period between 1/10 to 30/11, and for altitudes up to 11 km, had extinction coefficient values >10 Mm^-1. Also, indicatively, in Fig. 7 and Fig. 10, the aerosol profiles show that the extinction coefficient values in the middle of the layers were respectively 5 and 8 times higher than the values in the edges of the layers. It would be good if the authors take into consideration that the n_INP,D abundance could have been significantly higher in the time of the formation of the cloud, and discuss how this affect their conclusion that the immersion freezing process with uplifting support from a gravity wave is the main driver on the n_ice observed in this case.

**Figure 5:** The size distribution retrievals in Fig. 5 need further support with a) the provision of the corresponding retrieval errors (as these are provided in e.g. Veselovskii et al. (2012)) and b) the comparison with other studies for stratospheric smoke. The effect of the (quite) low AODs on the retrieval errors should be also discussed.

**Figure 6:** Please provide the products uncertainties in figures 6b and 6c. Also, I would suggest changing legend "col. mass" in fig. 6b to "col. mass conc.", and legend "Mass conc" to "Vert. mean. mass conc." for completeness.

**Figure 7:** "The 532 nm backscatter ratio (total-to-Rayleigh backscatter) peaks at 2.43". This parameter is not presented in the plot. Please include it or this give more information on this sentence to be in the context of the figure.

**Figure 7:** ". PSC optical thickness was 0.0125 at 532 nm(computed from backscatter values multiplied by a lidar ratio of 50 sr". Please explain in the relevant section (in page 9) why you chose this LR value (e.g. Ohneiser et al., (2021) paper).

**Figure 8:** The VDR quicklooks of these cases would be of interest to the reader. Additionally, the addition of the backscatter profiles from 0-15km, which can support the argument of the authors that "There were no regions with a negligible aerosol content" in **page 10, line 24.**

**Figure 13:** "The range-corrected 1064 nm signal in (a) is biased by an detector overload in the nearest height range to the lidar". Include the height you are referring to.

**Figure 13:** In Fig. 13b (as well as in Fig. 3f) the lidar signal above the cloud seems to be totally attenuated. Please provide additionally the collocated radar measurements to show the extent of the whole cloud and support the cloud top provided from the lidar.

**Figure 15:** To simplify the plot and make it more clear for the reader, the authors could consider moving the n_ice numbers to fig. 15c, where the profiles of these values are plotted.

**Figure 15:** "derived ranges of INP number concentrations nINP,I (for immersion freezing, blue and red, indicating the respective cirrus layers in (a)". Please revise this part to state clearly these n_INP concentrations from which profiles/days are derived.

**Technical corrections:**
Page 1, line 2-5: Long sentence. Consider splitting it to two.

Page 5, line 22: "procedures, and to": skip and

Page 6, line 3-8: "The linear depolarization ratio is defined as the cross-polarized-to-co-polarized backscatter ratio and allows us to sensitively distinguish spherical particles showing particle linear depolarization ratios (PLDR) close to zero from non-spherical aerosol particles showing PLDR values of typically 0.1-0.3. In the case of clouds, liquid-droplet layers show PLDR≈0 at layer base where light depolarizing multiple scattering is low, and PLDR of 0.4-0.6 in, e.g., cirrus layers. "Co" and "cross"

denote the planes of polarization parallel and orthogonal to the plane of linear polarization of the transmitted laser pulses, respectively". It will be better read if the "co" and "cross" definitions are closer to the parameter "cross-polarized-to-co-polarized backscatter ratio".

Page 7, line 11: "In summer, warm, moist and polluted air massed..". Typo: masses.

Page 7, line 31: "The internal vertical structures were rather smooth and indicate an aged smoke layer": change "and" to "which".

Page 8, line 16: "well established and obviously prohibited any mixing". Please delete "obviously".

Page 13, line 13: "As can be seen in seen Fig. 11". seen duplicate.

Figure 13: "... (a) is biased by an detector overload". Typo: a.

Page 14, lines 32-33: "...(OA, the main aersol component..). Typo: aerosol .

Page 18, line 1: "As an outlook". The authors could consider to revise this to "future work".

**References**
Petters, M.D., Parsons, M.T., Prenni, A.J., Demott, P.J., Kreidenweis, S.M., Carrico, C.M., Sullivan, A.P., McMeeking, G.R., Levin, E., Wold, C.E., Collett Jr., J.L., Moosmüller, H.: Ice nuclei emissions from biomass burning (2009) Journal of Geophysical Research Atmospheres, 114 (7), art. no. D07209, DOI: 10.1029/2008JD011532

Schill, G.P., DeMott, P.J., Emerson, E.W., Rauker, A.M.C., Kodros, J.K., Suski, K.J., Hill, T.C.J., Levin, E.J.T., Pierce, J.R., Farmer, D.K., Kreidenweis, S.M.: The contribution of black carbon to global ice nucleating particle concentrations relevant to mixed-phase clouds (2020) Proceedings of the National Academy of Sciences of the United States of America, 117 (37), pp. 22705-22711, DOI: 10.1073/pnas.2001674117

Levin, E.J.T., McMeeking, G.R., DeMott, P.J., McCluskey, C.S., Carrico, C.M., Nakao, S., Jayarathne, T., Stone, E.A., Stockwell, C.E., Yokelson, R.J., Kreidenweis, S.M.: Ice-nucleating particle emissions from biomass combustion and the potential importance of soot aerosol (2016) Journal of Geophysical Research, 121 (10), pp. 5888-5903, DOI: 10.1002/2016JD024879

Prenni, A.J., Demott, P.J., Sullivan, A.P., Sullivan, R.C., Kreidenweis, S.M., Rogers, D.C.: Biomass burning as a potential source for atmospheric ice nuclei: Western wildfires and prescribed burns (2012) Geophysical Research Letters, 39 (11), art. no. L11805, DOI: 10.1029/2012GL051915

Sassen, K., Khvorostyanov, V.I.: Cloud effects from boreal forest fire smoke: Evidence for ice nucleation from polarization lidar data and cloud model simulations (2008) Environmental Research Letters, 3 (2), art. no. 025006, DOI: 10.1088/1748-9326/3/2/025006

Boose, Y., Welti, A., Atkinson, J., Ramelli, F., Danielczok, A., Bingemer, H. G., Plötze, M., Sierau, B., Kanji, Z. A., and Lohmann, U.: Heterogeneous ice nucleation on dust particles sourced from nine deserts worldwide – Part 1: Immersion freezing, Atmos. Chem. Phys., 16, 15075–15095, https://doi.org/10.5194/acp-16-15075-2016, 2016.

Boose, Y., Baloh, P., Plötze, M., Ofner, J., Grothe, H., Sierau, B., Lohmann, U., and Kanji, Z. A.: Heterogeneous ice nucleation on dust particles sourced from nine deserts worldwide – Part 2: Deposition nucleation and condensation freezing, Atmos. Chem. Phys., 19, 1059–1076, https://doi.org/10.5194/acp-19-1059-2019, 2019.

Marinou, E., Tesche, M., Nenes, A., Ansmann, A., Schrod, J., Mamali, D., Tsekeri, A., Pikridas, M., Baars, H., Engelmann, R., Voudouri, K.-A., Solomos, S., Sciare, J., Groß, S., Ewald, F., and Amiridis, V.: Retrieval of ice-nucleating particle concentrations from lidar observations and comparison with UAV in situ measurements, Atmos. Chem. Phys., 19, 11315–11342, https://doi.org/10.5194/acp-19-11315-2019, 2019.

---

## Author Comment (AC1) · 17 Jun 2021

**Dear Dr Kar!**

**Thank you for careful reading of the long manuscript and the valuable comments on the CALIPSO observations.**

**Our answer in blue** to your comment in black.

Contrary to this assertion, we do find clear evidence of layers detected by CALIPSO over the high latitude regions. The layers have been classified as sulfate/other by the CALIPSO stratospheric aerosol subtyping algorithm because of the low backscatter (Kim et al., 2018). Figure 2 shows another example of aerosol layers detected on November 14, 2019.

It is therefore rather surprising that the authors chose to make the statement about the lack of layer detection by CALIPSO at high latitude UTLS region during the period of their observation.

**To avoid a lengthy discussion, we agree with your comment and removed the two paragraphs.**

On another point, in page 8, lines 12-15, the authors try to explain the low depolarization ratio (~0.01) in the aged smoke in terms of the core-shell structure collapsing and the particles becoming more spherical. This seems to contradict the results of Ohneiser et al. (2020) who found high depolarization ratio (up to 0.2 at 532 nm) for aged pyroCb plumes transported from Australia over Argentina in January 2020. In fact, as shown by Christian et al. (2020), the depolarization ratio in the pyroCb plumes from the Canadian pyroCb in August 2017 continued to increase with time for several weeks.

**Smoke, lifted by pryoCb into the stratosphere shows enhanced depolarization (20% at 532 nm). Because this lifting is fast (<120 min), there is no time for aging processes. As a consequence the particles are non-spherical when the enter the lower stratosphere**

**However, when lifting is caused by self-lifting effects, particle aging processes can take place over 3-5 days. At the end of (any) aging process, smoke particle typically show a perfect structure of a core part surrounded by a spherical shell. And when these smoke particles enter the lower stratosphere (by self lifting) they show low depolarization ratios. And that is what we observed now during the MOSAiC expedition. Note, that we still observed the inverse lidar ratio behavior (LR355 significantly lower than LR532) and high LR532 (on average 85sr) so that there is no way to assign these layers as volcanic sulfate aerosol layers.**

**You mention that Christian et al. (2020) observed that the depolarization in the stratosphere increased with time. We cannot support this (Baars et al, ACP, 2019). We saw a decrease with time (over month). Obviously, particle aging was totally prohibited in the observations discussed by Christian et al. (2020), and in the dry stratosphere, the particles could keep their non-spherical shape for a long time.**

---

## Author Comment (AC2) · 17 Jun 2021

Dear reviewer!

Thank you for careful reading of the long manuscript and all the valuable comments and suggestions. Before we provide our answers, step by step, let us summarize the main changes.

- We changed the title!
- In Sect. 2, we added typical uncertainties in the lidar products in Table 1, and  provide 4 paragraphs on validation efforts, as requested.
- Concerning the apparent contradiction of good lidar observation in an area with rather low aerosol content, we have the following answers: (1) There was complete darkness for 5 months, so almost 'unlimited' signal averaging was possible. (2) We do photon counting in all channels, no analog detection at all, so linear signal response over six orders of magnitude.  (3) We have radiosonde temperature and pressure profiles every six hours, so accurate Rayleigh scattering and temperature profile information (in the extinction computations) is available. Consequently, the small aerosol effects could obviously be accurately separated from the Rayleigh backscatter and extinction properties.
- We re-analyzed all data shown in the figures.
- The tropopause computations contained a bug, is now correct.
- We included new figures with backward trajectories (Fig, 12, Fig, 16).
- Fig. 14 (mixed-phase cloud closure study) now includes four radiosonde observations.
- The cirrus closure study is enlarged to make it more understandable. Now, we have four figures in Sect. 3.4 instead of two (submitted version). We cannot leave this study out. It is a highlight because, for the first time, the impact of aged smoke (organic material) on cirrus formation is discussed based on real-world observations.

**In the revised version of the manuscript we indicate the essential changes IN BOLD. Therefore, not every small change is indicated.**

**Step by step reply:** our answers in blue

**Overview:**

This study has some really interesting data that are definitely worth sharing with the scientific community in some form. It is great to see these high latitude data from locations where there were hardly any data at all before. And as the authors pointed out, these data offer very valuable information to contrast and compare with CALIPSO, which up until now has been one of the only sources of lidar information at remote Arctic sea ice locations (but which misses the highest latitude areas and has problems with lidar ratios).

However, the methodologies and associated uncertainties in this paper were not well described, and many of the conclusions were not well supported by the presented data. In particular, a lot of new and interesting techniques are used that are not well validated, but the resulting data presented as if they are known to be accurate. For this reason, I honestly do not know whether I should recommend this paper to be published or not, and I would like to re-evaluate it after the authors have been given a chance to better characterize the uncertainties and reframe the discussion in context of these uncertainties. See more specific comments

below.

**Specific comments:**

- The Punta Arenas and Cyprus data the authors cite for CCN and INP validation of this work (Jimenez et al., 2020a; Ansmann et al., 2020, 2019; Mamouri and Ansmann, 2016) were not actually validated with *in situ* data. I think it is really important to be upfront about this fact, which suggests an unknown degree of uncertainty in the CCN and especially INP estimates for these Arctic data. It should also be mentioned that the Arctic environment is colder and cleaner than these other locations, and has different types of aerosol particles, which might affect the estimates and render previous validations efforts less useful here.

**We included 4 paragraphs on validation studies in Sect.2.2. We include typical uncertainties in Table 1.**

**We changed the CCNC estimation. We make use of the multiwavelength lidar technique (Veselovskii, 2002). The uncertainties should then be around 30%. In this specific approach, we do not need to know the aerosol type so that uncertainties are lower than given in Table 1 (for the conversion method).**

**All the discussion about Arctic aerosols is given in Sect. 3.2. There is already so much paper work (including review articles) so that we try to keep the discussion short.**

- Many high altitude Arctic AOTs will be very small. How can we be sure that determinations about particle properties can be made at such small signals?

**We show uncertainty bars and we give AOT uncertainties in the discussion. AOTs down to 0.003 or even lower is not a problem for lidars. This is what our long experience tells us.**

- Can the authors take advantage of the available complementary MOSAiC data (e.g., with INP and CCN near the surface or from tethered balloon) to somehow better validate the data?

**No! Surface observations have nothing to do with the aerosol at 2.5 or 8-9 km height.**

- I have made various comments below asking the authors to better describe the uncertainties in various parameters. But I think it is very likely in many cases that uncertainties may not be easy to describe because cloud and aerosol parameters estimated from lidar depend not only on things like conversion coefficients and assumptions about mineral dust, but also on variables like optical thickness of the cloud, and the extent to which the lidar signal has been attenuated. For example, note how the signal has been attenuated beyond the top of the cloud in Fig. 3. Therefore, there is a fundamental challenge when trying to use the methods in this paper to compare quantities like estimated CDNC between clouds, or even between the base and top of clouds for a case study. I am not sure how the authors can address these issues. Possibly a sensitivity study might help.

**We used the multiwavelength lidar technique to obtain the particle number concentration n50. This quantity (n50, particles with radius > 50nm) is a good estimation for CCNC. The CCNC uncertainty is then of the order of 30%. The dual FOV polarization lidar technique determines the cloud properties for 75 m above cloud base! The values are NOT cloud mean values. Our dual-FOV lidar values represent freshly formed droplets,**

just above cloud base. This  is now emphasized several times. There is no problem with strong cloud attenuation. Finally, the atmospheric condition on 10 December (Sect. 3.3) were not complex so that a straighforward  lidar data analysis was possible.

- Places where uncertainties need to be better described:

  - P6L9: "*The retrieval of aerosol microphysical properties such as particle volume, mass, and surface area concentration and estimates of cloud-relevant properties (aerosol-type-dependent cloud condensation nuclei, CCN, and ice-nucleating particles, INPs) is performed by means of the POLIPHON (Polarization Lidar Photometer Networking) approach (Mamouri and Ansmann, 2016, 2017; Ansmann et al., 2019, 2020)*" Please describe the validation for and uncertainties in this measurement in greater detail. For example, in Jimenez et al. (2020b) it is mentioned that uncertainties in lidar-derived CCN are around 50%, but that is not mentioned here. Please quantify the uncertainties, discuss how were derived and where they cannot be quantified, and how this information affects the interpretation of these results.
    **We improved this in Sect 2.1 and 2.2 with all the information about uncertainties in Table 1, and 4 paragraphs filled with information about validation efforts. More is not possible in this paper with focus on MOSAiC results.**
  - P6L13: "*Alternatively to the POLIPHON method, we used the multiwavelength lidar inversion technique (Müller et al., 1999, 2014; Veselovskii et al., 2002, 2012) to derive microphysical properties of aerosols including the particle size distribution for detected pronounced aerosol layers.*" Please describe the validation for and uncertainties in this measurement in greater detail.
    **Done! Sect.2.2.**
  - Fig. 5: How well validated are these data? Can error bars in these measurements be applied to this figure?
    **Error bars are not helpful. Smoke shows just ONE mode (an aged accumulation mode, aged means here ...shifted to larger particles). This is found in many aircraft observations (e.g., Dahlkoetter et al. 2014). And in these simple cases, even ill-posed methods have no problems to identify and quantify the size distribution. We mention validation efforts in Sect.2.2.**
  - Fig. 6. These are extremely low AOTs. It would be helpful to indicate instrument detection limits on both figures, and to show uncertainty bars, as I would expect these to get increasingly large at low AOTs. The discussion of uncertainties in these data, and how they relate to the conclusions of the study should be further expanded upon in the text.
    **We had no detection-limit problems, and AOTs of 0.01 are not extremely small. Most subvisible cirrus show AOTs from 0.005 to 0.03, and there are many lidar papers on subvisible cirrus. But you need backscatter coefficients (Raman extinction does not work in such cases with low AOT), and then you need proper lidar ratios (for cirrus typically values around 30sr at 532nm, and in the case of the MOSAIc smoke layer we used 85 sr.).**
  - Fig 10: what are the detection limits?
    **We never checked that in detail because we have no problems to see the 532 and 1064 nm signals up to more than 30 km height. In the case of the 1064 nm signal (photon counting mode), the detection limit may be about 0.001 Mm-1 sr-1 in terms of backscatter.**
  - Fig. 11: Please discuss whether these are averages over the cloud layer, and if so how that cloud layer was determined. Please change to "estimated effective radius" and "estimated droplet number" in the figure and caption. Please describe in the

methods text how the uncertainty range was determined, and discuss the extent to which this uncertainty is meaningful.

**We improved the text keeping all the suggestions regarding retrieved or estimated into account. But we do not want to present a lengthy discussion that was given in foregoing papers. We tried to find a balance. The values show the microphysical droplet properties at 75 m above cloud base. That's it! We state that more clearly now. We cannot explain everything in large detail. The reader has to read the paper of Jimenez et al. (2020b), if he/she is interested.**

- P13L27: "In this way we estimated CCN concentrations of about 30-70 cm$^{-3}$ with an uncertainty of a factor of 2." Please describe how this uncertainty factor was estimated, and why this uncertainty estimate is meaningful.

**This is described in Mamouri and Ansmann (2016), impossible to repeat that in this MOSAiC paper. The uncertainties are found from simple correlation studies (log(n50) vs log(extinction coefficient)).**

- P14L2: "*Here, the particle number concentration n250 of dust particles with diameters > 500 nm is an input parameter and obtained from the respective lidar observation of the extinction coefficient in Fig. 12 and by assuming a dust fraction of 3-10% in the conversion of the extinction profile into the n250 profil (Mamouri and Ansmann, 2016).*" Please provide the uncertainties in the input parameter of dust particle concentrations with diameters > 500 nm and discuss the impact of these uncertainties on the INP estimates? Why assume a dust fraction of 3-10%? As it reads now, there seems to be very large uncertainties, with estimated INP levels based on unsubstantiated assumptions. Hopefully further discussion can clarify this.

**We improved the text, the dust fraction is estimated from the slightly enhanced particle depolarization ratio. We rechecked the depolarization ratios and concluded finally: 5% dust. The uncertainty in the INP estimate is always dominated by the DeMott parameterization (one order of magnitude uncertainty) and not by the 25% uncertainty in the estimate of n250.**

- P16L30: "*In this figure, the number concentration of large smoke particles n250 (with radii > 250 nm, lower axis) is shown as well. This number indicates the overall reservoir of favorable INPs (Ansmann et al., 2020).*" Please change to "*estimated number concentration of large smoke particles n250.*" Please explain why this estimate should give us the overall reservoir of favorable INPs, and discuss associated uncertainties.

**We improved that. The larger the particles the better their INP potential because surfacc characteristics (caves and cracks) play also a role. The reader has to read the INP paper of Kanji 2017…. to get a good idea about all this.**

- Section 3.4: I find the last 2 paragraphs of section 3.4 to be not useful and mainly conjecture because there are so many assumptions. I suggest removing these paragraphs and Fig. 15 entirely.

**No! Impossible! We cannot remove Sect 3.4! It is one of our highlights! And in the INP community, they wait exactly for such studies, disregarding how uncertain everything is. Progress in atmospheric research is often given by stimulating case studies, nobody cares too much about uncertainties in the INP branch, when doing projects for the first time. But we extended the**

**explanations in Sect. 3.4., and provide more figures, to make is a bit more easy to follow.**

- Other places further information is required:

    - For the smoke section, could the authors please clarify why it is definitely smoke, and not a mix of pollution and smoke?
    **We learned from the Arctic aerosol papers that the upper troposphere and lower stratosphere is usually very clean. So, the regularly occurring pollution layers are typically at lower heights. Therefore, why should we then have mixtures of smoke and pollution in the UTLS, instead of pure smoke? We cannot clarify this point. If there was pollution over 'remote' Siberia then this pollution was lifted as well. But the optical fingerprints clearly point to pure smoke in the UTLS.**
    - P7L26: "*This fourth mechanism is responsible for the occurrence of ULTS wildfire smoke over the North Pole region in the MOSAiC winter half year.*" Please provide evidence or the reference for the deduction that this mechanism is the predominant responsible pathway for this transport during this entire time.
    **We removed this discussion. It is not needed, when focusing just on the winter half year.**
    - P7L30: "*Figure 4 presents the optical properties of the smoke layer as measured on 11 December 2019 (Fig. 3g and h). The smoke layer extended from 8 to more than 18 km height.*" I don't see evidence in Fig. 3 of the smoke layer going past ~13 km. In Fig. 4, the instrument detection limits have not been shown. It would be helpful to add those for the reader to better interpret these plots, and to see how high a detectable layer extended.
    **The message of Figure 3 is: there is aerosol in the UTLS height range. Afterwards (Fig 4) we show more details, base, top, backscatter, extinction, depol retrieval. All three backscatter profiles (355, 532, 1064) show the top of the layer! Why should we discuss detection limit problems when all three profiles show the same layer structures?**
    - P8L1: "*No other aerosol type (or cloud type) produces an inverse spectral behavior in terms of the particle lidar ratio*" Please describe the lidar ratio of aged pollution plumes, and say why that could not be a main contributor to the haze event determined here to be mostly made up of smoke particles.
    **In Ohneiser et al. (2021), there is a table with typical lidar ratio pairs (355, 532) for very different aerosol types, including for aged pollution. I think that is sufficient, and we do not need to repeat it here…. For pollution, LR355 is larger than LR532.**
    - P9L16: "the smoke layer was continuously present and probably homogeneously distributed over large areas of the Arctic." If CALIPSO could not observe the layer, what reason is there to believe that the layer was homogeneously distributed over large areas of the Arctic?
    **We changed this. We removed such statements. We simply do not know.**
    - P9L21: "*Note that we corrected our stratospheric smoke observations in Fig. 6 for PSC effects.*" The authors should describe how they corrected for these observations.
    **We explain that in Sect.3.1.**
    - P13L10: "*The new method was originally designed for pure liquid-water cloud observations but can be applied to mixed-phase clouds as long as backscattering by ice crystals is negligible compared to droplet backscattering in the droplet-*

*dominated cloud top layer. This condition holds here with ice crystal backscatter coefficients of 5-10 $Mm^{-1}$ $sr^{-1}$ in the virga and thus also in the cloud top layer and droplet backscatter coefficients of the order of 700 $Mm^{-1}$ $sr^{-1}$.*" Please provide more information on why this backscatter coefficient can be considered negligible.

**We explain that in a bit more detail in Sect.3.3.**

- P13L18: "*Later on, the updrafts became obviously stronger, and supersaturation levels exceeded 0.2%...*" Please discuss the evidence behind this statement.

**We rephrased the text a bit.**

- P13L19: "*With increasing CDNC the effective radius (characterizing the typical droplet size) decreased and vice versa for constant water vapor conditions*" Please discuss the evidence behind this statement.

**Again, we provide a bit more information.**

- Fig. 13: This graph and the text describing it on P14 is a bit difficult to understand. The temperature in Fig. 13a is said to be derived from radiosonde, so why is there only one T value shown, and does it only correspond to a height of 2.5 km? I see RH in Fig. 13b, but no temperature at all? The estimated CCNC, CDNC, INPC, and CCNC values are provided in a range. Is this range meant for a single altitude? Or across the whole figure? Or is it a point measurement range? Please specify where the values are relevant for each parameter, and why the values are only shown for that location/set of locations.

**Figure 13 is now Figure 14, and has now 4 panels. We discuss the new figure in large detail, keeping all the questions and comments of both reviewers into consideration in Sect. 3.3.**

- P15L24: "*During the 7-day travel in the Arctic the Pacific airmass mixed with the smoke above 7 km. These smoke particles then served as ice nuclei when cirrus formed after further lifting.*" Please discuss the evidence behind this statement.

**We provide a backward trajectory figure (Fig 16 in Sect 3.4).**

- Other comments:

  - In the text, when discussing CDNC values, please change from "CDNC" to "estimated CDNC" to reflect the appropriate uncertainty and to avoid confusing readers.

  **Sometimes we changed it, sometimes we stay with … retrieved. Estimation can be misinterpreted as … this is just speculation. And that is definitely wrong.**

  - P14L20: "*The good match between CCNC and CDNC (liquid-water cloud closure) and between INPC and ICNC (ice cloud closure, see numbers in Fig. 13a) during the early phase of the altocumulus development indicates that the aerosol particles controlled the cloud properties and thus had a strong influence of the evolution of the observed altocumulus cloud system as long as the humidity conditions were favorable. It should be emphasized that such a closure study with consistent findings is only possible if primary ice and droplet nucleation dominates and secondary ice formation, ice breakup processes, crystal-crystal collision and aggregation processes, as well as droplet collision and coagulation, and strong mixing and entrainment processes are absent.*"

  This statement seems overly confident and simplistic given the very high

uncertainties involved (only some of which are discussed here). Please rephrase to reflect a more accurate level of uncertainty. As an example, if I were writing this study I might say,

*"During early altocumulus development in the Figure 13 case study, the estimated CCNC values outside of the cloud are in a similar range to the estimated CDNC levels within the cloud, as are the estimated INPC and ICNC (Fig. 13a). Thus, our data suggest that the estimated cloud active particle levels could be high enough to control the case study cloud given favorable moisture conditions and in the absence of other processes that might influence CDNC and ICNC levels (e.g., secondary ice formation). This hypothesis would be in line with numerous other Arctic studies that have previously observed this phenomenon (e.g., Mautritsen et al. (2011)). However, higher resolution in-cloud microphysical data are required to verify this lidar-based hypothesis."*

**We thank the reviewer for his effort! We used this text. Great!**

- P15L19: *"To demonstrate that the observed wildfire smoke particle were able to control cirrus evolution and life time we present the results of a first MOSAiC case study here. The observation is from 6 December 2019 (Fig. 3c and d)."* Is this cloud even a cirrus cloud? The lidar signal extends down to near the surface at times, and the top is below 8 km altitude. What has been done to ensure that this is not actually a mixed phase cloud? The temperatures near the base of the cloud appear to be as high as -30C or so, from Fig. 14b, and liquid water can be present at such temperatures in the Arctic.
**The cloud is a classical cirrus (or better text-book-like cirrus with top structures of freshly nucleated crystals) and nice, coherent ice virga from 00:00 to 24:00 UTC on 6 Dec. No indication for any liquid phase. Ice nucleation always starts at cloud top (where the probability for ice nucleation is largest). The temperatures above 6 km were at all lower than -40C. On 7 Dec, (00:30 UTC, and later on…) there is a liquid layer around 2.5 km height, yes…. But that is another story…**
- P15L29: *"This part of the smoke layer (above 9.3 km) can be regarded as the main reservoir of INPs."* Why is it assumed that this aerosol layer is in contact with the ice cloud? To me it looks distinctly separate for most of the time.
**We improved a bit the discussion in Sect.3.4. Yes, in this case, we had the same impression….. maybe ice crystals formed on smoke and afterwards scavenged and removed the rest of the smoke particles… Disregarding this impression, we have no idea about the exact smoke conditions during ice nucleation process….. Therefore we show different surface area profiles from 2, 6, and 7 December…to give a reasonable range..**
- I like the introduction, it really gets the reader interested in the study.
**Thank you!**
- P.3, paragraph starting on L4: Here or elsewhere, you might also consider mentioning relevant Arctic high altitude smoke findings from Schill et al. (2020).
**We mention that in Sect. 1 and 3.4. Schill et al. (2020b).**
- P4.L22 *"… HSRL is of advantage during the summer half year (when Raman lidar observations are of limited use)"* Please specify why (and if relevant, which) Raman lidar observations are of limited use during the summer. Also, can't HSRL also be used during the winter? If so, for clarity please explain to the reader what additional

capabilities the Raman lidar provided that the HSRL could not.

**We leave out a discussion on HSRL here. We will use the data in future. But even now (June 2021) we did not see any results. We asked for data, but the HSRL data need to be quality checked.**

- P4L29: "*Di Biagio et al. (2018) were the first to run lidars (mounted on an ensemble of autonomous drifting buoys) in the Central Arctic, …*" The authors might consider mentioning that these data were collected on buoys.

**Was already mentioned, and is explained in Ohneiser et al. (2021) as well.**

- P6 " *'Co' and 'cross' denote the planes of polarization parallel and orthogonal to the plane of linear polarization of the transmitted laser pulses, respectively.*" This sentence should probably go in the previous section where the authors first mention the co- and cross terms.

**Improved! Sect.2.1 and 2.2**

- Fig. 3: It might be easier on the reader to just state: "range-corrected 1064 nm signal" and "linear depolarization ratio" above the columns in the figure instead of in the caption. To avoid confusion, the authors might also want to note in the caption and/or on P7L3 that that the y- and z-axis limits were varied between panels in order to highlight different features.

**All this is improved!**

- P8L5: "*The size distributions of the smoke particles were obtained from the Polly observation by applying the lidar inversion method to the layer-mean backscatter and extinction information (Veselovskii et al., 2012).*" This information would be better placed in the methods section.

  **Improved. The lidar product section (Sect 2.2) is much longer now.**

- Fig. 6a: This figure is not intuitive to me. Please tell readers what the height and base of the bars indicates (the top and bottom of a smoke layer?). Please tell them whether the colors are the relative fraction, or the dominant feature at that altitude (or something else). I am confused about the colors also because in the caption it says "The color in the bars provides information about the smoke particle concentration in terms of particle extinction coefficient at 532 nm." Please explain exactly what the particle extinction coefficient tells us about estimated smoke particle concentration. Please state in the figure and not just the caption that colors relate to particle extinction coefficient at 532 nm. Again, how do detection limits play into these bars? Please state whether the bars are only the observations above detection limits of the lidar. If the observations are below the detection limits, please either get rid of them, or clearly state why the data are still useful (I would guess they would not be). To avoid confusion, perhaps get rid of the height levels redundantly shown on the right side of the figure. Are the black dots the tropopause on that day? If so, an arrow from the word "tropopause" pointing to the dots might help clarify things. I know it was mentioned in the text, but can the authors mention in the caption as well in just a few words how the tropopause was determined?

**We took all the comments into account, and improved the figure (but leave in the height numbers in (a), right axis) as well as the discussion. We are a bit surprised. Intuitively , the length of the bar show the smoke layer from base to top… (what else?) and the color in the bar the strength of extinction coefficient (as in all these color plots you can find in literature. Ok, here we have just day by day**

observations, and want to show that by isolated bars..)

- Fig 6b and 6c captions: Please change "column mass concentration" and "vertical mean particle mass concentration" to "estimated column mass concentration" and "estimated vertical mean particle mass concentration" to indicate appropriate uncertainty

**Done!**

- P9L21: "*Note that we corrected our stratospheric smoke observations in Fig. 6 for PSC effects.*" This note should go in earlier with discussion of Fig. 6.

**Done!**

- P9L22: "*This type is made up of supercooled liquid ternary solutions that consist of $H_2SO_4$, $HNO_3$, and $H_2O$.*" Speculation on the chemistry may be beyond the scope of this paper. I suggest saying "likely consist" instead of "consist."

**Done!**

- P9: "*The temperature at PSC base height showed values of $-78°C$ and at the backscatter maximum the Polarstern radiosonde measured a temperature of $-86°C$.*" Wow, that is cold!

    **Yes!**

- P10L12: "*Height-resolved lidar observations of Arctic haze, prevailing during the late winter and early spring months, are rare (Di Pierro et al., 2013; Di Biagio et al., 2018).*" I suggest rephrasing this. The CALIPSO observations have taken observations in clear conditions over the entire Arctic since 2006, taking observations of plenty Arctic haze events.

**Improved! We agree…**

- P10L15: "*However, knowledge about the vertical layering structures of Arctic haze is still limited and mostly based on snapshot-like aircraft observations performed during field campaigns, preferably in spring.*" Again, I am not sure that is entirely true, given the extensive CALIPSO observations.

    **Improved as well…**

- P10L27: "*Type-Ia PSCs consist of nitric acid trihydrate (NAT) crystals and produce significant depolarization of backscattered laser light.*" Suggest rephrase to "…are thought to consist of …"

**Done!**

- P12L30: "*After nucleation, the ice crystals grew fast to sizes of 50–100 µm within minutes (Bailey and Hallett, 2012) and immediately started falling out of the altocumulus layer. The ice crystals partly evaporated on the way down, but partly reached the ground as precipitation.*" Please clarify here whether you are talking about findings from the Bailey and Hallet, 2012 study, or whether you are talking about results observed during MOSAiC.

**Improved, …this is a finding of Bailey and Hallet**

- Fig. 12: Please replace "CCN" with "Estimated CCN" in the figure and caption.

    **Done!**

- P15L19: This paragraph would benefit from a Figure showing the trajectories being discussed.

    **Such a figure is added**

- P16L34: "*As mentioned, ice nucleation occurs during updraft periods, more precisely when a certain (threshold) supersaturation level is exceeded.*" The authors may want to mention that ice nucleation also requires cold enough

temperatures.

**We enlarged the entire discussion (now with four figures, before we had just two…) and give equations and discussing all influencing effects in more detail.**

- Fig. 15: Again, please put estimated ahead of any parameters that were not directly measured and that include substantial assumptions in the caption.

**Partly improved…partly we prefer 'retrieved'…**

**Technical comments:**

- Title: "an introductory" should be changed to "an introduction." But maybe also consider making the title more succinct to make it more appealing to readers. Note: most readers will likely not know what UTLS is, suggest dropping it from title.

**Improved!**

- L5: "… aboard **the** Polarstern."

**Done!**

- Caption, Fig. 2 (and also corresponding text p. 4, L.9): "Figure 2. Polarstern drifting in the Arctic ice on 10 April 2020 (left panel) and measurement containers for in situ aerosol monitoring (the two first containers on the left side and the first container on the right side), and for remote sensing of aerosols and clouds (right panel). The OCEANET container of TROPOS is the third one on the left side." Could the authors please clarify whether they meant third one to the back, or the one in the front?

**Done! ..third to the back…**

- P6L12:"Hofer et al. (2020) exemplary shows…" Did the authors mean something like, "Hofer et al. (2020) is an example showing…"?

**Done! Hofer, for example, show…**

- P7L28 "poleward"

**Done!**

- P14L2: profile not profil.

**Done!**

---

## Author Comment (AC3) · 17 Jun 2021

Dear reviewer!

Thank you for taking the time to read the long manuscript and for preparing a long list of constructive suggestions and comments. Before we provide our answers, step by step, let us summarize the main changes.

- We changed the title!
- In Sect. 2, we added typical uncertainties in the lidar products in Table 1, and provide 4 paragraphs on validation efforts, as requested.
- Concerning the apparent contradiction of good lidar observation in an area with rather low aerosol content, we have the following answers: (1) There was complete darkness for 5 months, so almost 'unlimited' signal averaging was possible. (2) We do photon counting in all channels, no analog detection at all, so linear signal response over six orders of magnitude. (3) We have radiosonde temperature and pressure profiles every six hours, so accurate Rayleigh scattering properties and temperature profile information (in the extinction computations) is available. Consequently, the small aerosol effects could be accurately separated from the Rayleigh backscatter and extinction properties.
- We re-analyzed all data shown in the figures.
- The tropopause computations contained a bug, is now correct.
- We included new figures with backward trajectories (Fig, 12, Fig, 16).
- Fig. 14 (mixed-phase cloud closure study) now includes four radiosonde observations.
- The cirrus closure study is enlarged to make it more understandable. Now, we have four figures in Sect. 3.4 instead of two (submitted version). We cannot leave this study out. It is a highlight because, for the first time, the impact of aged smoke (organic material) on cirrus formation is discussed based on real-world observations.

**In the revised version of the manuscript we indicate the essential changes IN BOLD. Therefore, not every small change is indicated.**

**Step by step reply: our answers in blue**

The paper "UTLS wildfire smoke over the North Pole region, Arctic haze, and aerosol-cloud interaction during MOSAiC 2019/20: An introductory" presents and discusses remote sensing observations obtained during the MOSAiC campaign. The unique dataset provides new opportunities to explore aerosol-cloud interactions at the North Pole. Persistent smoke layers originating from large scale fires are discussed and analyzed in terms of their properties as well as the ability to act as Cloud Condensation Nuclei and Ice Nuclei (CCN and IN). Observations of the Arctic haze during the campaign are presented and discussed also, aiming to enrich the current observational dataset with winter time measurements.

The study falls within the scope of ACP. The manuscript is well-written and structured, the presentation is clear, the language is fluent and the quality of the figures are high. In order to help improve the manuscript, I would kindly suggest the authors to take into account the following comments.

**General comment**:
Please provide the typical uncertainties of the lidar-derived aerosol and cloud microphysical properties discussed, originating from the conversion assumptions and the uncertainties of the optical properties derived from the lidar measurements. Additionally, please discuss the effect of the low aerosol

concentrations presented here, on these retrievals. Please provide information on independent validation studies of the lidar-derived microphysical products (e.g. using in-situ measurements).

**We show typical uncertainties in Table 1 and we have four paragraphs on validation efforts. We leave out to discuss lidar retrieval aspects at clean conditions. We think, the discussion of our findings show how clean the atmosphere was. And regarding winter, our impression was that the polar regions are no longer very clean. The uncertainty bars in the figures with lidar results show that we were able to measure the aerosol. But of course, long signal averaging times of partly 24 hours were needed.**

**Specific comments:**

**Page 2, lines 29-32:** "The MOSAiC lidar observations (together with the radar observations of the ARM mobile facility) allow us, for the first time, to investigate the role of smoke in ice formation processes". The authors should specify the Arctic region in the sentence, since there have been several studies exploring the potential of smoke particles to act as IN (.i.e. Peters et al., 2009; Prenni et al., 2012; Levin et al., 2016; Schill et al., 2020). Or specify if the statement is meant for combined lidar and radar observations in the Arctic.

**There have been no studies on aged smoke (organic substances controlling INP potential), yet. All the publications (including the ones you mention) deal with fresh smoke and focus on soot particles. Aged smoke particles are rather different regarding their INP potential.**

**Page 3, lines 1-3:** "A unique opportunity is thus given to explore to what extent the wildfire smoke particles, providing a significantly enhanced number of sites for heterogeneous chemical processes (chlorine and bromine activation), contributed to the strong ozone depletion": Since the article provides hints on the role of smoke particles on ozone depletion, rather than "findings", please rephrase accordingly.

**We briefly mention the ozone aspect in Sect.1. The potential smoke impact on ozone depletion is discussed by Ohneiser et al (2021).**

**Page 3 lines 4-30:** "The article is organized as follows… Sect. 4 finally provides some concluding remarks": This section is hard to follow, since it mixes the scientific objectives of the study with the proposed methodology/techniques and the article layout. Please divide this part in three paragraphs, with the first containing the scientific objectives of the study (e.g. "Organic aerosol particles are ubiquitous in the atmosphere around the world and there is an urgent need to investigate..."), the second one containing the proposed methodology/techniques and the third containing the structure of the article with very brief descriptions for its section (e.g. "... In Sect. 3.2, we present two cases of Artic haze observations performed in February and March 2020").

**We rearranged this part a bit.**

**Page 3, lines 21-22:** "Recently introduced new remote sensing analysis concepts (closure studies) (Ansmann et al., 2019) are applied for the first time to Arctic clouds". Please also include the work of Marinou et al., (2019).

**Done!**

**Page 5, lines 9-10:** "which permits accurate aerosol and cloud profiling from about 800 m to 30-40 km height". Please specify separately the information on aerosol and cloud detection ranges and provide information on the accuracy of the aerosol and cloud retrievals for different altitude ranges. Which are the typical signal-to-noise (SNR) values of this lidar for aerosol layers at 10, 15 and 20km a.s.l? What are the uncertainties of the lidar-derived properties at these altitudes?

**We do not think that this is a good idea. This paper shall attract readers who are interested in atmospheric science with focus on aerosol and clouds. We think that we do not need such detailed lidar information in this MOSAiC paper.**

**Page 5, lines 16-19:** "This technique enables us to determine multiple scattering by droplets in liquid-water dominated cloud layers and to determine from this multiple scat-tering information cloud microphysical properties (e.g. effective droplet size and cloud extinction coefficient) (Jimenez et al.,2020a). The method is based on depolarization ratio (ratio of cross-to-co-polarized backscatter coefficient) observations at the two FOVs". This part is more appropriate for section 2.2 where the rest of the lidar products and retrievals are presented. I suggest to move it before **page 6, line 15-16**: "Details of the retrieval of microphysical properties of liquid-water cloud layers can be found in Jimenez et al. (2020a, b)".
**We moved the description to Sect. 2.1 and 2.2**

**Page 5, lines 26-27:** "we use the preliminary radiosonde products that were directly available during the expedition". Explain the "preliminary" definition in the radiosondes used, and why you used these instead of the consolidated radiosonde products.
**Meanwhile, we got the quality checked ones (Maturilli et al., 2021).**

**Page 6, lines 9-16:** "The retrieval of aerosol microphysical properties such as particle volume, mass, and surface area concentration and estimates of cloud-relevant properties (aerosol-type-dependent cloud condensation nuclei, CCN, and ice-nucleating particles, INPs) is performed by means of the POLIPHON (Polarization Lidar Photometer Networking) approach (Mamouri and Ansmann, 2016,2017; Ansmann et al., 2019, 2020). Hofer et al. (2020) exemplary shows the full set of POLIPHON aerosol products in the cases of an 18-month Polly campaign in Dushanbe, Tajikistan, for central Asian aerosol. Alternatively to the POLIPHON method,we used the multiwavelength lidar inversion technique (Müller et al., 1999, 2014; Veselovskii et al., 2002, 2012) to derive microphysical properties of aerosols including the particle size distribution for detected pronounced aerosol layers. Details of the retrieval of microphysical properties of liquid-water cloud layers can be found in Jimenez et al. (2020a, b)". Please provide a short description of the assumptions used for the aerosol and cloud microphysical properties retrieved from the lidar products. Please comment on independent validation studies for these products (e.g. with in-situ measurements as in Marinou et al. (2019) study). Please quantify and discuss the uncertainties of the aerosol and clouds microphysical retrievals for the observations presented on this study.
**We try to find a balance between explaining the methods and referring to the literature if it becomes too complicated. We discuss in more detail the uncertainties in Sect. 3.1 to 3.4. We have Table 1 in addition with uncertainty information. We find that both reviewers are a bit too critical concerning the uncertainties and validation efforts. There are so many 'questionable' papers based on satellite remote sensing and airborne in situ observations in the literature, and there is no big deal with uncertainties, validations, and to emphasis 'Estimated products'.**

**Page 7, line 2:** "The measured linear depolarization ratio in the right panels of Fig. 3 allows us to precisely distinguish cirrus from layered mixed-phase clouds as explained above". Above you mention in **page 3, line 15:** "We start with a case of a shallow mixed-phase cloud consisting of a liquid-water layer on top of the ice virga zone" and in **page 6, line 6:** "...in the case of clouds, liquid-droplet layers show PLDR≈0 at layer base where light depolarizing multiple scattering is low, and PLDR of 0.4-0.6 in, e.g., cirrus layers", but you haven't explained how you distinguish cirrus from layered mixed-phase clouds in depolarization measurements above. So I suggest to skip "as explained above '' or explain it.

**We rearranged the text and better introduce the particle linear depolarization ratio in Sect. 2.1 and 2.2.**

**Page 7, lines 22-25:** "The light-absorption-related lifting occurs during the spread of the smoke over the respective hemisphere and continues as long as the smoke layers are optically dense enough (aerosol optical thicknessAOT>1-2 at 500 nm) with the consequence that the smoke reaches, e.g., the Central Arctic at heights up to 5-10 km above the tropopause". Please provide a reference for these AOT required conditions.

**We provide a reference (Boers et al., 2010) and also state that the self-lifting aspect is discussed in detail in Ohneiser et al. (2021).**

**Page 7, line 33:** "The 532 nm lidar ratio is much larger than the 355 nm lidar ratio. ": please quantify how larger it was.

**More than 20sr. This is now mentioned in Sect.3.1.**

**Page 8, line 5:** "These specific optical properties are linked to the narrow size distribution of absorbing smoke particles which form a well-defined accumulation mode as shown in Fig. 5". Please provide references to support this claim. Moreover, discuss the role of the shape and refractive index of the smoke particles in defining the unique optical properties measured.

**We rephrased this part of the discussion. All in all, we keep the discussion short. We avoid a long discussion on particle shape because the depolarization ratio was low and indicated spherical particles. More details are given in the accompanying paper of Ohneiser et al. (2021).**

**Page 8, line 15:** "With increasing age the core structure obviously collapses, gets compact, and the particles become more and more spherical with time (Baars et al., 2019)". Please revise obviously as probably.

**We changed the text….**

**Page 8, line 27:** "Downward mixing and transport into the lower troposphere had no impact on the UTLS AOT as well". Please explain this statement in more detail. How can this be supported, when the AODs observed are decreasing with time from 0.12 to <0.03 during the time period discussed?

**We changed the text. The strong vortex controlled the weather pattern and isolated to some extent the air mass over the North Pole. That is all what we conclude in the revised version.**

**Page 9, line 7:** " But this smoke layer had no clear boundaries, at least no clear upper boundary (see Figure 4a)" and **page 8, line 30**: "The layer-mean 532 nm smoke extinction coefficients in Fig. 6c (obtained from the ratio of AOT divided by the layer geometrical depth in Fig. 6a)". Please include a comment on the effect of the unclear layer's boundary to the lidar retrievals presented in this work (e.g. the effect to the AOD and layer top heights).

**We rephrased that. For us, the top was well defined, but for CALIPSO obviously not. We use the 1064 backscatter ratio of 1.1 as threshold to define the top. This is written in Sect. 3.1.**

**Page 9, lines 1-10:** "It is noteworthy to mention that the CALIPSO data base...aerosol observations and corroborate our hypothesis". Please revise taking into consideration the Interactive comment of Jayanta Kar on the detection of the smoke layers from CALIPSO.

**Yes! We removed the two paragraphs with statements to CALIPSO observations.**

**Page 9, line 21:** "Note that we corrected our stratospheric smoke observations in Fig. 6 for PSC effects". Please explain how.

**Done now, in Sect.3.1.**

**Page 9, lines 33-34:** "According to Vaughan et al. (2020), the volcanic aerosol layer consisting of sulfuric-acid-containing water droplets (75% H2SO4, 25% H2O) formed above the tropopause with maximum heights reaching 21 km". In Vaughan et al. 2020 paper they reported that the volcano "...send a plume of ash and sulphur dioxide into the stratosphere...", "the ash and sulphur dioxide plume initially moved westward before being entrained in a cyclonic circulation over the North Pacific" and "During the latter half of June and in early July, pyroconvection over Canada injected layers of smoke and ice clouds into the lower stratosphere (similar to the case described by Vaughan et al. (2018)), making it difficult to distinguish the progression of volcanic ash remnants using the CALIOP profiles". Why do you not mention the ash contribution on these layers? As the smoke layers from these fires arrived in the MOSAiC altitudes and are discussed herein, why do you exclude possible mixture with the volcanic ash?

**Usually, ash particles fall out quickly. The eruption was in June, so that the ash was probably removed in July and August 2019. At least, the depolarization ratios do not indicate any occurrence of ash. Regarding stratospheric smoke, Kloss et al. (2021) mentioned that pyroCb-related smoke layers were of minor importance in 2019.**

**Page 9, line 34 - page 10, line 2:** "From the 355 nm Raman lidar observations at Capel Dewi Atmospheric Observatory, United Kingdom (52.4∘N, 4.1∘W) it can be concluded that the 532 nm AOT was about 0.03 over UK in August 2019, of the order of 0.01 in December, and of around 0.005 during the first months of 2020, respectively". Please provide a reference for these AOTs.

**We rephrased all this. We went deeply into the literature (also regarding the Sarychev volcanic eruption in June 2009). Raikoke and Sarychev are neighbor volcanoes and both erupted in June (2009 vs 2019), both injected almost the same amount of SO2, and the max AOT of Sarychev was 0.02 at 500nm, and the max AOT of Raikoke was expected to be 0.025. Later on, the 532 nm AOT should be 0.01 to 0.005 over the Arctic in autumn and winter, respectively, as it was the case after Sarychev eruption. All this is discussed in large detail in the accompanying paper of Ohneiser et al. (2021).**

**Page 10, lines 10-14:** "The original and primary goal of the shipborne MOSAiC lidar measurements was to provide, for the first time, a seasonally and height-resolved characterization of tropospheric aerosols and clouds for the North Pole region. Especially the coverage of the winter half year can be regarded as a valuable new contribution to Arctic aerosol research. Height-resolved lidar observations of Arctic haze, prevailing during the late winter and early spring months, are rare (Di Pierro et al., 2013; Di Biagio et al., 2018)". As there are plenty of CALIPSO overpasses in the Arctic region, please rephrase this part including for example the specific latitudes of the campaign.

**Done! Now, we even have the study of aerosol profiles by Yang et al. (2021), i.e., CALIPSO-based climatological observations from June 2006 to December 2019 (see the results in Ohneiser et al., 2021).**

**Page 10, line 22:** "The measurements are representative for many days during the winter months". Please quantify the days arctic haze was observed in the period of the experiment.

**We rephrased that. Arctic haze was present almost every day.**

**Page 10, lines 23-24:** "The most striking feature in both figures is that aerosol layers were detected everywhere up to the tropopause, and because of the smoke layer even from 8 to almost 20 km

height". This is not visible in these figures which are up to 15 and 16 km. Please state if you refer to fig. 6, which shows smoke up to 17km during March and April, and smoke up to 20km in February.

**It is visible in all these figures! Clean would mean, blue colors from 5 to 16 km height. And this is not the case.**

**Page 10, lines 27-28:** "Type-Ia PSCs consist of nitric acid trihydrate (NAT) crystals and produce significant depolarization of backscattered laser light". Please rephrase to "...are thought to consist of ...".

**OK!**

**Page 11, lines 6-7:** "The Ångström exponent for the extinction coefficient was around 1.7 in the lofted layer above 3 km height. The lidar ratios were high with values close to 100 sr in the lofted layer on 4 March". Please provide the uncertainties or standard deviation of these values.

**Done! 15 sr**

**Page 11, lines 6-7:** "The 532 nm AOT was 0.024 (4 February, for the lowest307 km height) and 0.022 (4 March, for the lowest 5 km)...". Please provide the uncertainties/errors of these values.

**Done! 10-20%**

**Page 12, lines 29-30:** "Favorable conditions with cloud top temperatures around −28.5C at 2.6 km height (at 03:00 UTC) were given for heterogeneous ice formation via immersion freezing, i.e., ice nucleation on INPs immersed in the water droplets". Please provide relevant reference for the "favorable conditions".

**Done! Kanji et al. (2017)**

**Page 12, lines 30-32:** "After nucleation, the ice crystals grew fast to sizes of 50–100µm within minutes (Bailey and Hallett, 2012) and immediately started falling out of the altocumulus layer". Please clarify whether you refer to findings from Bailey and Hallet (2012) or to result from this specific case. If the second appy, please provide a short description on the measurements/methods used for these findings.

**Done! Bailey and Hallet is meant.**

**Page 13, lines 3-4:** "As discussed below in detail, there were always 20-200 droplets per cm3 in the altocumulus top layer, but only 0.1 to 1 ice crystals per liter". Please rephrase so as to be clear that these concentrations are estimated and not measured.

**Done!**

**Page 13, lines 12-14:** "This condition holds here with ice crystal backscatter coefficients of 5-10 Mm−1sr−1 in the virga and thus also in the cloud top layer and droplet backscatter coefficients of the order of 700 Mm−1sr−1". Please include the information that these values are not shown in the study.

**Done!**

**Page 13, line 15:** "As can be seen in seen Fig. 11, the CDNC values were about 20 cm−3 in the beginning and around 100 cm−3 later on". Please rephrase to "estimated values" or "retrieved values"

**Done!**

**Page 13, lines 16-18:** "Obviously, updraft velocity was weak and correspondingly water super saturation levels were below 0.2% so that fewer particles were activated to become cloud droplets as predicted (CCNC values in Fig. 12). Later on, the updrafts became obviously stronger, and

supersaturation levels exceeded 0.2% so that more CCN nucleated cloud droplets as predicted by the retrievedCCNC values". Please rephrase or justify "obviously".

**We rephrased a bit.**

**Page 13, line 20:** "The cloud extinction coefficient showed typical values from 10-20 km−1most of the time in the droplet-dominated cloud layer". The plot shows cloud extinction coefficient values <10 km^-1 from 7:45 until the end of the cloud (~3hrs) and values between 10-20 km^-1 for ~2.5hrs. Please rephrase accordingly.

**We corrected that.**

**Page 13, line 30:** "Soot and mineral dust particles are good candidates to serve as INPs". Please provide relevant references (i.e. Sassen and Khvorostyanov, 2008; Boose et al., 2016; 2019).

**We removed this sentence. Dust is the only good candidate because Kanji et al. (2020) and Schill et al. (2020a) found out that soot in a bad immersion freezing INP (at temperatures > -30C).**

**Page 13, lines 31-33:** "Dust is left as potential INP. Our polarization lidar observations indicated the presence of a dust fraction of 3-10% according to the slightly enhanced particle depolarization ratio (not shown) above 2 km height..." Earlier, in page 9, you mentioned the influence of aerosol particles from "a strong eruption of the Raikoke volcano in the Kuril Islands…". Why do you exclude the possibility of the presence of ash non-spherical particles? Please discuss the effect of possible ash particle presence on the INP retrievals you provide.

**No! There was no ash!**
**We re-checked the depolarization ratio computations. Dust fraction is now assumed to be 5%.**

**Page 14, lines 1-2:** "We used the INP parameterization scheme of DeMott et al. (2010) to estimate the dust INP concentration for immersion freezing. Here, the particle number concentration n250 of dust particles with diameters >500 nm is an input parameter and obtained from the respective lidar observation of the extinction coefficient in Fig. 12 and by assuming a dust fraction of 3-10% in the conversion of the extinction profile into the n250 profil". Please discuss why you use the INP parameterization scheme of DeMott et al. (2010), which was developed with minimum presence of dust particles in the analysed samples, to convert the lidar derived dust n250 concentrations to INP, instead of the dust-tailored INP parameterization scheme of DeMott et al. (2015). Can the authors comment on the effect of the dust-tailored parameterization to the lidar-derived INPC in the study?

**We used DeMott (2015). We gave the wrong reference.**

**Page 14, line 17:** "until a dry air mass approached, leading to a strong decrease in relative humidity and dissolution of the stratiform cloud deck": Please support this discussion with updraft and wind data, if available in MOSAiC. Moreover, please provide a colorbar with more points so that it is easier for the reader to understand which values you are referring to.

**Updraft speed was not measured during MOSAiC, although 4 Halo Doppler lidars were aboard Polarstern. We think the humidity figure in panel (d) clearly shows the dry air mass. And the 17 UTC radiosonde in panel( b) shows that as well.**

**Page 14, line 19:** "ice crystals could partly reached the ground as precipitation". for convenience to the readers, you could refer to figure 3f, where this is evident.

**We mention that now when discussing Fig. 14. We do not know if we should guide the reader to check again Fig 3 f. The humidity field in Fig. 14 indirectly shows that virga reached the ground. On the other hand, the 1064 nm signal cannot be used to show that because it is biased in the near range because of too large backscatter… we mention that in the figure caption.**

**Page 14 line 20:** "The good match between CCNC and CDNC (liquid-water cloud closure) and between INPC and ICNC". Please rephrace as "the good match between the estimated CCNC.." or "the good match between the retrieved CCNC..".

**Guided by both reviewers we offer now a much more sensitive discussion. We often mention that we deal with estimations and retrieval products and that the uncertainties are large etc.**

**Page 14, lines 32-33:** "Organic aerosol (OA, the main aerosol component in wildfire aerosol) is besides dust and marine particles ubiquitous in the atmosphere". Please comment on the presence of ash particles, as mentioned in previous comments.

**There was no ash! We do not like to increase the complexity of the discussion if it not necessary.**

**Page 15 lines 20-25:** "HYSPLIT backward trajectory analysis ... During the7-day travel in the Arctic the Pacific airmass mixed with the smoke above 7 km". Consider including a figure showing the trajectories discussed, even in an appendix.
**Done!**

**Page 16, lines 13-15:** "Ice supersaturation conditions are usually given or produced during updrafts (e.g., during the ascending period of a gravity wave) that could, in principle, be detected and measured with the AMF-1 Doppler radar". Is this information available for the case discussed here?
**No! The radar saw only the lower part of the strong virga.**

**Page 17, line 2:** "600 s may represent here a typical time period of the lifting phase of a gravity wave". Please provide a reference on this, or discuss if these updrafts were actually observed during MOSAiC.
**This is not observed, and a reference is not given because that is concluded from our own observations during the last 10 years.**

**Page 17, lines 2-5 and lines 13-16:** "As can be seen the nICE and nINP,I values (blue and red bars) are in the same range of values which suggests that organic particles may be able to control the evolution of the cirrus layer via the immersion freezing mode ... The impact of deposition INP nINP,D (cyan and orange bars) is comparably weak in this example… However, the successful closure, indicated by a reasonable match between nINP,I and nICE, indicates that the wildfire smoke was able to trigger cirrus formation (before homogeneous freezing can take place on stratospheric background or even liquid smoke particles) and control of the further evolution of the ice cloud system". The authors should consider the possibility that the discussed cloud was formed in an aerosol reacher environment. As they show in fig. 6, all the dust plumes observed in the period between 1/10 to 30/11, and for altitudes up to 11 km,  had extinction coefficient values >10 Mm^-1. Also, indicatively, in Fig. 7 and Fig. 10, the aerosol profiles show that the extinction coefficient values in the middle of the layers were respectively 5 and 8 times higher than the values in the edges of the layers. It would be good if the authors take into consideration that the n_INP,D abundance could have been significantly higher in the time of the formation of the cloud, and discuss how this affect their conclusion that the immersion freezing process with uplifting support from a gravity wave is the main driver on the n_ice observed in this case.
**We rearranged the entire Sect. 3.4 and tried to consider the comments of the reviewers.**

**Figure 5:** The size distribution retrievals in Fig. 5 need further support with a) the provision of the corresponding retrieval errors (as these are provided in e.g. Veselovskii et al. (2012)) and b) the comparison with other studies for stratospheric smoke. The effect of the (quite) low AODs on the retrieval errors should be also discussed.

**We discuss the method, we give uncertainties, we mention the Arctic haze study of Mueller 2004. In that publication the Arctic haze size distribution measured in situ was compared to the lidar-derived size distribution of Arctic haze. Good agreement was found. The size distributions are so clear in the**

**case of wildfire smoke. There is only one mode, no complex structures at all as many aircraft studies show. We do not see any reason to stress again how uncertain this retrieval is.**

**Figure 6:** Please provide the products uncertainties in figures 6b and 6c. Also, I would suggest changing legend "col. mass" in fig. 6b to "col. mass conc.", and legend "Mass conc" to "Vert. mean. mass conc." for completeness.
**We improve this!**

**Figure 7:** "The 532 nm backscatter ratio (total-to-Rayleigh backscatter) peaks at 2.43". This parameter is not presented in the plot. Please include it or this give more information on this sentence to be in the context of the figure.
**We give more details in the caption.**

**Figure 7:** ". PSC optical thickness was 0.0125 at 532 nm(computed from backscatter values multiplied by a lidar ratio of 50 sr". Please explain in the relevant section (in page 9) why you chose this LR value (e.g. Ohneiser et al., (2021) paper).
**The PSC LR of 50 is taken from the CALIPSO LR input data base.**

**Figure 8:** The VDR quicklooks of these cases would be of interest to the reader. Additionally, the addition of the backscatter profiles from 0-15km, which can support the argument of the authors that "There were no regions with a negligible aerosol content" in **page 10, line 24.**
**The values are close to zero. It makes no sense to show a color plot.**

**Figure 13:** "The range-corrected 1064 nm signal in (a) is biased by an detector overload in the nearest height range to the lidar". Include the height you are referring to.

**We provide more information in Sect. 3.3.**

**Figure 13:** In Fig. 13b (as well as in Fig. 3f) the lidar signal above the cloud seems to be totally attenuated. Please provide additionally the collocated radar measurements to show the extent of the whole cloud and support the cloud top provided from the lidar.
**Former Figure 13 is now Figure14. The radiosonde profiles in (a) and (c) corroborate that the clouds were shallow.**

**Figure 15:** To simplify the plot and make it more clear for the reader, the authors could consider moving the n_ice numbers to fig. 15c, where the profiles of these values are plotted.

**We rearranged the figures…**

**Figure 15:** "derived ranges of INP number concentrations nINP,I (for immersion freezing, blue and red, indicating the respective cirrus layers in (a)". Please revise this part to state clearly these n_INP concentrations from which profiles/days are derived.

**We rephrased the text, we have now three lidar figures instead of two, … everything is now more simple and better to understand, we think…**

**Technical corrections:**
Page 1, line 2-5: Long sentence. Consider splitting it to two.
**Done!**

Page 5, line 22: "procedures, and to": skip and
**Done!**

Page 6, line 3-8: "The linear depolarization ratio is defined as the cross-polarized-to-co-polarized backscatter ratio and allows us to sensitively distinguish spherical particles showing particle linear depolarization ratios (PLDR) close to zero from non-spherical aerosol particles showing PLDR values of typically 0.1-0.3. In the case of clouds, liquid-droplet layers show PLDR≈0 at layer base where light depolarizing multiple scattering is low, and PLDR of 0.4-0.6 in, e.g., cirrus layers. "Co" and "cross" denote the planes of polarization parallel and orthogonal to the plane of linear polarization of the transmitted laser pulses, respectively". It will be better read if the "co" and "cross" definitions are closer to the parameter "cross-polarized-to-co-polarized backscatter ratio".
**We rearranged it accordingly**

Page 7, line 11: "In summer, warm, moist and polluted air massed..". Typo: masses.
**OK!**

Page 7, line 31: "The internal vertical structures were rather smooth and indicate an aged smoke layer": change "and" to "which".
**Done!**

Page 8, line 16: "well established and obviously prohibited any mixing". Please delete "obviously".

**Done!**

Page  13, line 13: "As can be seen in seen Fig. 11". seen duplicate.

**Improved!**

Figure 13: "... (a) is biased by an detector overload". Typo: a.
**Improved!**

Page 14, lines 32-33: "...(OA, the main aersol component..). Typo: aerosol .
**Improved!**

Page 18, line 1: "As an outlook". The authors could consider to revise this to "future work".

**Improved!**

---

## Author Response (AR2)

Dear Editor (Dear Geraint)!

We improved the manuscript and considered (almost) all recommendations of the reviewers. The changes are indicated in bold in the revised version.

We want to mention that we included a sentence (in the ABSTRACT) concerning the contribution of the Raikoke sulfate aerosol to the 532 nm AOT over the Arctic … and added some text to Raikoke in Sect.3.1 as well.

Furthermore, we added a sentence in the Conclusions (at the end) that a super site for remote sensing and in situ observation (balloon, UAVs, aircraft) in the high Arctic for year-around aerosol and cloud observations in the free troposphere up to the lower stratosphere would be desirable…

Our step-by-step answers in blue!

**Reviewer #1:**

The authors did a very good job responding to my comments and answering my questions. I like the new Table 1.

**Thank You!**

Fig. 14: As a very minor recommendation for the authors to consider if they wish: they could mention that the paper's INP concentrations of 0.1 to 0.5 L-1 are in line with previous Arctic in situ observations at similar temperatures of -28.5 oC, especially given their uncertainty ranges in Table 1. The observations that I am aware of range from 0.001 to ~2.5 L-1 between -25 and -28.5 oC (Creamean et al., 2019; Hartmann et al., 2020; Mason et al., 2016; Wex et al., 2019).

Creamean, J. M., Cross, J. N., Pickart, R., McRaven, L., Lin, P., Pacini, A., Hanlon, R., Schmale, D. G., Ceniceros, J., Aydell, T., Colombi, N., Bolger, E., and DeMott, P. J.: Ice Nucleating Particles Carried From Below a Phytoplankton Bloom to the Arctic Atmosphere, 46, 8572–8581, https://doi.org/10.1029/2019GL083039, 2019.

Hartmann, M., Adachi, K., Eppers, O., Haas, C., Herber, A., Holzinger, R., Hünerbein, A., Jäkel, E., Jentzsch, C., Pinxteren, M. van, Wex, H., Willmes, S., and Stratmann, F.: Wintertime Airborne Measurements of Ice Nucleating Particles in the High Arctic: A Hint to a Marine, Biogenic Source for Ice Nucleating Particles, 47, e2020GL087770, https://doi.org/10.1029/2020GL087770, 2020.

Mason, R. H., Si, M., Chou, C., Irish, V. E., Dickie, R., Elizondo, P., Wong, R., Brintnell, M., Elsasser, M., Lassar, W. M., Pierce, K. M., Leaitch, W. R., MacDonald, A. M., Platt, A., Toom-Sauntry, D., Sarda-Estève, R., Schiller, C. L., Suski, K. J., Hill, T. C. J., Abbatt, J. P. D., Huffman, J. A., DeMott, P. J., and Bertram, A. K.: Size-resolved measurements of ice-nucleating particles at six locations in North America and one in Europe, 16, 1637–1651, https://doi.org/10.5194/acp-16-1637-2016, 2016.

Wex, H., Huang, L., Zhang, W., Hung, H., Traversi, R., Becagli, S., Sheesley, R. J., Moffett, C. E., Barrett, T. E., Bossi, R., Skov, H., Hünerbein, A., Lubitz, J., Löffler, M., Linke, O., Hartmann, M., Herenz, P., and Stratmann, F.: Annual variability of ice-nucleating particle concentrations at different Arctic locations, 19, 5293–5311, https://doi.org/10.5194/acp-19-5293-2019, 2019.

**We forgot this point last time…. So, now this statement is given in Sect. 3.3 (in the text part) and all four references are included in the reference list.**

Technical comments:

P6L23: I am not sure a manuscript in preparation should be cited here?

**Is replaced now by the Ohneiser et al., ACPD, 2021 version**

P3L31: "to what extent"

**Improved**

P5L15: "above the ground" (or maybe more accurately, above the surface?)

**Improved ..above the surface**

P14L27: "The goal was to …"

**Improved**

**Reviewer #2**

The authors did a great effort to improve the manuscript, and included several new parts and methods. The manuscript has changed a lot.

**Thank You!**

Please find below few remaining minor comments to the new additions of the manuscript. It would be optimum for the readers if the authors can revise the text based on these comments.

Nomenclature of variables in the figures: I suggest using the same nomenclature in the figures for the extinction coefficient variable throughout the paper. Specifically, Figure 4,10,13 have "Extinction cf" while Figure 15,17,18 "σ".

**We replaced sigma by Extinction cf. in Figs. 15a, 17a, and 18a.**

Lidar ratio symbol (page 9, line 34): Is there a reason why for the extinction-to-backscatter ratio the symbol L is chosen in this paper, instead of the symbol S or the abbreviation LR which are used in the literature? Consider to revise or include a small comment on the paper for this selection.

**We did not use S for lidar ratio because S stands for super saturation in cloud research papers. It is like T for temperature. We should not violate that. I think we do not need to state that explicitly.**

Page 18, line 31: " Lifting phases of gravity waves can be as long as 20 minutes (1200 s) as our Doppler lidar and radar observations conducted in several field campaigns during the last 10 years indicate". If available, consider including some references for this.

**We found the nice paper of Kalesse and Kollias (2017)  on vertical wind statistics over Oklahoma (13 years of ARM radar observations in cirrus), and they support our assumption (600s). They found that gravity waves have typical durations of 19 +/- 7 min  over Oklahoma in winter! Gravity waves are different in the tropics and for the rest (midlatitudes and high latitudes). Then, the updraft period of the full gravity wave is about 10 min = 600s. We do not give all these details, we just provide the reference in Sect. 3.4.**

**By the way, we provide the following statement in the beginning of Sec 3.4 now: According to a study of Barahona et al. (2017), Arctic ice clouds tend to form almost exclusively by heterogeneous ice nucleation with a contribution of only 10% by homogeneous freezing. We give this sentence, because it is so much in line with our study and findings.**

Concerning the uncertainties introduced and discussed: As the uncertainties are different for different mixing states, the authors should consider adding in Table 1 a comment with the information on the mixing state these uncertainties are representative of. See specific description of the inconsistencies below.

**We mention this point in Sect. 2.2 now. A short statement in the Table would be confusing … because some explanation would be necessary, and that would lead to a large caption text.**

Page 7, line 31: In the referenced paper of Haarig et al. 2019, Table 1, the uncertainty of the $n_{50}$ is a Factor of 2 and for $n_{250}$ is 30%, while in the Table 1 of this work the uncertainty provided for $n_{50}$ is 50%, and for $n_{250}$ is <=25%. Additionally, from the POLIPHON method one can easily calculate that the type-separated extinction coefficient uncertainties, on a layer where mixtures of different aerosol types prevail, can exceed 30% for the non-dominant aerosol particles. For example, for an aerosol layer with particle depolarization ratio of 10±1% (dust mixtures), using the POLIPHON method with $\delta_{dust}$ = 30±3% και $\delta_{nondust}$ = 5±2% the weight calculated is 0.236±0.052%. For a layer with $\beta p_{total}$=1±0.1Mm^(-1)sr^(-1) (uncertainty defined in this work), the pure dust backscatter uncertainty is propagated as: $\beta p_{dust}$= 0.236±0.057 Mm^(-1)sr^(-1) and for the non dust: $\beta p_{nondust}$= 0.764±0.092 Mm^(-1)sr^(-1). With $LR_{dust}$ = 45±11 Sr, $LR_{smoke}$ = 85±21 Sr (<25% uncertainty as defined in this paper): $ap_{dust}$= 10.6±3.6 Mm^(-1) => 34% error on the dust extinction coefficient, and $ap_{smoke}$ = 64.9±17.9 Mm^(-1) => 28% error on the smoke extinction coefficient. POLIPHON calculation from this point for the dust mass concentration gives: $M_d$ = 17.7±6.3 => 36% uncertainty. This is an example of a case where the uncertainties provided in Table 1 are lower than the error propagated uncertainties from the measurements and method. That is why a comment on Table 1 on the mixing state representative for these uncertainties would be very useful for the reader.

The effect of the mixing state on the retrievals is partially mentioned in page 7, line 35 for the CCNC retrievals: "Comparisons with airborne in situ measurements showed that CCNC can be obtained with an uncertainty of about 30% (inversion of multiwavelength data) to 50% (conversion of the 532 nm extinction coefficient) when the aerosol type (and thus the typical aerosol size distribution) is known, and about a factor of 2 if the aerosol type is not well known or mixtures of different aerosol types prevail..". But not for the rest retrievals (page 7 line 28): "Regarding the aerosol microphysical properties, the comparisons showed that particle number concentrations, surface area, volume and mass concentrations can be obtained with an uncertainty of 25-50% (see Table 1) ..".

**So, we improved this in Sect. 2.2, by adding some sentences on this.**